# Kinetic modeling studies of SOA formation from $\alpha$-pinene ozonolysis

Kathrin Gatzsche[1], Yoshiteru Iinuma[1,*], Andreas Tilgner[1], Anke Mutzel[1], Torsten Berndt[1], and Ralf Wolke[1]

[1]Leibniz Institute for Tropospheric Research (TROPOS), Leipzig, Germany
[*]now at: Okinawa Institute of Science and Technology Graduate University (OIST), Okinawa, Japan

*Correspondence to:* K. Gatzsche (gatzsche@tropos.de)

**Abstract.** This paper describes the implementation of a kinetic gas-particle partitioning approach used for the simulation of secondary organic aerosol (SOA) formation within the SPectral Aerosol Cloud Chemistry Interaction Model (SPACCIM). The kinetic partitioning considers the diffusion of organic compounds into aerosol particles and the subsequent chemical reactions in the particle phase. The basic kinetic partitioning approach is modified by the implementation of chemical backward reaction of the solute within the particle phase as well as a composition dependent particle-phase bulk diffusion coefficient. The adapted gas-phase chemistry mechanism for $\alpha$-pinene oxidation has been updated due to the recent findings related to the formation of highly oxidized multifunctional organic compounds (HOMs). Experimental results from a LEAK (Leipziger Aerosolkammer) chamber study for $\alpha$-pinene ozonolysis were compared with the model results describing this reaction system.

The performed model studies reveal that the particle-phase bulk diffusion coefficient and the particle phase reactivity are key parameters for SOA formation. Using the same particle phase reactivity for both cases we find that liquid particles with higher particle-phase bulk diffusion coefficients have 310-times more organic material formed in the particle phase compared to higher viscous semi-solid particles with lower particle-phase bulk diffusion coefficients. The model results demonstrate that, even with a moderate particle phase reactivity, about 61 % of the modeled organic mass consists of reaction products that are formed in the liquid particles. This finding emphasizes the potential role of SOA processing. Moreover, the initial organic aerosol mass concentration and the particle radius are of minor importance for the process of SOA formation in liquid particles. A sensitivity study shows that a 22-fold increase in particle size merely leads to a SOA increase of less than 10 %.

Due to two additional implementations, allowing backward reactions in the particle phase and considering a composition dependent particle-phase bulk diffusion coefficient, the potential overprediction of the SOA mass with the basic kinetic approach is reduced by about 40 %. HOMs are an important compound group in the early stage of SOA formation because they contribute up to 65 % of the total SOA mass at this stage. HOMs also induce further SOA formation by providing an absorptive medium for SVOCs (semi-volatile organic compounds). This process contributes about 27 % of the total organic mass. The model results are very similar to the LEAK chamber results. Overall, the sensitivity studies demonstrate that the particle reactivity and the particle-phase bulk diffusion require a better characterization in order to improve the current model implementations and to validate the assumptions made from the chamber simulations.

The successful implementation and testing of the current kinetic gas-particle partitioning approach in a box model framework will allow further applications in a 3-D model for regional scale process investigations.

## 1 Introduction

The Earth's radiative budget is directly and indirectly influenced by atmospheric aerosols due to scattering and absorption of solar as well as terrestrial radiation (Haywood and Boucher, 2000; IPCC, 2013). Additionally, atmospheric aerosols affect cloud formation and provide an interface for heterogeneous chemical reactions (Andreae and Crutzen, 1997). Aerosols also markedly affect human health, in particular, respiratory and cardiovascular systems can be damaged due to exposure to aerosol particles (Harrison and Yin, 2000; Pope and Dockery, 2006). In many locations, organic matter (OM) represents the largest fraction of the aerosol mass, whereby the aerosol fine fraction consists of up to about one half of secondary organic aerosol (SOA, Jimenez et al., 2009; Hallquist et al., 2009). SOA can be divided into two main types; gasSOA (Hallquist et al., 2009) that is formed from gas to particle conversion and aqSOA (Ervens et al., 2011) that is formed from aqueous phase processes. This study investigates only gasSOA and will only be referred to as SOA henceforth.

SOA is formed via the oxidation of gas-phase organic precursor compounds by ozone, hydroxyl radical, nitrate radical, or photolysis (Hallquist et al., 2009). Thereby, the functionalization and binding ability of the product molecules result in decreased volatility (Donahue et al., 2012). If the vapor pressure of the oxidized organic compounds is low enough, they condense on a pre-existing particle surface or form new particles through nucleation. SOA formation is very complex because of the variety of organic precursor compounds in the atmosphere and numerous atmospheric degradation pathways. Due to the complexity no comprehensive mechanistic knowledge, including a complete particle product distribution, exits (Goldstein and Galbally, 2007). The modeling approach, which is mainly utilized for gas-to-particle phase partitioning of semi-volatile organic compounds, based on gas-particle equilibrium for these compounds, was proposed by Pankow (1994). The phase states of SOA particles have been investigated in more detail by a wide variety of measurement techniques recently. First indications of an amorphous (semi-)solid state for biogenic SOA particles were deduced by Virtanen et al. (2010) and were established in following studies (Virtanen et al., 2011; Cappa and Wilson, 2011; Vaden et al., 2011; Saukko et al., 2012; Zelenyuk et al., 2012). These studies have provided a first qualitative estimate of the particle phase state, however, quantitative values such as the particle-phase bulk diffusion coefficients are essential for model description. For this purpose, new measurement techniques of particle viscosity for small sample masses have been developed and verified (Zelenyuk et al., 2012; Renbaum-Wolff et al., 2013b, a; Kuimova, 2012; Hosny et al., 2013). Bulk viscosity measurements demonstrate that SOA particles only exist at a high relative humidity (RH > 75 %) in a liquid state (Renbaum-Wolff et al., 2013a; Kidd et al., 2014). At a lower relative humidity, the organic particles exhibit a higher viscosity indicating a semi-solid or glassy phase state (Renbaum-Wolff et al., 2013a; Abramson et al., 2013; Pajunoja et al., 2014; Zhang et al., 2015; Grayson et al., 2016). Additionally, Grayson et al. (2016) have shown that viscosity increases when the production mass concentration of SOA decreases, however, the experimentally formed SOA mass exceeded atmospheric mass concentrations. Recently, microviscosity measurements of SOA particles have revealed a high intra-molecular heterogeneity of SOA particles concerning their phase state (Hosny et al., 2016). The measured micro-

viscosity values provided by Hosny et al. (2016) are lower than the results of bulk measurements reported by Renbaum-Wolff et al. (2013a), Zhang et al. (2015), and Grayson et al. (2016). However, the data by Hosny et al. (2016) fit well with particle-phase diffusion coefficient measurements by Price et al. (2015), which were converted by the Stokes-Einstein (SE) relation to viscosity. The break-down of the SE relation for supercooled liquids and glasses is well known from literature (Tarjus and Kivelson, 1995; Mendoza et al., 2015). This fact is also reported near the glass transition temperature for sugars (Champion et al., 1997; Rampp et al., 2000; Zhu et al., 2011; Power et al., 2013) and protein (Shiraiwa et al., 2011). All measurements indicate that the viscosity of SOA particles increases with decreasing relative humidity and that a semi-solid phase state is very likely achieved. Therefore, absorptive partitioning approaches, which are based on steady equilibrium between gas and particle phase (Pankow, 1994), may indirectly overestimate particle-phase diffusion and thus the SOA formation due to continuous partitioning.

A significant progress in the development of non-equilibrium partitioning models has been made in recent years, whereby the particle phase is resolved in different ways. Shiraiwa et al. (2010) presented KM-SUB, a novel model approach to resolve the particle phase in multiple layers, and interface the gas phase and particle phase with the introduction of a sorption layer and a surface layer. For the description of the fluxes between the many layers of the particle, diffusion coefficients for the gas and the particle phase are necessary. Thus, the phase state of the aerosol particles will influence the partitioning of gaseous compounds. The model was further developed to KM-GAP (Shiraiwa et al., 2012, 2013a, b). Later, Roldin et al. (2014) provided a second multi-layer model, ADCHAM, which has a similar layer structure to KM-SUB/KM-GAP and has been developed for the simulation of chamber studies. A third model approach with a kinetic description of VOC partitioning suitable for regional and global atmospheric models has been recently proposed by Zaveri et al. (2014) and it has been verified in MOSAIC (Model for Simulating Aerosol Interactions and Chemistry). In this approach, the particle phase is not resolved in different layers, but bulk particle-phase diffusion is considered for the solute semi-volatile species. The utilization of particle-phase bulk diffusion coefficients in partitioning approaches increases the degree of freedom for the phase transfer and, therefore, the particle-phase bulk diffusion coefficient needs to be estimated.

Within this study, the kinetic partitioning approach by Zaveri et al. (2014) have been applied and further developed. Therefore, the kinetic partitioning approach was deployed the first time to a comprehensive gas-phase chemistry mechanism, describing $\alpha$-pinene ozonolysis and box model simulations have been achieved for sensitivity and chamber studies. Since the kinetic partitioning is a more complex approach than the absorptive partitioning (Pankow, 1994), we conducted extensive sensitivity studies to explore the influence of the individual parameters on SOA formation. Particularly, particle-phase bulk diffusion co-efficients, mass accommodation coefficients, and rate constants for particle phase reactions in the way of oligomerization are uncertain or less characterized for SOA particles. Therefore, sensitivity studies were conducted to reveal their influence on SOA formation. In addition to these more technical studies, two further investigations were carried out to study the influence of highly oxidized multifunctional organic compounds (HOMs) and a composition dependent particle-phase bulk diffusion coefficient. HOMs have been successfully identified in laboratory and field studies recently (Ehn et al., 2012, 2014; Zhao et al., 2013; Jokinen et al., 2014; Mentel et al., 2015; Mutzel et al., 2015; Berndt et al., 2016). Their possible existence was already proposed in 1998 (Kulmala et al., 1998), but their influence on the early growth of fresh SOA particles is the subject of on-

going investigations (Riipinen et al., 2012; Donahue et al., 2012, 2013). The consideration of HOMs in gas-phase chemistry mechanisms seems to be indispensable because the total molar HOM yield from the reactions of $\alpha$-pinene with OH as well as $O_3$ is about 6 % (Berndt et al., 2016), also the predicted vapor pressures of HOMs are rather low (Kurtén et al., 2016). Thus, in the second part of this modeling study, the gas-phase chemistry mechanism has been extended to include the measured HOM

yields for $\alpha$-pinene ozonolysis in order to examine their influence on the initial formation of SOA and the overall SOA yield. The second investigation focuses on the importance of the particle-phase bulk diffusion coefficient of SOA particles for the overall SOA mass yield. The particle-phase bulk diffusion coefficient might be rather composition dependent than constant and due to a lower self-diffusion coefficient of the organic material, increasing organic matter in the particle phase decreases the weighted particle-phase bulk diffusion coefficient. This investigation is also important for modeling of chamber experiments

where wet seed aerosols are often used because water is known to have a plasticizer effect on SOA (O'Meara et al., 2016). The implementation of a composition dependent particle-phase bulk diffusion coefficient within the kinetic approach of Zaveri et al. (2014) constitutes a further development of the basic approach and the applicability of this new feature is tested in this study. Moreover, this study provides how a composition dependent particle-phase bulk diffusion coefficient can be applied and how it influences the SOA formation. The second further development of the kinetic approach concerns the particle-phase

reactivity. Additional backward reactions in the particle phase have been considered to enable a treatment of reversible particle-phase reactions under formation of a reaction equilibrium. The effect of this process is also subject of the extensive sensitivity studies.

## 2    Material and methods

### 2.1    Chamber experiments

$\alpha$-Pinene ozonolysis experiments were carried out in the aerosol chamber LEAK (Leipziger Aerosolkammer). A detailed description of the LEAK chamber together with the available equipment can be found in Iinuma et al. (2009) and Mutzel et al. (2016). Briefly, LEAK is a cylindrical, $19\,m^3$ Teflon bag with a surface-to-volume ratio of $2\,m^{-1}$. The experiments were performed in the presence of ammonium sulfate seed particles, which aerosol size distribution span a narrow range around a mean particle radius of 35 nm. $O_3$ was generated by UV irradiation of $O_2$ using an $O_2$ flow rate of $3\,L\,min^{-1}$. $\alpha$-Pinene

($\approx$ 58 ppb) was injected into the LEAK chamber with a microliter-syringe. The oxidation of $\alpha$-pinene was carried out under 55 % relative humidity (RH). Carbon monoxide (CO) was used during the experiment as an OH scavenger. The $\alpha$-pinene consumption was monitored with a proton-transfer-reaction mass spectrometer (PTR-MS) over a reaction time of 2 hours. Throughout the chamber experiment, the particle-size distribution was measured every 11.5 min with a scanning mobility particle sizer (SMPS). Ozone was monitored with a 60 s time resolution. An average density of $1\,g\,cm^{-3}$ was applied to

calculate the produced organic particle mass ($\Delta$OM). The particle phase was sampled on filters at the end of the experiments and analyzed afterwards by an Ultra Performance Liquid Chromatography coupled to Electrospray Ionisation Time-of-Flight Mass Spectrometer (UPLC/ESI(-)-TOFMS, Waters, MA, USA). $1.8\,m^3$ of the chamber volume was collected on a PTFE filter (borosilicate glass fiber filter coated with fluorocarbon, 47 mm in diameter, PALLFLEX T60A20, PALL, NY, USA), which was

connected to a denuder (URG-2000-30B5, URG Corporation, Chapel Hill, NC, USA, Kahnt et al., 2011) to avoid gas-phase artifacts. Furthermore, PTR-MS measurements were performed in order to monitor the key oxidation products of the $\alpha$-pinene ozonolysis, such as pinonaldehyde.

## 2.2 Model framework

### 2.2.1 Box model SPACCIM (original code)

This study uses the SPectral Aerosol Cloud Chemistry Interaction Model (SPACCIM, Wolke et al., 2005) and has extended it by implementing a particle phase partitioning approach. The original SPACCIM version published by Wolke et al. (2005) is an adiabatic air parcel model incorporating a complex size-resolved cloud microphysical model and a multiphase chemistry model. Briefly, the interaction between both models is implemented by a coupling scheme. This coupling enables both models to run independently, as far as possible, and to apply their own time step control. The microphysics scheme of the SPACCIM model framework is based on the work of Simmel and Wurzler (2006) and Simmel et al. (2005). The cloud microphysical model describes the growth and shrinking processes of aerosol particles by water vapor diffusion as well as nucleation and growth/evaporation of cloud droplets or deliquescent particles. Other microphysical processes such as impaction of aerosol particles and collision/coalescence of droplets are also explicitly considered. The implementation of the dynamic growth rate in the condensation/evaporation processes and the droplet activation is implemented based on the Köhler theory (Köhler, 1936). The size-resolved cloud microphysics of deliquesced particles and droplets including cloud droplet formation, evolution, and evaporation is considered using a one-dimensional sectional approach. Further microphysical features of SPACCIM are already described in Wolke et al. (2005) and results owing to these processes are presented in Tilgner et al. (2013); Rusumdar et al. (2016); Hoffmann et al. (2016). The implemented multiphase chemical model applies a high-order implicit time integration scheme, which utilizes the specific sparse structure of the model equations (Wolke and Knoth, 2002). SPACCIM was originally developed for parcel model studies, whereby, the considered air parcel can follow real or artificial trajectories. However, the partitioning of organic gases towards the particle phase was not considered in the original SPACCIM and the model was not exclusively designed for application on aerosol chamber studies. The existing model framework has been extended by gas-to-particle mass transfer via a kinetic partitioning approach (Zaveri et al., 2014), see Sect. 2.2.2 for details. Due to the focus of these studies on modeling aerosol chamber studies of gasSOA formation for monodisperse aerosol without entrainment and coagulation, microphysical processes are not included in the results of this study.

### 2.2.2 Implementation of a kinetic partitioning approach

The existing model framework was extended by the implementation of the kinetic partitioning approach established by Zaveri et al. (2014). The general assumption of this approach is based on the description of the diffusive flux of a solute in the particle phase via Fick's second law extended by a particle phase reaction of the solute:

$$\frac{\partial A_i(r,t)}{\partial t} = D_{\mathrm{b},i} \frac{1}{r^2} \frac{\partial}{\partial r}\left(r^2 \frac{\partial A_i(r,t)}{\partial r}\right) - k_{\mathrm{c},i} A_i(r,t). \tag{1}$$

Thereby, the utilized parameters are the particle-phase concentration $A_i$ of the solute $i$ as a function of the radius $r$ and the time $t$, the particle-phase bulk diffusion coefficient of the solute $D_{b,i}$, and the chemical reaction rate constant $k_{c,i}$ of the solute within the particle phase. Equation (1) is given in spherical coordinates. As a fundamental simplification the diffusion coefficient is assumed to be constant. Therefore, a particle-phase bulk diffusion coefficient $D_{b,i}$ is introduced. This assumption simplifies the calculation of the integral (Eq. 1). In Zaveri et al. (2014) two solutions of Eq. (1) are described. The first solution is only for usage in a Lagrangian box model in case of a "closed system". The second approach is applicable to general systems treated with 3-D Eulerian models. Therefore, the distinction of two reaction regimes concerning their reaction rates and the application of the two-film theory (Lewis and Whitman, 1924) in case of slow reactions is utilized. For a polydisperse aerosol distribution, the following equations are proposed by Zaveri et al. (2014) for fast reactions ($k_{c,i} \geq 0.01\,\mathrm{s}$):

$$\frac{\mathrm{d}\overline{C}_{\mathrm{a},i,m}}{\mathrm{d}t} = \xi_m k_{\mathrm{g},i,m} \left( \overline{C}_{\mathrm{g},i} - \overline{C}_{\mathrm{a},i,m} \frac{S_{i,m}}{Q_i} \right) - k_{\mathrm{c},i} \overline{C}_{\mathrm{a},i,m}, \tag{2}$$

and for slow reactions ($k_{c,i} < 0.01\,\mathrm{s}$):

$$\frac{\mathrm{d}\overline{C}_{\mathrm{a},i,m}}{\mathrm{d}t} = \xi_m K_{\mathrm{g},i,m} \left( \overline{C}_{\mathrm{g},i} - \overline{C}_{\mathrm{a},i,m} S_{i,m} \right) - k_{\mathrm{c},i} \overline{C}_{\mathrm{a},i,m}, \tag{3}$$

whereby, $\xi_m$ denotes the surface area of the respective size-section $m$:

$$\xi_m = 4\pi r_{\mathrm{p},m}^2 N_m, \tag{4}$$

and $S_{i,m}$ is the saturation ratio:

$$S_{i,m} = \frac{C_{\mathrm{g},i}^*}{\sum_j \overline{C}_{\mathrm{a},j,m}}. \tag{5}$$

Therein, $\overline{C}_{\mathrm{a},i,m}$ denotes the total average concentration of a solute $i$ in size-section $m$, with $M$ the number of size-sections. $Q_i$ represents the ratio of the average particle-phase concentration $\overline{A}_i$ to the surface concentration $A_i^{\mathrm{S}}$ at steady-state and is named quasi-steady-state term (see Appendix A). $N_m$ denotes the number concentration, $r_{\mathrm{p},m}$ the respective particle radius, $k_{\mathrm{g},i}$ is the gas-side mass transfer coefficient, and $K_{\mathrm{g},i}$ is the overall gas-side mass transfer coefficient, which is needed for the application of the two film theory (see Appendix A for details). The saturation ratio $S_{i,m}$ (Eq. 5) represents the ratio of the effective saturation vapor concentration $C_{\mathrm{g},i}^*$ and the total average organic aerosol mass concentration $\sum_j \overline{C}_{\mathrm{a},j,m}$. A more detailed derivation of Eq. (2) and (3) is provided in Appendix A following Zaveri et al. (2014).

The total particle-phase concentration in this kinetic approach depends on the gas-phase concentration of the condensable organic compounds, the transport of these compounds to the particle surface and the transport into the particle. As demonstrated by Mai et al. (2015), three different limitations for SOA formation exist for a kinetic approach: gas-phase-diffusion-limited partitioning, interfacial-transport-limited partitioning, and particle-phase-diffusion-limited partitioning. Liquid particles are almost well-mixed, inferring a nearly uniform concentration profile of the organic solutes in the particle phase. However for SOA modeling considering semi-solid or solid particles, a concentration gradient from the surface to the particle inner is established for the organic solutes due to the longer diffusion times. Thus, the organic solutes mainly concentrate on the particle

surface and the condensation of organic compounds is performed according to Raoult's law. The distribution of the organic solutes in the particle can be calculated as steady-state solution of Eq. (1). Accordingly, the total organic mass in the particle bulk is given as the integral of the solute concentration over the sphere volume (see Eq. (A3) in Appendix A). SPACCIM only treats these bulk concentrations without consideration of a particle layer structure or a separate surface layer. Therefore, the

5 representation of buried molecules and their related processes (Perraud et al., 2012; Zhou et al., 2013) are not comprised by SPACCIM.

The mass balance equation in SPACCIM for the gas phase after implementation of the gas-to-particle transfer via the kinetic approach yields to:

$$
\frac{dC_{g,i^*}}{dt} = \underbrace{R_{g,i^*}\left(t, C_{g,1}, \ldots, C_{g,N_g}\right)}_{\text{I: gas phase chemistry}} - \underbrace{\kappa_i \sum_m L_m k_{t,m,i} \left[C_{g,i^*} - \frac{C_{a,m,i}}{H_i}\right]}_{\substack{\text{II: gas-to-aqueous-phase} \\ \text{mass transfer}}} - \underbrace{\lambda_i \sum_m \xi_m k^\circ_{g,m,i} \left[C_{g,i^*} - C_{a,m,i} S^\circ_{m,i}\right]}_{\substack{\text{III: gas-to-particle-phase} \\ \text{mass transfer}}}
$$

$$
+ \underbrace{\mu \left[c_{g,i^*} - c_{g,\text{ent}}\right]}_{\substack{\text{IV: entrainment/} \\ \text{outflow}}}, \tag{6}
$$

$$
i^* = 1, \ldots, N_g; \ i = 1, \ldots, N_{\text{aq}}, N_{\text{aq}+1}, \ldots, N_a; \ m = 1, \ldots, M,
$$

with

$$
k^\circ_{g,m,i} = k_{g,m,i} \ \text{and} \ S^\circ_{m,i} = \frac{S_{i,m}}{Q_i} \quad (k_{c,i} \geq 0.01\,\text{s}) \quad \text{or} \quad k^\circ_{g,m,i} = K_{g,m,i} \ \text{and} \ S^\circ_{m,i} = S_{i,m} \quad (k_{c,i} < 0.01\,\text{s}). \tag{7}
$$

The mass balance equation for the particle phase is given by:

$$
\frac{dC_{a,i}}{dt} = \underbrace{L_m R_{a,i}\left(t, C_{a,1}, \ldots, C_{a,N_{\text{aq}}}, C_{a,N_{\text{aq}+1}}, \ldots, C_{a,N_a}\right)}_{\text{I: aqueous and particle phase chemistry}} + \underbrace{\kappa_i L_m k_{t,m,i}\left[C_{g,i^*} - \frac{C_{a,m,i}}{H_i}\right]}_{\substack{\text{II: gas-to-aqueous-phase} \\ \text{mass transfer}}} + \underbrace{\lambda_i \xi_m k^\circ_{g,m,i}\left[C_{g,i^*} - C_{a,m,i} S^\circ_{m,i}\right]}_{\substack{\text{III: gas-to-particle-phase} \\ \text{mass transfer}}}
$$

$$
+ \underbrace{F\left(c_{1,i} - c_{M,i}\right)}_{\substack{\text{IV: mass transfer} \\ \text{by microphysics}}} + \underbrace{\mu \left[c_{m,i} - c_{m_{\text{ent}},i}\right]}_{\substack{\text{V: entrainment/} \\ \text{outflow}}}. \tag{8}
$$

Here, $R_{g,i^*}$ stands for all chemical reactions which take place in the gas phase (compounds $i^* = 1, \ldots, N_g$) and $R_{a,i}$ for chemical reactions in the aerosol phase, which comprises aqueous $i = 1, \ldots, N_{\text{aq}}$ and particulate $i = N_{\text{aq}+1}, \ldots, N_a$ species. The second term on the right side of Eq. (6) describes the mass transfer in the aqueous aerosol phase (compounds $i = 1, \ldots, N_{\text{aq}}$)

20 utilizing the Schwartz approach (Schwartz, 1986) with the dimensionless Henry's law coefficient $H_i$ and the mass transfer coefficient $k_{t,m,i}$:

$$
k_{t,m,i} = \left(\frac{r^2_{p,m}}{3D_g} + \frac{4r_{p,m}}{3\nu\alpha_{\text{aq},i}}\right)^{-1}. \tag{9}
$$

Thereby, $D_g$ denotes the gas diffusion coefficient, $\nu$ the molecular speed, and $\alpha_{\text{aq},i}$ the mass accommodation coefficient for the aqueous compounds ($0 \leq \alpha_{\text{aq},i} \leq 1$). The prefactor $\kappa_i$ of the Henry term denotes the solubility index (1 means soluble

and 0 insoluble) of the aqueous solute. $L_m$ indicates the volume fraction $[V_m/V_{\text{box}}]$ of the $m$-th droplet class inside the box volume and $C_{a,m,i}$ denotes the corresponding aqueous aerosol phase concentration in the $m$-th droplet class. The parameters in the gas-to-particle mass transfer term are described above, whereby similar to the solubility index a partitioning index $\lambda_i$ (1 for partitioning in the particle phase and 0 for staying in the gas phase) is introduced for the individual compounds ($i = N_{\text{aq}+1}, \ldots, N_a$). There is the possibility to describe time-dependent entrainment/detrainment by the rate $\mu_i$ for the different gas-phase (Eq. 6, term IV) and aerosol-phase (Eq. 8, term V) species. For the aerosol species, a mass transfer between the different droplet classes is also possible due to microphysical processes (e.g., by aggregation, break up, and condensation), which is described by term $F$ in Eq. (8, term IV). In the present study, the terms of entrainment/detrainment in Eq. (6) and (8) can be neglected, because no entrainment is considered in the closed system of the box model simulations. Additionally, the terms of aqueous-phase transfer in the particle phase (Eq. 6 and 8, term II) are omitted because aqSOA formation is not considered in this SPACCIM model study.

### 2.2.3 Gas-phase chemistry mechanism

The basic gas-phase chemistry mechanism for the degradation of $\alpha$-pinene, that is suitable for a box model or regional scale model, has been proposed by Chen and Griffin (2005). The oxidation mechanisms take into account the organic degradation protocol established by Jenkin et al. (1997) as well as experimental results and formation pathways of SVOCs (Jenkin et al., 2000; Winterhalter et al., 2000). The host gas-phase chemistry mechanism for the $\alpha$-pinene oxidation is the Caltech Atmospheric Chemistry Mechanism (CACM, Griffin et al., 2002; Pun et al., 2003).

The oxidation mechanism for $\alpha$-pinene degradation underlies a number of simplifications to reduce the amount of species and associated reactions in the model. The reduced computing time will allow the use of this mechanism in future 3-D model studies. Therefore, the mechanism does not include the reactions on the carbon positions in hydrocarbon skeletons that are less likely to be attacked by the OH radical (Chen and Griffin, 2005). Peroxy radicals ($RO_2$) are explicitly considered in the mechanism if their further chemical reactions can lead to the formation of multifunctional carbonyl compounds that may form SVOCs (Chen and Griffin, 2005; Griffin et al., 2002). Moreover, peroxy radicals are summed up in the model to simplify $RO_2$ cross permutations and self-reactions. Further details concerning the gas-phase chemistry mechanism are provided in the publication of Chen and Griffin (2005). The utilized chemical degradation reactions for $\alpha$-pinene are compared with the MCM v3.3.1 (Master Chemical Mechanism) and the reaction rate coefficients have been updated, if necessary. Additionally, the recently measured HOM yields (Berndt et al., 2016) have been implemented in the existing gas-phase chemistry mechanism recalculating the former branching ratios for the last part of the sensitivity studies. The degradation reactions for $\alpha$-pinene are summarized in Table S1 in the Supplement.

The vapor pressures of the SVOCS, which are assumed to partition into the particle phase, have been estimated with the group contribution method called EVAPORATION described by Compernolle et al. (2011). The tool is available online at the following URL (http://tropo.aeronomie.be/models/evaporation_run.htm). The vapor pressures of the HOMs from the reaction of $\alpha$-pinene with OH are provided by Berndt et al. (2016) and were therein calculated with the COSMO-RS approach (Eckert and

Klamt, 2002) as well as compared with estimates from SIMPOL (Pankow and Asher, 2008). Additionally, HOM vapor pressures are calculated with EVAPORATION for a comparison with the COSMO-RS values (see Table S4 in the Supplement for details). Up to now, the structural formulas of the HOMs from the $\alpha$-pinene ozonolysis are not known from quantum chemical calculations. The HOM molecules formed from the reaction of $\alpha$-pinene with $O_3$ might contain more functional groups than

for the reaction of $\alpha$-pinene with OH. Therefore, the HOMs from the ozone reaction pathway might have lower vapor pressures as indicated in the study of Kurtén et al. (2016) concerning vapor pressures of HOMs.

## 2.3    Performed sensitivity studies

According to Eqs. (2) and (3) the formed SOA mass predominantly depends on the following parameters: particle-phase bulk diffusion coefficient $D_\mathrm{b}$, pseudo-first-order rate constant for particle phase reactions $k_\mathrm{c}$, particle radius $r_\mathrm{p}$, initial particle phase

organic mass concentration $OM_0$, and the mass accommodation coefficient $\alpha$. Sensitivity studies have been performed to characterize the influence of these five parameters on the SOA formation. This study has been limited to a single size-section $m$ of particles. Polydisperse test cases have been performed, but for the conducted sensitivity studies, the consideration of a polydisperse aerosol distribution will increase the degree of freedom as well as the complexity. Further, for the simulation of the LEAK chamber studies, this feature was not required because of the nearly monodisperse aerosol spectrum existent

within this type of experiment. For the sake of clarity, the results of the sensitivity studies regarding the mass accommodation coefficient $\alpha$ and initial particle phase organic mass concentration $OM_0$ are presented in the Supplement (see Sects. 2.2 and 2.3 in the Supplement, respectively). An overview of the parameters evaluated in the sensitivity studies is given in Table 1. The case studies $6-8$ of Table 1 refer to the consideration of HOMs in the gas-phase chemistry mechanism and two further developments of the kinetic partitioning approach.

## 20    3    Results and discussion

### 3.1    Sensitivity studies

#### 3.1.1    Impact of the particle-phase bulk diffusion coefficient $D_\mathrm{b}$ on SOA formation

To investigate the influence of the particle phase state on SOA formation, the particle-phase bulk diffusion coefficient $D_\mathrm{b}$ was varied from $10^{-9}\,\mathrm{m^2\,s^{-1}}$ to $10^{-21}\,\mathrm{m^2\,s^{-1}}$ (see Table 1, study 1). This range covers liquid particles, e.g. droplets with dissolved salts associated with $D_\mathrm{b}=10^{-9}\,\mathrm{m^2\,s^{-1}}$, and semi-solid/viscous particles starting at about $D_\mathrm{b} < 10^{-14}\,\mathrm{m^2\,s^{-1}}$ and ending up

with $D_\mathrm{b} = 10^{-21}\,\mathrm{m^2\,s^{-1}}$. The latter value corresponds to a particle with the texture of pitch (Koop et al., 2011). The model results for the variation of $D_\mathrm{b}$ are shown in Fig. 1. Although, $D_\mathrm{b}$ is varied over thirteen orders of magnitude, the sensitivity study reveals that in general three main regimes for SOA formation occur. For liquid particles, $10^{-9}\,\mathrm{m^2\,s^{-1}} \geq D_\mathrm{b} \geq 10^{-14}\,\mathrm{m^2\,s^{-1}}$, the highest SOA mass is observed. Thereby, rapid formation is achieved for $10^{-9}\,\mathrm{m^2\,s^{-1}} \geq D_\mathrm{b} \geq 10^{-13}\,\mathrm{m^2\,s^{-1}}$. A slower

SOA formation is observed for $D_\mathrm{b} = 10^{-14}\,\mathrm{m^2\,s^{-1}}$. However, for long equilibration times (about $20-24$ hours) and fast particle phase reactions (Fig. 1), about $90\,\%$ of the maximum SOA mass is still achieved. Moreover, between $D_\mathrm{b} = 10^{-14}\,\mathrm{m^2\,s^{-1}}$

**Table 1.** Varied model parameters in the sensitivity studies, whereby some cross checks concerning the sensitivity have been conducted.

| # | Main parameter varied | Range | Additional parameter varied | Adjusted values | Remarks |
|---|---|---|---|---|---|
| 1 | $D_b$ in $m^2\,s^{-1}$ | $10^{-9} - 10^{-21}$ | $k_c$ in $s^{-1}$ | $1, 10^{-1}, 10^{-2}, 10^{-4}, 10^{-6}$ | $r_p = 35\,nm$; $OM_0 = 5.8 \times 10^{-2}\,g\,g^{-1}$ |
| 2 | $k_c$ in $s^{-1}$ | $1 - 10^{-6}$ | $D_b$ in $m^2\,s^{-1}$ | $10^{-12}, 10^{-14}, 10^{-18}$ | $r_p = 35\,nm$; $OM_0 = 5.8 \times 10^{-2}\,g\,g^{-1}$ |
| 3a | $\alpha$ [§] (dimensionless) | $1 - 10^{-2}$ | $D_b$ in $m^2\,s^{-1}$ and | $10^{-12}, 10^{-14}, 10^{-18}$ | $r_p = 35\,nm$; $OM_0 = 5.8 \times 10^{-2}\,g\,g^{-1}$ |
| 3b | | | $k_c$ in $s^{-1}$ | $1, 10^{-1}, 10^{-2}, 10^{-4}$ | $r_p = 35\,nm$; $OM_0 = 5.8 \times 10^{-2}\,g\,g^{-1}$ |
| 4a | $r_p$ in nm | $11 - 240$ | $k_c$ in $s^{-1}$ and | $1, 10^{-1}, 10^{-2}, 10^{-4}, 10^{-6}$ | $OM_0 = 5.8 \times 10^{-2}\,g\,g^{-1}$ |
| 4b | | | $D_b$ in $m^2\,s^{-1}$ | $10^{-12}, 10^{-14}, 10^{-18}$ | $OM_0 = 5.8 \times 10^{-2}\,g\,g^{-1}$ |
| 5 | $OM_0$ [§] in $g\,g^{-1}$ | $10^{-5} - 5.8 \times 10^{-2}$ | $D_b$ in $m^2\,s^{-1}$ | $10^{-12}, 10^{-14}, 10^{-18}$ | $r_p = 35\,nm$ |
| 6a | HOMs considered[†] | | $k_c$ in $s^{-1}$ and | $1, 10^{-2}, 10^{-4}, 10^{-6}$ | $r_p = 35\,nm$ and |
| 6b | | | $D_b$ in $m^2\,s^{-1}$ | $10^{-12}, 10^{-14}, 10^{-18}$ | $OM_0 = 10^{-6}\,g\,g^{-1}$ |
| 7 | $k_b$ in $s^{-1}$ [*] | $1 - 10^{-6}$ | $k_c$ in $s^{-1}$ | $1, 10^{-2}, 10^{-4}$ | $D_b = 10^{-12}\,m^2\,s^{-1}$, $r_p = 35\,nm$ |
| 8 | Weighted particle-phase bulk diffusion coefficient | $D_{org} = 10^{-12}$ or $10^{-14}\,m^2\,s^{-1}$ | $k_b$ in $s^{-1}$ | $10^{-2} - 10^{-6}$ | $k_c = 10^{-2}\,s^{-1}$, $r_p = 35\,nm$ |

[*] Implementation of backward reactions in the particle phase, for more details see Sect. 3.3

[†] See reactions 1b and 3b in Table S1 in the Supplement

[§] Results are presented in the Supplement

and $D_b = 10^{-15}\,m^2\,s^{-1}$, a reduced SOA mass is observed caused by the phase transition from liquid to semi-solid particles in this particle-phase bulk diffusion coefficient range (Koop et al., 2011). This second formation regime is characterized by the longer diffusion time as the particle-phase bulk diffusion coefficients are lower. Accordingly, the equilibration time between the gas and the particle phase is longer. SOA formation is delayed by the reduced diffusion into the particle phase, but can still be observed. The third regime is characterized by an extremely low or no SOA formation. For values $D_b < 10^{-16}\,m^2\,s^{-1}$ the SOA formation is inhibited because the condensed organic material does not diffuse sufficiently into the particle bulk. According to the classification of Mai et al. (2015), this case is named particle-phase-diffusion-limited partitioning. After the

formation of a thin organic shell/film around the particle, no effective SOA formation takes place because of the long mixing time inside the particle. Thus, the gas-phase concentrations of the condensing organic compounds as well as the interfacial transport of these compounds are not the limiting factors of SOA formation under these conditions.

The three observed regimes for the variation of the particle-phase bulk diffusion coefficient are not shifted by the choice of the particle-phase reaction chemical rate constant $k_c$. The formed SOA mass is indeed reduced due to the lower particle phase reaction rate. For $D_b = 10^{-14}\,\mathrm{m^2\,s^{-1}}$ less than 90 % of the maximum SOA mass is achieved (Fig. 1 b). Logarithmic scaled plots for the variation of the particle-phase bulk diffusion coefficient are provided in the Supplement (see Fig. S1 in the Supplement) to illustrate the wide range of modeled SOA.

Viscosity measurements have mostly shown that organic aerosols become semi-solid, when relative humidity drops below $\approx 80/75\,\%$ (Renbaum-Wolff et al., 2013a; Zhang et al., 2015; Grayson et al., 2015). Afterwards, measured viscosity values were converted in diffusion coefficients via the Stokes-Einstein relation (Einstein, 1956). As a result, calculated particle-phase bulk diffusion coefficients for RH < 80 % are lower than $5 \times 10^{-14}\,\mathrm{m^2\,s^{-1}}$ (Renbaum-Wolff et al., 2013a), which implies semi-solid organic aerosols. In the following, the measurement results by Renbaum-Wolff et al. (2013a), Zhang et al. (2015), and Grayson et al. (2016) are discussed from a perspective of the model results that are calculated with different particle-phase bulk diffusion coefficients. The estimated particle-phase bulk diffusion coefficients from measurements imply that the simulated regime, where SOA mass formation is highest, might be relevant for wet conditions only ($\approx \mathrm{RH} > 90\,\%$). The second regime, with intermediate SOA production, could only be reached within a narrow relative humidity range ($\approx 90\,\% > \mathrm{RH} > 70\,\%$). In this range, the transition from liquid to semi-solid particles might occur following the available viscosity measurements (Renbaum-Wolff et al., 2013a; Zhang et al., 2015). We can assume $D_b < 10^{-16}\,\mathrm{m^2\,s^{-1}}$ (Renbaum-Wolff et al., 2013a, results of poke-and-flow technique) for a relative humidity below 70 %. The sensitivity study reveals almost no SOA formation. Considering the relative humidity levels typical for chamber studies (0 to 70 % RH), it is quite difficult to interpret observed SOA formation in line with the stated particle-phase bulk diffusion coefficients (Renbaum-Wolff et al., 2013a) and related model results from the present study. However, under ambient conditions the broad spectrum of particle-phase bulk diffusion coefficients and the linked SOA mass have to be taken into account. A recent investigation concerning SOA phase state using fluorescence lifetime imaging (FLIM) by Hosny et al. (2016) provides a lower microviscosity than bulk viscosity measurements over the full RH-range (Renbaum-Wolff et al., 2013a; Zhang et al., 2015; Grayson et al., 2016). Therefore, local friction of a single particle is lower than bulk friction and hence local diffusion can be more efficient than bulk diffusion.

### 3.1.2 Importance of the pseudo-first-order rate constant of particle phase reactions $k_c$ on SOA formation

The pseudo-first-order rate constant of particle phase reactions $k_c$ has been varied from $1\,\mathrm{s^{-1}}$ to $10^{-6}\,\mathrm{s^{-1}}$ to examine the influence of high and low particle phase reactivity on SOA formation in the second sensitivity study (see Table 1, study 2). As indicated in Eq. (1), the organic compounds, which are partitioned from the gas phase into the particle phase, can further react in the particle phase with a constant reaction rate $k_c$. The from the gas into the particle phase partitioned organic compounds are named p-products. The products, which have been caused due to the reactions in the particle phase, are termed r-products. The r-products do not stay in equilibrium with the gas-phase compounds and, therefore, can not evaporate from the particle

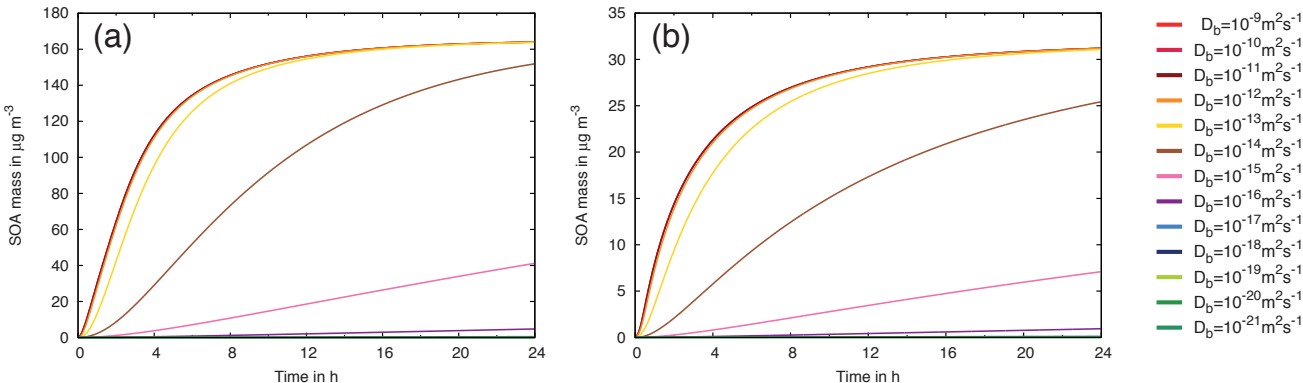

**Figure 1.** Simulated SOA mass for different particle-phase bulk diffusion coefficients in the range of $D_b$: $10^{-9}\,\mathrm{m^2\,s^{-1}} - 10^{-21}\,\mathrm{m^2\,s^{-1}}$ (model case 1 of Table 1) using a particle reaction rate constant of a) $k_c = 1\,\mathrm{s^{-1}}$ and b) $k_c = 10^{-6}\,\mathrm{s^{-1}}$.

phase. The r-products comprise particle-phase compounds resulting from aging of the condensed organic compounds (e.g., dimers, trimers or oligomers). Fig. 2 depicts the results of the particle-phase rate constant variation for liquid and semi-solid particles. A higher rate constant leads to an increased SOA mass independent of the phase state. The pseudo-first-order rate constant can also be understood as a reactive SOA uptake. The efficiency of this reactive uptake is different for liquid and

semi-solid particles. For liquid particles ($D_b = 10^{-12}\,\mathrm{m^2\,s^{-1}}$) the mass enhancement factor is about 6 for a seven order of magnitude higher $k_c$. This means that the SOA mass is increased due to a higher particle phase reactivity, since the organic mass is shifted more effectively from the partitioning products (p-products) to the reacted products (r-products). For semi-solid particles, the kinetic approach is predominantly sensitive for particle-phase reactions as indicated in Zaveri et al. (2014). The particle-phase-diffusion-limited partitioning for the viscous aerosol particles has been step-wise raised due to the increased

particle-phase reaction rate constant (see Fig. 2b). The mass enhancement factor is about 2.5 for a seven orders of magnitude higher reactivity. It is noted that organic aerosol particles might exhibit such low particle-phase diffusivities $D_b \leq 10^{-18}\,\mathrm{m^2\,s^{-1}}$ mainly under dry conditions (RH $< 40\,\%$). As a result, a higher pseudo-first-order rate constant for a chemical particle phase reaction leads to a distinct increase in SOA mass. Processes such as dimerization, oligomerization or other SOA aging processes can be represented in SOA models by means of particle reactions. Rate constants for oligomerization processes are not well

characterized and only estimates from product studies exist. For example, Hosny et al. (2016) simulated the evolution of oleic acid droplets with the kinetic multilayer model KM-SUB (Shiraiwa et al., 2010) including monomer to tetramer products for the ozonolysis of oleic acid, comparing the results with experimental data.

### 3.1.3   Influence of the particle radius $r_p$ on SOA formation

The effect of the variation of the particle radius $r_p$ in the range from 11 nm to 240 nm on SOA formation is analyzed in

this subsection. Additionally, the particle-phase bulk diffusion coefficient and the particle phase reaction rate constant have been varied to characterize the influence of the different parameters on the SOA formation for different particle sizes (see

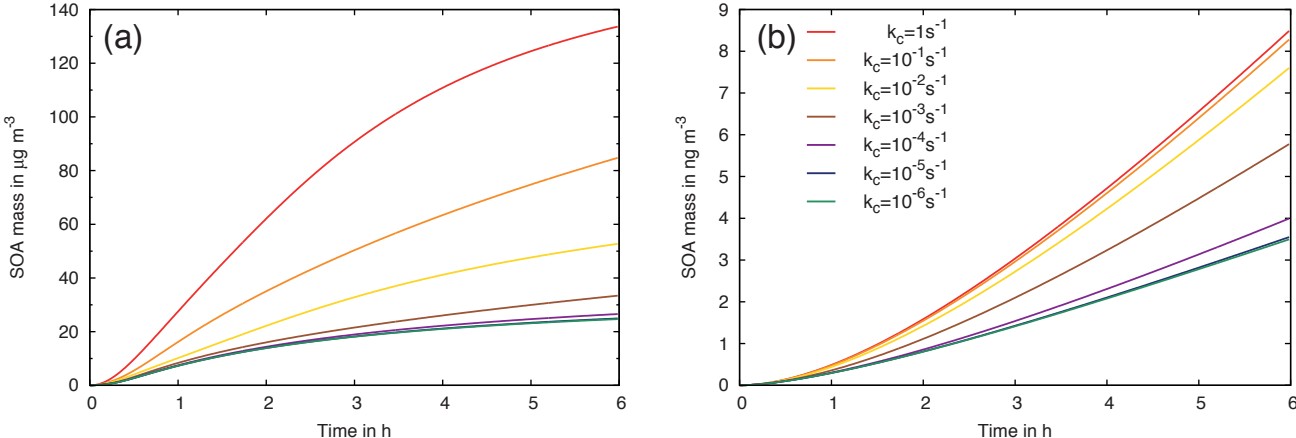

**Figure 2.** Simulated SOA mass for different pseudo-first-order rate constant of particle reactions $k_c$ (model case 2 of Table 1) using particle-phase bulk particle diffusion coefficients for a) liquid ($D_b = 10^{-12}\,\text{m}^2\,\text{s}^{-1}$) and b) semi-solid ($D_b = 10^{-18}\,\text{m}^2\,\text{s}^{-1}$) aerosol particles.

Table 1, model study 4a and 4b). The influence of the particle radius on the SOA formation is moderate for liquid aerosol particles ($D_b = 10^{-12}\,\text{m}^2\,\text{s}^{-1}$) when particle phase reactions are fast (Fig. 3a). For an about 22-fold larger particle, the SOA mass increase is by less than 10 %. The maximum formed SOA mass is comparatively low for semi-solid particles and the particle size determines whether organic compounds condense or not (Fig. 3b). Therefore, the influence of the particle radius is higher for semi-solid particles. However, the formed SOA mass is rather small due to the limited diffusion into the particle and it should be kept in mind that this diffusion coefficient is limited to dry conditions. For liquid particles, the increase in SOA mass caused by the larger particle size is for moderate particle-phase reactions about 40 % (see Fig. S5a in the Supplement). For semi-solid particles and moderate particle reactions, the same trend is observed but only very low SOA concentrations are formed (see Fig S5b in the Supplement). Due to the particle-phase-diffusion-limited partitioning for semi-solid particles combined with only moderate particle-phase rate constants, the formed SOA mass reaches quite small values, which are not observed under typical chamber study experiment conditions. Furthermore, the normalized increase of organic mass is highest for the smallest particles (Fig. S4a and S4b). This means smaller particles are characterized by a more effective uptake of organic mass related to the initial organic aerosol mass. This finding agrees with the surface ratio between smaller and larger particles. The influence of the particle radius $r_p$ on SOA formation is characterized as moderate, but noticeable, particularly for semi-solid particles. The influence of the particle size is considerably smaller compared to the particle-phase bulk diffusion coefficient or the particle reactivity.

### 3.2 Importance of HOMs for initial SOA formation

Recently, ELVOCs (Ehn et al., 2014; Jokinen et al., 2015) and HOMs (Mutzel et al., 2015; Berndt et al., 2016) have been found to play a major role in aerosol formation from monoterpenes. Mutzel et al. (2015) have demonstrated the existence of HOMs

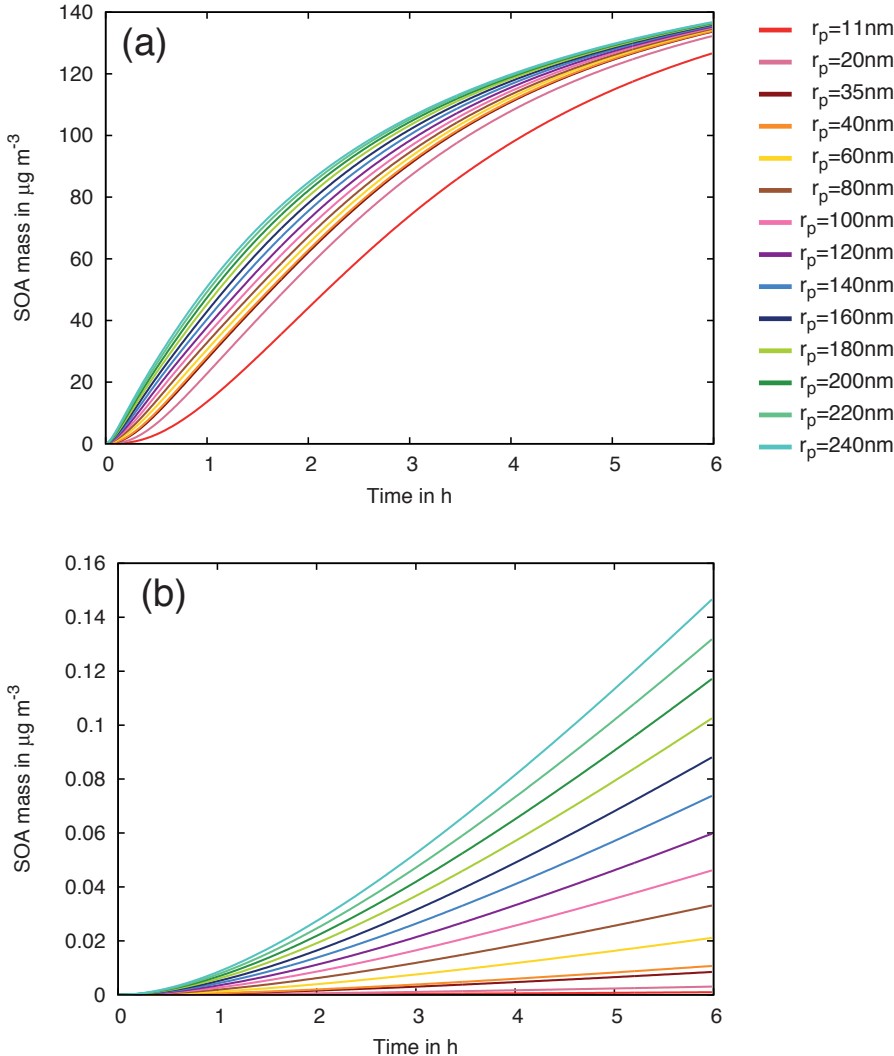

**Figure 3.** Model results of SOA mass for the variation of the particle radius $r_\mathrm{p}$ using a constant $k_\mathrm{c} = 1\,\mathrm{s}^{-1}$ (case 4a and 4b of Table 1) for a) liquid ($D_\mathrm{b} = 10^{-12}\,\mathrm{m}^2\,\mathrm{s}^{-1}$) and b) semi-solid ($D_\mathrm{b} = 10^{-18}\,\mathrm{m}^2\,\mathrm{s}^{-1}$) aerosol particles.

during LEAK chamber experiments and ambient measurements at the TROPOS field site Melpitz. Laboratory measurements of Berndt et al. (2016) provide molar yields of HOMs for the $\alpha$-pinene ozonolysis and $\alpha$-pinene OH oxidation of 3.4 % and 2.4 %, respectively. The existing gas-phase chemistry mechanism of Chen and Griffin (2005) does not consider the formation of HOMs. Therefore, the mechanism of Chen and Griffin (2005) has been updated with the HOM formation from $\alpha$-pinene

5  implementing the yields measured by Berndt et al. (2016). The HOM compounds are lumped to one compound group and added to the gas-phase chemistry mechanism (see Table S1, reactions 1b and 3b). As an estimate for the vapor pressure of

this compound group, the average vapor pressure of the compounds listed in Table S4 from COSMO-RS have been taken. To investigate the influence of the HOMs on the SOA mass, we have varied the pseudo-first-order rate constant of particle reactions and the particle-phase bulk diffusion coefficient (see Table 1 model case 6a and 6b for details). Thus, the effect of HOMs is characterized under different particle phase conditions and reaction regimes, however, for HOMs no further reactions

in the particle phase have been considered in this study. Almost no initial organic mass $OM_0$ is utilized for the simulations with HOMs. Nevertheless, the conducted simulations account not for new particle formation. The simulations without initial organic mass is consistent to previous studies, initialized with inorganic seed aerosol particles. The simulation results of Sect. 3.1.2 have been chosen as a reference case to characterize the influence of HOMs on the formed SOA mass. In contrast the reference simulations have been conducted with an initial organic aerosol mass concentration of $OM_0 = 5.8 \times 10^{-2}\,\mathrm{g\,g^{-1}}$.

The particle radius is 35 nm for both simulation setups.

In Fig. 4a the formed SOA mass for liquid particles ($D_b = 10^{-12}\,\mathrm{m^2\,s^{-1}}$) is compared with simulations with and without consideration of HOMs under additional variation of the pseudo-first-order rate constant of particle reactions. For the liquid phase state ($D_b = 10^{-12}\,\mathrm{m^2\,s^{-1}}$) the formed SOA mass is always increased due to the consideration of HOMs. A rapid condensation of HOMs occurs due to their low vapor pressures (Fig. 4a and Fig. S6 in the Supplement). This circumstance leads to an ef-

fective SOA formation immediately with oxidation of $\alpha$-pinene. Consequently, condensed HOMs serve as an organic medium on the particles and support the subsequent condensation of SVOCs on the particle surfaces. When considering HOMs, the initialization with initial organic particle mass is not required because the formed SOA mass quickly exceeds $OM_0$ due to the immediate partitioning of the HOMs. Fig. 4b depicts the same sensitivity study for the phase transition zone between a liquid and semi-solid phase state ($D_b = 10^{-14}\,\mathrm{m^2\,s^{-1}}$). The main difference compared to the results for liquid particles is the

convergence of the simulated SOA mass for $k_c = 1\,\mathrm{s^{-1}}$ for both cases. However, for slower chemical reactions in the particle phase, the SOA mass is increased when considering HOMs. An explanation for this behavior might be the missing chemical reactions in the particle phase for HOMs. Very fast chemical particle phase reactions can increase the uptake flux of organic gaseous compounds and compensate longer diffusion times for viscous particles. This may be the reason for the convergence of formed SOA mass for $k_c = 1\,\mathrm{s^{-1}}$ combined with $D_b = 10^{-14}\,\mathrm{m^2\,s^{-1}}$. For semi-solid particles ($D_b = 10^{-18}\,\mathrm{m^2\,s^{-1}}$, Fig. 4c),

the SOA formation without consideration of HOMs is more effective. This circumstance is caused by the missing particle-phase reactions for HOMs because the utilized kinetic approach is for $D_b < 10^{-15}\,\mathrm{m^2\,s^{-1}}$ mainly sensitive to particle-phase reactions (Zaveri et al., 2014). Further, for semi-solid particles a particle-phase-diffusion-limited partitioning of HOMs occur with proceeding simulation time (Mai et al., 2015). Therefore, the equilibrium between the gas and the particle phase is quickly established for HOMs and their effect on the SOA mass differ from that for liquid particles. As indicated before, semi-solid

aerosol particles might occur for a limited range of atmospheric conditions when the relative humidity decreases below 40 %. Therefore, the HOMs might affect SOA formation as seen in Fig. 4a and 4b, regardless of their reactivity in the particle phase. Fig. 5 demonstrates how much SOA mass is contained in the different component groups throughout the simulation time. In general, the HOMs provide about 27 % of the simulated final total SOA mass and initiate SOA mass formation. For the chosen pseudo-first-order rate constant of particle reactions $k_c = 10^{-4}\,\mathrm{s^{-1}}$, the remaining SOA mass is subdivided equally between

the partitioning and the reacted products, whereby, for the liquid particles the mass contributed by the r-products increases

over time. Accordingly, HOMs are important for fresh SOA formation since they provide the organic medium for SVOCs to partition into the aerosol phase and secondly induce further organic particle growth.

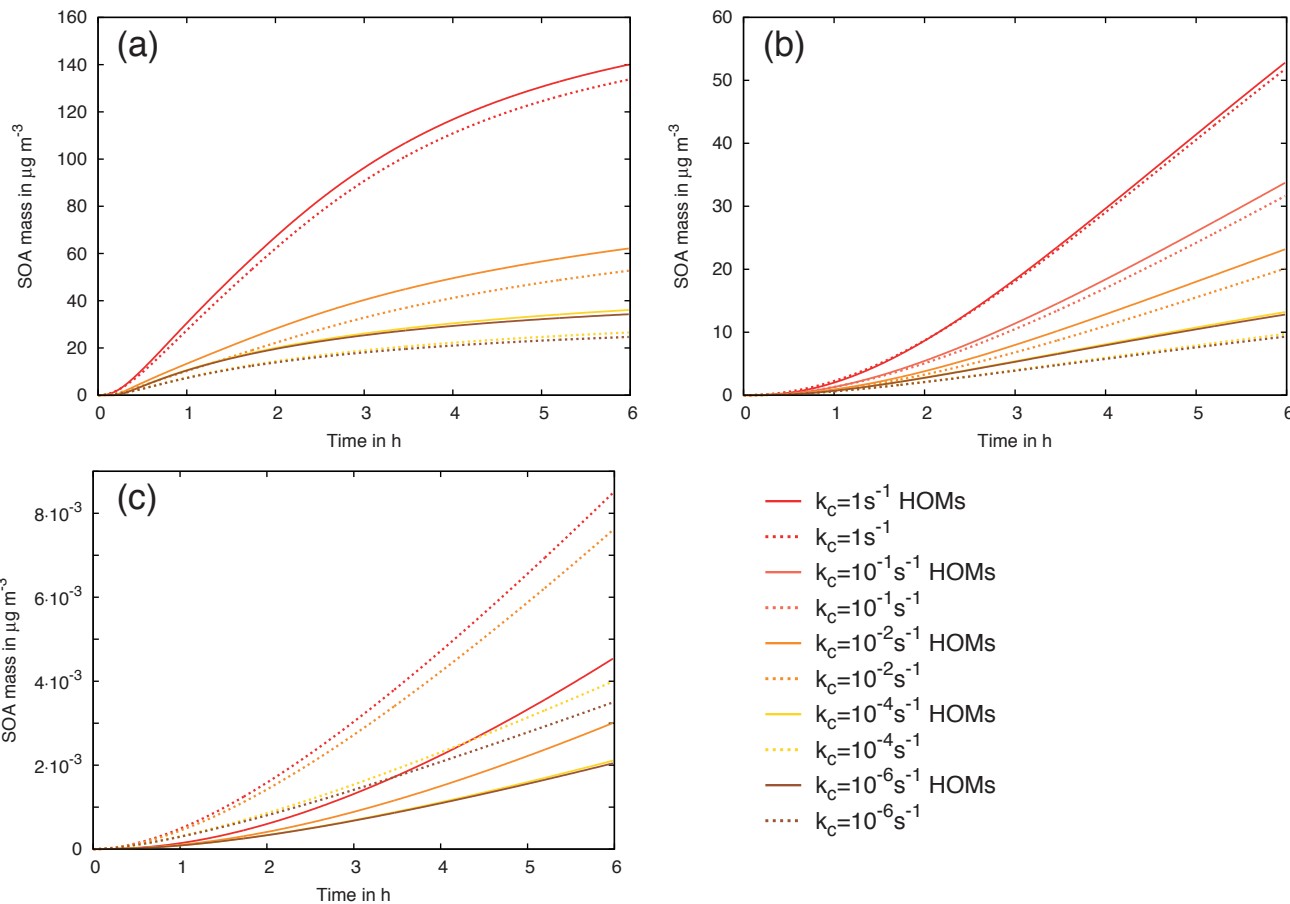

**Figure 4.** Simulated SOA mass including HOMs and additional variation of the pseudo-first-order rate constant of particle reactions $k_c$ (case 6a and 6b of Table 1) for a) liquid ($D_b = 10^{-12}\,\mathrm{m^2\,s^{-1}}$), b) transition between liquid and semi-solid ($D_b = 10^{-14}\,\mathrm{m^2\,s^{-1}}$) as well as c) semi-solid ($D_b = 10^{-18}\,\mathrm{m^2\,s^{-1}}$) aerosol particles.

### 3.3   Representation of reversible SOA formation pathways and the impacts on SOA

An additional backward reaction for every partitioned species has been implemented to further improve/refine the approach of Zaveri et al. (2014) and, thus, investigate the influence of particle phase chemical reactions in more detail. Backward reactions describe the reaction from the aged organic particle phase compounds to the original partitioned organic compounds that exchange directly with the gas phase. Organic aerosol-phase reactions can be irreversible reactions such as oxidation reactions or reversible reactions as for instance dimerization/oligomerization (Hallquist et al., 2009; Ziemann and Atkinson, 2012). For

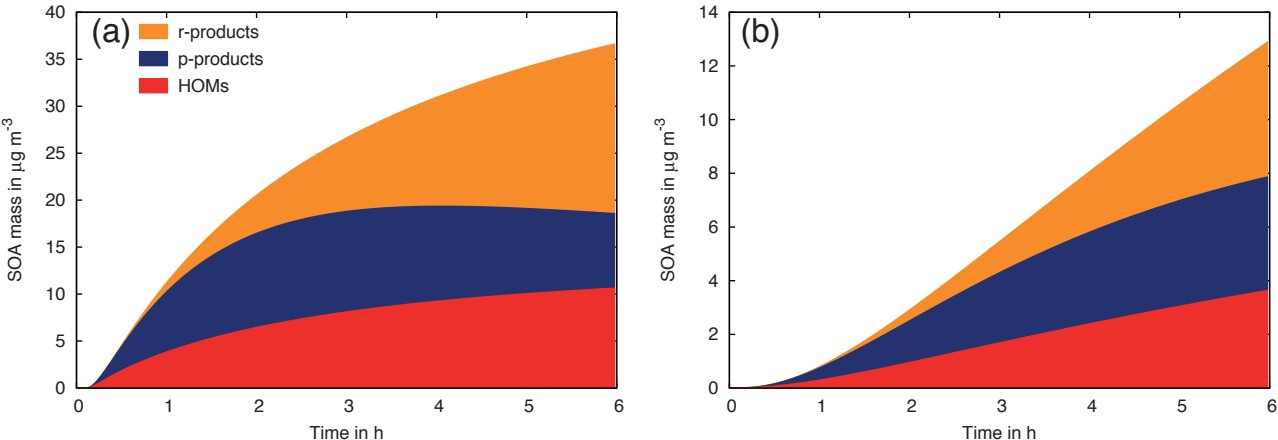

**Figure 5.** Simulated SOA mass including HOMs resolved for the different components for a) liquid ($D_b = 10^{-12}\,\mathrm{m^2\,s^{-1}}$) and b) the transition between liquid and semi-solid ($D_b = 10^{-14}\,\mathrm{m^2\,s^{-1}}$) aerosol particles and $k_c = 10^{-4}\,\mathrm{s^{-1}}$. Therein, p-products represent compounds which are partitioned from the gas phase in the particle phase and the r-phase includes compounds which are chemically processed by particle phase reactions.

the observed particle-phase dimerization, different possible reaction mechanisms can be found in the literature: (i) hemiacetal formation due to reactions between alcohols and aldehydes or carbonyl compounds (Iinuma et al., 2004; Ziemann and Atkinson, 2012; Carey and Sundberg, 2007), (ii) peroxyhemiacetal formation between hydroperoxides and carbonyl compounds (Tobias and Ziemann, 2000; Ziemann and Atkinson, 2012), (iii) aldol reaction products from the acid-catalyzed dimerization of a ketone

or aldehyde (Carey and Sundberg, 2007; Casale et al., 2007), and (iv) esterfication due to reactions of carboxylic acids with alcohols (Surratt et al., 2006; Ziemann and Atkinson, 2012; Carey and Sundberg, 2007). Thermodynamic calculations indicate ester formation and peroxyhemiacetal formation as most likely (Barsanti and Pankow, 2006; DePalma et al., 2013) and suggest hemiacetal formation as thermodynamically unfavorable (Barsanti and Pankow, 2004; DePalma et al., 2013). Therefore, an irreversible representation of the aerosol chemistry might lead to an overprediction of the formed SOA mass, which means that

can be only considered as an upper limit approach. The formed oligomers are complex compounds, which consist of a few monomer units. The oligomer equilibrium can be influenced by ambient conditions such as the temperature, relative humidity, and the chemical composition of the aerosol. A reversible representation of oligomerization reactions can be considered by means of an implemented backward reaction. E.g. Roldin et al. (2014) treats the kinetics of the reversible dimerization also with two separate reactions. However, an advanced kinetic treatment of particle-phase reactions is utilized considering monomer

concentrations combined with second-order rate constants and dimer first-order degradation rates separated for bulk and surface layers. However, measurement data concerning dimerization reaction rates are scarce for condensed organic compounds and vary over several orders of magnitude (Antonovskii and Terent'ev, 1967). For the sensitivity study concerning the influence of the backward reaction on the predicted SOA mass, a simplified approach is tested. Therefore, we considered different backward

reaction rate constants for particle-phase reactions (see Table 1, case 7) in addition to the pseudo-first-order rate constants. The implemented reactions are given in Table S3 in the Supplement. Fig. 6 shows the influence of the backward reactions in the liquid particle phase on the formation of SOA. For a backward reaction rate of $k_b=10^{-6}\,\mathrm{s}^{-1}$ the SOA formation is almost equal to the cases without a backward reaction. The formed SOA concentrations decrease for the cases with fast backward

reactions $k_b \geq 10^{-3}\,\mathrm{s}^{-1}$. This value is lower than for the reference case with a 10-fold lower chemical rate constant (see Fig. 6a and 6b). Fast particle-phase reactions (see Fig. 6a) combined with backward reactions $k_b \geq 10^{-2}\,\mathrm{s}^{-1}$ induce slower SOA formation and decrease SOA concentrations with respect to a second reference case ($k_c =10^{-1}\,\mathrm{s}^{-1}$). The SOA formation was faster than the base case with $k_c=10^{-1}\,\mathrm{s}^{-1}$ at the beginning, when $k_c=1\,\mathrm{s}^{-1}$ combined with $k_b=10^{-3}\,\mathrm{s}^{-1}$. However, for model runs considering backward reactions, the formed SOA is lower than for the reference case without backward reactions.

Fig. 6b reveals a similar SOA formation for a more moderate chemical rate constant in the particle phase $k_c=10^{-2}\,\mathrm{s}^{-1}$ in comparison to the previous results using fast reactions. In combination with the slowest backward reactions $k_b=10^{-6}\,\mathrm{s}^{-1}$, the formed SOA mass is almost equal to the reference case with $k_c=10^{-2}\,\mathrm{s}^{-1}$ and no backward reactions. Model simulations using backward reaction constants of $k_b \geq 10^{-3}\,\mathrm{s}^{-1}$ show lowered SOA production. The formed SOA mass is lower than the reference case using $k_c=10^{-3}\,\mathrm{s}^{-1}$. During the first hour of the simulation, the SOA formation evolves almost in the same way for all

different cases. Afterwards, the three cases with a slow backward reaction $k_b=10^{-6}-10^{-4}\,\mathrm{s}^{-1}$ follow the SOA formation of the reference case with $k_c=10^{-2}\,\mathrm{s}^{-1}$. The SOA formation for backward reactions with $k_b=10^{-2}\,\mathrm{s}^{-1}$ and $k_b=10^{-3}\,\mathrm{s}^{-1}$ behaves in the same way as described above for fast particle phase reactions (see Fig. 6b). The main benefit of the implementation of a sufficiently fast backward reaction ($k_b \leq 10^{-2}\,\mathrm{s}^{-1}$) is the asymptotic curve shape of the SOA mass for proceeding simulation times. This behavior is also observed during chamber studies (Ng et al., 2006), which indicate an equilibrium state of the gas and the particle phase after a proceeding oxidation time and concomitant consumption of the hydrocarbon.

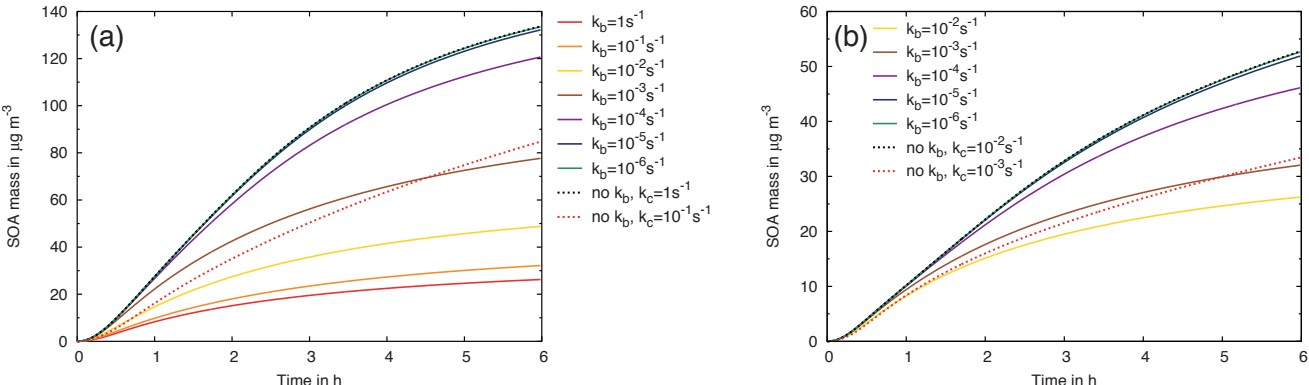

**Figure 6.** Simulated SOA mass including a chemical backward reaction in the particle phase for liquid ($D_b = 10^{-12}\,\mathrm{m^2\,s^{-1}}$) aerosol particles (case 7 of Table 1) with a) a fast chemical reaction in the particle phase $k_c=1\,\mathrm{s}^{-1}$ and b) a reduced rate constant $k_c=10^{-2}\,\mathrm{s}^{-1}$, whereby backward reactions with different rate constants $k_b$ are additionally included. The reference simulations for the respective $k_c$ without backward reactions (indicated by "no $k_b$" in the key) are shown with dashed lines.

### 3.4 Impact of a weighted particle-phase bulk diffusion coefficient considering particle composition on SOA formation

This subsection examines the particle-phase bulk diffusion coefficient extensively, namely the influence of the effective particle-phase bulk diffusion coefficient on SOA formation. Inorganic seed is usually utilized for chamber experiments. The seed is often composed of a dissolved salt (e.g., ammonium sulfate) and water. For water and ionic molecules in concentrated aqueous solutions, diffusion coefficients are well known. A temperature-dependent diffusion coefficient for water is provided by Holz et al. (2000). Diffusion coefficients for ions in aqueous solution are listed in Cussler (2009). However, diffusion coefficients for organic compounds are not known and depend on the other compounds that exist in the particle phase. Literature data are available for bulk viscosity values of SOA under different RH conditions (Renbaum-Wolff et al., 2013a; Zhang et al., 2015; Grayson et al., 2016). However, these measurements represent the phase state of processed SOA particles after several hours in the chamber and various treatment steps. Also the conversion of viscosity to diffusion coefficients is an uncertainty factor as outlined by Marshall et al. (2016). The main goal of this sensitivity study is to calculate a composition weighted particle-phase bulk diffusion coefficient $D_{\mathrm{m}}$ for the particles over the entire simulation time. The individual diffusion coefficients are weighted by applying a Vignes type rule (Vignes, 1966; Wesselingh and Bollen, 1997) concerning the mole fractions $x_i$:

$$D_{\mathrm{m}} = \left[D_{\mathrm{org}}^{x_{\mathrm{org}}}\right] \times \left[D_{\mathrm{inorg}}^{x_{\mathrm{inorg}}}\right] \times \left[D_{\mathrm{water}}^{x_{\mathrm{water}}}\right]. \tag{10}$$

Equation (10) describes the calculation of a mean or weighted particle-phase bulk diffusion coefficient $D_{\mathrm{m}}$ under consideration of the compound specific diffusion coefficients for water $D_{\mathrm{water}}$, dissolved ions $D_{\mathrm{inorg}}$ (here ammounium sulfate), and organic compounds $D_{\mathrm{org}}$ including their corresponding mole fractions $x_{\mathrm{water}}$, $x_{\mathrm{inorg}}$, and $x_{\mathrm{org}}$, respectively (more details are given in the Supplement). However, the diffusion coefficient of the pure organic compound mixture is not known from measurements or tabulated for the single compounds. Therefore, we utilize the results from Sect. 3.1.1 to estimate $D_{\mathrm{org}}$ in an appropriate way. We conducted two sensitivity studies, one with $D_{\mathrm{org}} = 10^{-12} \, \mathrm{m^2 \, s^{-1}}$ and the second with $D_{\mathrm{org}} = 10^{-14} \, \mathrm{m^2 \, s^{-1}}$ under additional variation of the pseudo-first-order rate constant of particle reactions $k_{\mathrm{c}}$ (see Table 1, case 8). The impact of increasing organic mole fractions of aerosols on the time evolution of SOA formation has been investigated with this approach. Thereby, the variation of the chemical rate constants for particle phase reactions induces different time evolutions of organic mass increase (see Sect. 3.1.2). In Fig. 7a the modeled SOA formation is shown for applying a weighted particle-phase bulk diffusion coefficient with $D_{\mathrm{org}} = 10^{-12} \, \mathrm{m^2 \, s^{-1}}$ and for comparison with a constant particle-phase bulk diffusion coefficient of $D_{\mathrm{b}} = 10^{-12} \, \mathrm{m^2 \, s^{-1}}$, respectively. The main difference of this study is the slightly faster increase of SOA mass at the beginning of the simulation for the weighted particle-phase bulk diffusion coefficient. The reason for this behavior is the higher initial particle-phase bulk diffusion coefficient in the order of $D_{\mathrm{m}} = 10^{-12} \, \mathrm{m^2 \, s^{-1}}$ of the seed composed of ammonium sulfate and water. The simulation results under consideration of $D_{\mathrm{org}} = 10^{-14} \, \mathrm{m^2 \, s^{-1}}$ are compared to the results using a constant particle-phase bulk diffusion coefficient of $D_{\mathrm{b}} = 10^{-12} \, \mathrm{m^2 \, s^{-1}}$ (Fig. 7b) and $D_{\mathrm{b}} = 10^{-14} \, \mathrm{m^2 \, s^{-1}}$ (Fig. 7c). A faster uptake of organic mass occurs in the first minutes for a weighted particle-phase bulk diffusion coefficient (Fig. 7b). With an increasing organic mass the weighted particle-phase bulk diffusion coefficient drops below a value of $D_{\mathrm{b}} = 10^{-12} \, \mathrm{m^2 \, s^{-1}}$ with the consequence of slower and less effective SOA formation (about 37 % decrease for $k_{\mathrm{c}}$=1 s$^{-1}$). Fig. 7 reveals that the total SOA mass is increased by about $40-50$ % for a weighted particle-phase bulk diffusion coefficient ($D_{\mathrm{org}} = 10^{-14} \, \mathrm{m^2 \, s^{-1}}$) compared to simulation results

with a constant particle-phase bulk diffusion coefficient of $D_b = 10^{-14}\,\mathrm{m^2\,s^{-1}}$. Additionally, the SOA formation is faster for the weighted particle-phase bulk diffusion coefficient.

Overall, the simulations have shown that the obtained model results are sensitive to a composition-dependent particle-phase bulk diffusion coefficient, which is obvious when compared with results for constant particle-phase bulk diffusion coefficient.

The assumption of a slower diffusion coefficient for the organic material might be justified due to the high bulk viscosity from measurements (Renbaum-Wolff et al., 2013a; Zhang et al., 2015; Grayson et al., 2016). The application of a modified Vignes type rule for the calculation of a weighted particle-phase bulk diffusion coefficient is already mentioned and applied by Lienhard et al. (2014, 2015) and Price et al. (2015) for the water diffusion coefficient in SOA particles. The applicability of Eq. (10) within the kinetic approach of Zaveri et al. (2014) is checked and verified under the utilization of the model by Zobrist

et al. (2011) as basis for evaluation (S. O'Meara, personal communication). Fig. S7a and S7b in the Supplement display the differences for the numerical solution from the model of Zobrist et al. (2011) and the analytical solution of the kinetic approach with the weighted particle-phase bulk diffusion coefficient for $D_{org} = 10^{-11}\,\mathrm{m^2\,s^{-1}}$ and $D_{org} = 10^{-13}\,\mathrm{m^2\,s^{-1}}$, respectively. For both assumed self-diffusion coefficients of the organic fraction, the numerical and analytical solution are equal within $1 \times 10^{-6}\,\mathrm{s}$ and $1 \times 10^{-4}\,\mathrm{s}$. Thus, Eq. (10) can applied to the kinetic approach instead of a constant bulk diffusion coefficient.

However, improved particle-phase bulk diffusion coefficient data depending on relative humidity and organic mass loading will be needed to improve current model implementations.

### 3.5 Comparison with performed LEAK chamber measurements

Selected simulation results have been compared with measurements from the aerosol chamber LEAK for $\alpha$-pinene ozonolysis investigations. The simulations show only results for the narrow range of the investigated parameter setup. Figure 8a and

8b present simulation results for the combination of the three newly implemented model features from Sects. 3.2 to 3.4 in comparison with the results from the base kinetic approach and measurements. Wall loss effects are not considered in this study, due to the short experiment time (2 h). However, for longer experiment duration particle and gas wall loss might be an important process for chamber studies and have to be considered in modeling (Zhang et al., 2014). The simulations shown in Fig. 8a are conducted with an effective particle-phase bulk diffusion coefficient considering an organic diffusion coefficient of

$D_{org} = 10^{-12}\,\mathrm{m^2\,s^{-1}}$. The pseudo-first-order rate constant of particle reactions is set to $k_c = 10^{-2}\,\mathrm{s^{-1}}$ and only the backward reaction constants have been varied. Simulations with a particle-phase bulk diffusion coefficient of $D_b = 10^{-12}\,\mathrm{m^2\,s^{-1}}$ and $k_c = 10^{-2}\,\mathrm{s^{-1}}$ (ref. case I) as well as $k_c = 10^{-3}\,\mathrm{s^{-1}}$ (ref. case II) are selected as reference cases. In general, the reference simulation with $k_c = 10^{-3}\,\mathrm{s^{-1}}$ is in good agreement with the LEAK measurements (Fig. 8a, ref. case II). However, the simulated SOA mass is systematically underestimated for the first 1.5 h. This might be caused by not considered HOM yields and a

too low particle-phase bulk diffusion coefficient for the early stage of SOA formation. For $k_c = 10^{-2}\,\mathrm{s^{-1}}$ this initial underestimation is marginal (Fig. 8a, ref. case I). On the other hand, the overestimation of SOA becomes obvious for $k_c = 10^{-2}\,\mathrm{s^{-1}}$ after 1 h simulation time and characterizes the simulated SOA mass till the end of the experiment time. Therefore, the formed SOA mass for the base parametrization appears to lead to initial underestimation or final overestimation. Consideration of the weighted particle-phase bulk diffusion coefficient and HOMs lead to a faster SOA mass increase at the beginning of the

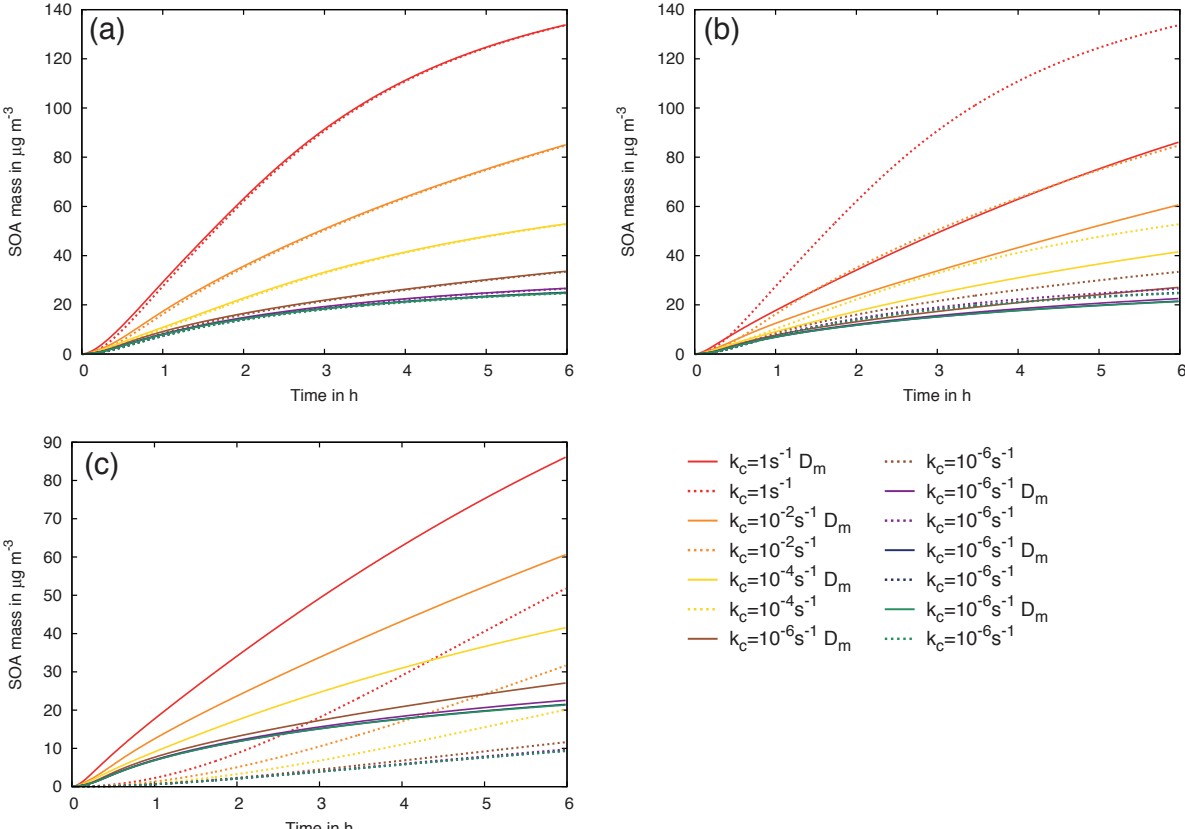

**Figure 7.** Simulated SOA mass considering an effective particle-phase bulk diffusion coefficient under additional variation of the pseudo-first-order rate constant of particle reactions $k_c$ (case 8 of Table 1). a) Organic material is considered with a diffusion coefficient of $D_{org} = 10^{-12}\,\mathrm{m^2\,s^{-1}}$ in the weighted particle-phase bulk diffusion coefficient $D_m$ (solid lines) and for comparison the results for $D_b = 10^{-12}\,\mathrm{m^2\,s^{-1}}$ are shown (dashed lines); b) Organic material is considered with a diffusion coefficient of $D_{org} = 10^{-14}\,\mathrm{m^2\,s^{-1}}$ in the weighted particle-phase bulk diffusion coefficient $D_m$ (solid lines) and for comparison the results for $D_b = 10^{-12}\,\mathrm{m^2\,s^{-1}}$ are shown (dashed lines); c) Organic material is considered with a diffusion coefficient of $D_{org} = 10^{-14}\,\mathrm{m^2\,s^{-1}}$ in the weighted particle-phase bulk diffusion coefficient $D_m$ (solid lines) and for comparison the results for $D_b = 10^{-14}\,\mathrm{m^2\,s^{-1}}$ are shown (dashed lines).

simulation when the organic amount is low in the particle phase. The decreasing particle-phase bulk diffusion coefficient due to the uptake of further organic material and the backward reactions in the particle phase induce a flattening of the mass increase. Nevertheless, the SOA mass is highly overestimated, which might be caused by the high particle-phase bulk diffusion coefficient of the organic material. The simulations with an effective particle-phase bulk diffusion coefficient (Fig. 8a) are re-produced with a smaller diffusion coefficient for the organic material $D_{org} = 10^{-14}\,\mathrm{m^2\,s^{-1}}$. The results for these simulations are presented in Fig. 8b where the same reference simulations as shown in Fig. 8a are utilized. Figure 8b reveals that all five simulations with the weighted particle-phase bulk diffusion coefficient start nearly at the same time as observed in the exper-

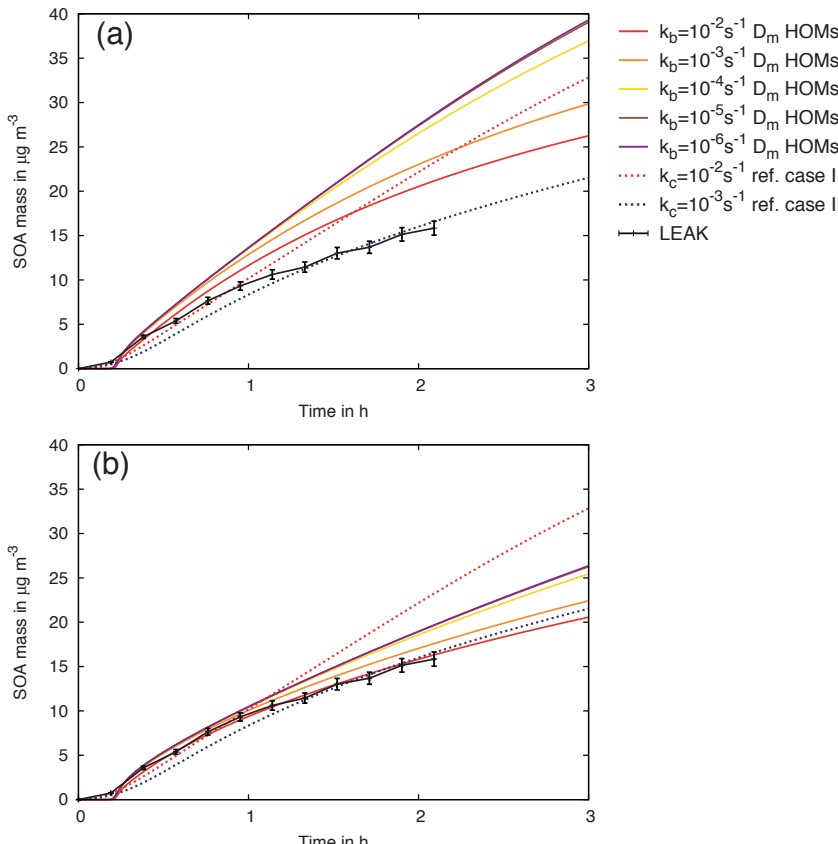

**Figure 8.** Simulated SOA mass considering an effective particle-phase bulk diffusion coefficient as well as HOMs, a constant pseudo-first-order rate constant of particle reactions $k_c = 10^{-2}\,\text{s}^{-1}$ and under additional variation of the chemical backward reaction rate constant of particle reactions $k_b$ in comparison with aerosol chamber measurements from LEAK. a) Organic material is considered with a diffusion coefficient of $D_{org} = 10^{-12}\,\text{m}^2\,\text{s}^{-1}$ in the weighted diffusivity $D_m$ (solid lines); b) Organic material is considered with a diffusion coefficient of $D_{org} = 10^{-14}\,\text{m}^2\,\text{s}^{-1}$ in the weighted diffusivity $D_m$ (solid lines). For comparison, in both plots the results for a constant particle-phase bulk diffusion coefficient $D_b = 10^{-12}\,\text{m}^2\,\text{s}^{-1}$ combined with two different pseudo-first-order rate constants of particle reactions $k_c = 10^{-2}\,\text{s}^{-1}$ (ref. case I) and $k_c = 10^{-3}\,\text{s}^{-1}$ (ref. case II) are included (dashed lines).

iment with the formation of SOA. After 1 h simulation time, it is obvious that the simulated concentration profile agrees well with the experimentally observed SOA mass when using a backward reaction rate constant of $k_b=10^{-2}\,\text{s}^{-1}$. The high initial particle-phase bulk diffusion coefficient, $D_m \approx 2 \times 10^{-9}\,\text{m}^2\,\text{s}^{-1}$ (see Fig. 9), for the aqueous ammonium sulfate seed particles enables a fast diffusion in the aerosol particles. Thus, immediately partitioning HOMs can be absorbed quickly into the particle phase. Within the first 30 min of the simulation time, the SOA mass sharply increases and the weighted particle-phase bulk diffusion coefficient drops about three orders of magnitude. Consequently, the mixing time in the particle phase increases and this leads to a slower SOA mass formation. This process is depicted in Fig. 9, where for the first hour of simulation time the major

changes in the weighted particle-phase bulk diffusion coefficient and the SOA mass can be seen. After the weighted particle-phase bulk diffusion coefficient has reduced to a value $D_\mathrm{m} \approx 10^{-13}\,\mathrm{m^2\,s^{-1}}$, the longer mixing time will cause a slower SOA formation as already shown in Fig. 1b. This effect is further pronounced due to continuous SOA formation and a concomitant decrease in particle-phase diffusion. By means of the implementation of HOMs, no initial organic particle mass is necessary

to enable partitioning after a short oxidation time ($\approx 8\,\mathrm{min}$). The effective particle-phase bulk diffusion coefficient is reduced and the mixing time increases by increasing the organic mass over the time, slowing down SOA mass formation (see Fig. 9). Furthermore, the chemical backward reactions in the particle phase induce an equilibrium state, e.g. for the oligomer formation. Accordingly, an equilibrium is also achieved between the gas and the p-products in the particle phase for the semi-volatile compounds. Additionally, to the good agreement of simulated total SOA mass, the flattening with increasing organic mass

better represents the time profile of SOA. Here, HOMs provide about 27 % of the simulated total SOA mass at the end of the simulation time (see Fig. 10). This points out the important role of HOMs for initial SOA formation. Additionally, the chemical analysis of the filter measurements from LEAK revealed that organic peroxides contribute to the formed organic mass, which agrees with the simulated partitioning of HOMs into the particle phase. Moreover, Fig. 10b shows that the HOMs mainly parti-

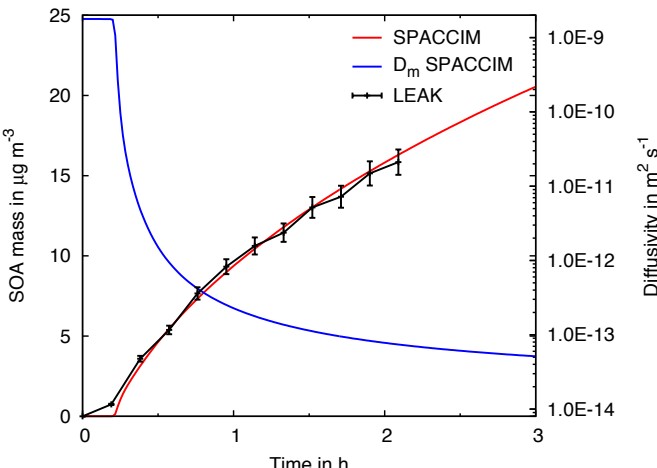

**Figure 9.** Simulated SOA mass as shown in Fig. 8 for $k_\mathrm{c} = 10^{-2}\,\mathrm{s^{-1}}$, $k_\mathrm{b} = 10^{-2}\,\mathrm{s^{-1}}$, under consideration of HOMs, and with the weighted diffusion coefficient utilizing $D_\mathrm{org} = 10^{-14}\,\mathrm{m^2\,s^{-1}}$ in comparison to the measured SOA mass from the LEAK experiment. The corresponding weighted particle-phase bulk diffusion coefficient $D_\mathrm{m}$ from the simulation is displayed on the secondary y-axis.

tion in the first minutes of the simulation into the particle phase. For the simulation of SOA formation shown in Fig. 8 and Fig.

10, the vapor pressure estimates of the HOM compounds have been taken as given by Berndt et al. (2016) for the calculation with COSMO-RS (Eckert and Klamt, 2002). COSMO-RS is based on quantum chemical methods and the calculation of the molecular surface (Eckert and Klamt, 2002), which enables a more accurate estimation of thermodynamic properties and might be more precise than group contribution methods (Kurtén et al., 2016). For comparison, the vapor pressures of the HOMs have been estimated with the group contribution methods SIMPOL (Pankow and Asher, 2008) and EVAPORATION (Compernolle

et al. (2011), see Table S4 in the Supplement) and accordingly, the SOA formation have been simulated for every method. The total SOA mass deviates maximally by 11 % from the SOA mass formed with COSMO-RS vapor pressure estimates (see Fig. S8a in the Supplement). However, the temporal curve shape of SOA formation and the relative contribution of the three product classes to the total organic mass deviates between model simulations utilizing different HOM vapor pressures. Thus, for the vapor pressures estimated by EVAPORATION, the HOMs contribute in the first 15 minutes of SOA formation between 97 and 100 % to the organic mass (see Fig. S8d in the Supplement). This is a higher contribution as for the other model simulations (see Fig. S8b and S8c in the Supplement) and the time period for this high contribution is longer. Due to the fact that the vapor pressures of the HOMs from $\alpha$-pinene ozonolysis might be lower than the utilized values, the relative contribution of HOMs to the initial SOA formation could be higher than indicated by the simulation with COSMO-RS vapor pressures.

Gas-phase concentrations of the reactants $\alpha$-pinene and ozone (see Fig. 11a and 11b) as well as a first reaction product named

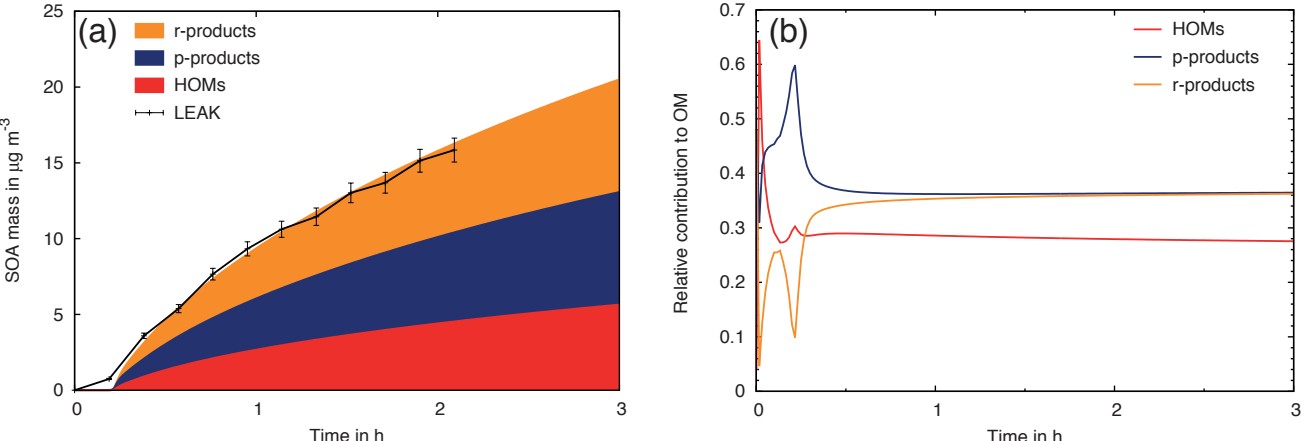

**Figure 10.** a) Simulated SOA mass as shown in Fig. 8b for $k_b = 10^{-2}\,\mathrm{s}^{-1}$ resolved for the different components and compared with aerosol chamber measurements from LEAK. b) Relative contribution of the individual product classes to the total organic mass.

pinonaldehyde (Fig. 11c) have also been compared with smog chamber measurements. Corresponding gas-phase concentrations for the preferred model setup of Fig. 8b, with $k_b = 10^{-2}\,\mathrm{s}^{-1}$, of the reactants $\alpha$-pinene and ozone (see Fig. 8a and 8b) as well as a first reaction product named pinonaldehyde (Fig. 8c) have also been compared with smog chamber measurements. The simulation for the $\alpha$-pinene depletion is in a very good agreement with the measured concentration decrease of $\alpha$-pinene (Fig. 11a). The depletion of ozone is slightly overestimated by the model (see Fig. 11b). After half an hour, the measured ozone concentration decreases not so fast as initially started. Experimentally, ozone is measured with an ozone monitor (49c Ozone Analyzer, Thermo Scientific, USA) and this device is based on measuring absorption on characteristic wavelengths. For measuring ozone, the absorption at 254 nm is utilized. According to Docherty et al. (2005), the ozonolysis of $\alpha$-pinene yields up to 47 % organic peroxides. As organic peroxides absorb light at 254 nm, an overestimation of the signal detected by the ozone monitor caused by the high amount of organic peroxides cannot be excluded. Therefore, with the increase of the hydroperoxide concentration over the experiment time the overestimation of the ozone concentration by the monitoring

system might increase and the underestimation by the model can be caused. Further, the measured gas-phase concentration of pinonaldehyde is underestimated by the model (Fig. 11c). This cannot be caused by an excessive partitioning of pinonaldehyde into the particle phase because pinonaldehyde is characterized by a high saturation vapor pressure and there is no effective partitioning into the particle phase. However, the formation of pinonaldehyde is measured by a proton-transfer-reaction mass

spectrometer (PTR-MS) at *m/z* 169 ($[M+H]^+$). The PTR-MS technique enables only the detection of the *m/z* ratio. No further information were obtained. Thus, compounds or fragments with the same *m/z* were detected as well resulting in an overestimation of the pinonaldehyde concentration measured by PTR-MS. This circumstance can cause the underestimation of the gas-phase concentration of pinonaldehyde (see Fig. 11c). To investigate the underestimation of pinonaldehyde concentration additionally from the site of the model, we evaluated the branching ratios of the $\alpha$-pinene with ozone reaction. Based on the

results of Berndt et al. (2003), the pinonaldehyde yield was artificially increased in the mechanism to investigate the sensitivity of the pinonaldehyde on this yield. The results of this modification in the gas-phase chemistry mechanism are shown within Figs. 11a to 11d. For the reactants, no difference between the two simulations is observed (Fig. 11a and 11b). However, the pinonaldehyde concentrations fit very well with the measured values from the smog chamber (Fig. 11c). Due to the high vapor pressure of pinonaldehyde, the SOA mass decreases by about 20 % due to the increased pinonaldehyde yield in the modified

gas-phase chemistry mechanism (see Fig. 11d). Nevertheless, the results of the modified kinetic partitioning approach fit better with the measurements than the reference simulations.

### 3.6 Limitations of the present studies

The presented model studies do not account for the Kelvin effect. The Kelvin effect describes the change of the vapor pressure due to a curved liquid-vapor interface and is especially important for small particles because of their higher curvature (Seinfeld

and Pandis, 2006; Pruppacher and Klett, 2010). The vapor pressure of a compound $i$ over a flat surface $p_{\text{sat},i}$ (atm) can be corrected to the partial vapor pressure over a curved interface $p_i^{\ominus}$ (atm, Seinfeld and Pandis, 2006). The correction factor depends strongly on the particle size and the surface tension of the considered aerosol particle/droplet. The surface tension varies with the composition of the aerosol particle (Facchini et al., 1999; Hitzenberger et al., 2002; Ervens et al., 2004, 2005), e.g. it is increased by dissolved salts (Seinfeld and Pandis, 2006) and decreased by organic compounds (Facchini et al., 1999;

Ervens et al., 2005). However, for the estimation of the vapor pressures for the partitioning compounds a group contribution method (EVAPORATION, Compernolle et al., 2011) is applied in this study. An investigation of O'Meara et al. (2014) reveals that the vapor pressure estimates from the different group contribution methods vary from each other and deviate from existing measurements up to six orders of magnitude. Further, Kurtén et al. (2016) showed the differences between the vapor pressures estimated by three different group contribution methods and COSMO-RS (conductor-like screening model for real solvents,

Eckert and Klamt, 2002). Therein, 8 orders of magnitude lower vapor pressures are estimated by group contribution methods than COSMO-RS for some highly oxidized monomers. Therefore, the correction of the vapor pressure by the Kelvin effect might be in the order of the error range of the applied group contribution method. For this reason, we have not considered the Kelvin effect in our calculations.

This study utilizes a simplified scheme to consider particle-phase reactions in order to account for SOA aging. The kinetic

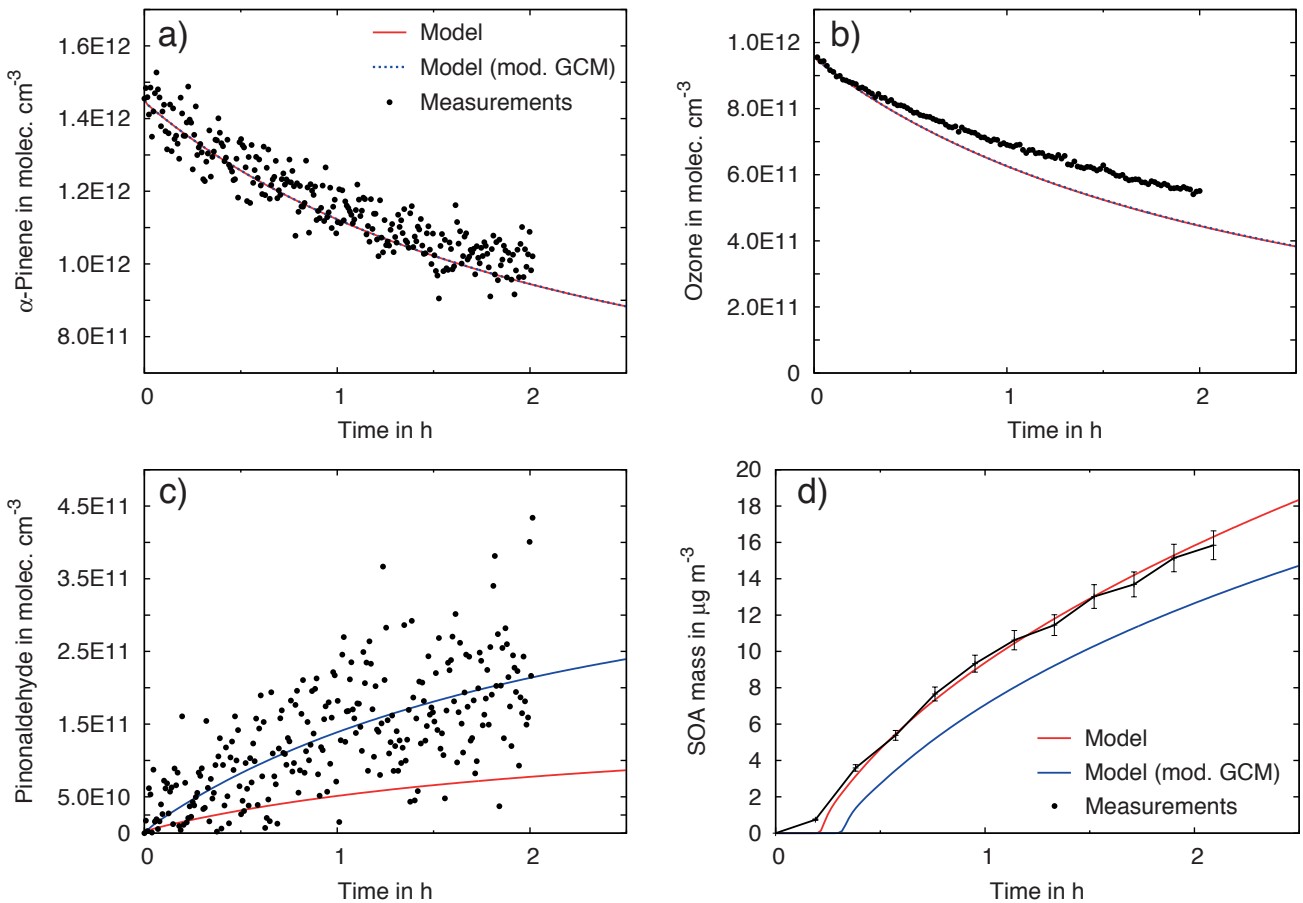

**Figure 11.** Simulation results for the preferred case of Fig. 8b with $k_b = 10^{-2}\,\mathrm{s^{-1}}$ (solid red line) and an additional model run with modified gas-phase chemistry mechanism (GCM, pinonaldehyde yield as provided by Berndt et al. (2003) dashed/ solid blue line) in comparison with measurements from LEAK (black dots). Gas-phase concentrations for the reactants a) $\alpha$-pinene and b) ozone are shown as well as a first-order product c) pinonaldehyde and d) the SOA mass.

approach of Zaveri et al. (2014) is divided into two reaction cases according to the rate of the particle-phase reactivity based on the achievement of the steady state. In Sect. 3.3, a modification of the particle-phase reactivity is presented in order to improve the representation of SOA aging under preservation of the basic classification/separation in fast and slow particle-phase reactions. This simplified approach is appropriate for application in 3-D models, treating organic compounds in lumped groups, and saves computational effort. However, for future chamber simulations with the focus on SOA processes combined with advanced measurement data, accounting for SOA aging or oxidation state, an improved representation of particle-phase reactivity will be implemented to further develop SPACCIM. Therefore, the pseudo-first-order rate constants will be replaced, e.g. by second-order equilibrium reactions under consideration of equilibrium rates provided by Barsanti and Pankow (2004)

for hydrate and hemiacetal formation or thermodynamic calculations for equilibrium constants of DePalma et al. (2013) for individual dimers.

## 4 Conclusions

The kinetic partitioning approach by Zaveri et al. (2014) has been implemented in the SPACCIM model in this study. Extensive sensitivity studies were performed to investigate the dependence of the SOA formation on i) the particle-phase bulk diffusion coefficient $D_b$, ii) the chemical reaction rate constant of the solute within the particle phase $k_c$, iii) particle radius $r_p$, and iv) the initial organic particle phase mass $OM_O$. The influence of HOMs on SOA formation was additionally investigated. Moreover, the kinetic approach was extended by both a chemical backward reaction of the solute within the particle phase $k_b$ and a composition-dependent particle-phase bulk diffusion coefficient $D_m$.

Overall, the conducted sensitivity studies reveal that the particle-phase bulk diffusion coefficient is the key parameter for the simulation of SOA formation and processes. In liquid particles ($D_b = 10^{-9} - 10^{-13}\,\mathrm{m^2\,s^{-1}}$) 310-times or 66-times more SOA is formed than in higher viscous aerosol particles ($D_b = 10^{-17} - 10^{-21}\,\mathrm{m^2\,s^{-1}}$), using a high or a negligible particle-phase reactivity, respectively. For a wide range of particle-phase bulk diffusion coefficients ($D_b = 10^{-9} - 10^{-14}\,\mathrm{m^2\,s^{-1}}$) almost the same SOA mass can be produced if long equilibration times are considered. However, on the time scale of chamber experiments, the observed equilibration time for SOA formation is shorter than that observed from the simulation with a constant particle-phase bulk diffusion coefficient of $D_b = 10^{-14}\,\mathrm{m^2\,s^{-1}}$. Nevertheless, for a high particle-phase bulk diffusion coefficient ($D_b > 10^{-14}\,\mathrm{m^2\,s^{-1}}$), the SOA mass is overestimated by about 40 % if the initial increase of organic mass is in good agreement due to a fast and irreversible particle phase reaction. The pseudo-first-order rate constant for particle reactions $k_c$ is shown to be a second key parameter for the description of organic mass in the particle phase to reflect oligomerization and aging. A large fraction of the formed SOA (61 % for model case 6) consists of chemically processed organic compounds, for liquid particles and a moderate rate of chemical particle reactions ($k_c = 10^{-4}\,\mathrm{s^{-1}}$). Up to now, the rate constants for such processes are not extensively evaluated, which introduces a large uncertainty to the applicability of this parameter in the model. Therefore, further kinetic and mechanistic studies are needed to better characterize the aerosol particle reactivity and the resulting contribution to the SOA processing. Additionally, the results of the sensitivity studies have revealed that the SOA mass continues to be formed if there are chemical backward reactions in the particle phase are neglected or the particle-phase bulk diffusion coefficient is not reduced due to the increase in organic material. However, the simulated temporal curve shape of constant SOA formation is in contrast to the result of the experiment. The performed studies with the advanced model show a benefit for SOA modeling particularly for the predicted SOA concentration-time profile and the overall SOA mass. The overprediction of the SOA mass has been reduced by about 40 % and the simulated temporal curve shape of SOA formation shows a much better agreement with measured SOA yields from the LEAK chamber. Besides the development of the partitioning approach, the extension of the gas-phase chemistry mechanism for $\alpha$-pinene considering HOMs has been shown to be a key factor for modeling SOA formation, particularly at the early stage of the chamber experiments. HOMs play a major role for initial SOA formation from $\alpha$-pinene because up to 65 % of OM is provided by them at the early stage of the simulation

and about 27 % of the SOA mass is formed by HOMs at the end of the simulation. Additionally, due to the consideration of the low-volatile HOMs, no need for initial organic particle mass exists and a better agreement with the observed SOA time profiles can be achieved. In conclusion, this study has (i) demonstrated the applicability of the kinetic approach (Zaveri et al., 2014) in the SPACCIM model, (ii) revealed the main key factors controlling the SOA formation, (iii) pointed out the current uncertainties/limitations of the approach, and (iv) showed the needs for further laboratory measurements as well as advanced model comparisons with chamber experimental data.

## Appendix A: Description of the kinetic partitioning approach according to Zaveri et al. (2014)

The existing model framework has been extended by the implementation of the kinetic partitioning approach established by Zaveri et al. (2014). The basic assumption of this approach is the description of the diffusive flux of a solute in the particle phase via Fick's second law extended by a particle phase reaction of the solute:

$$\frac{\partial A_i(r,t)}{\partial t} = D_{b,i} \frac{1}{r^2} \frac{\partial}{\partial r} \left( r^2 \frac{\partial A_i(r,t)}{\partial r} \right) - k_{c,i} A_i(r,t). \tag{A1}$$

Thereby, the utilized parameters are the particle-phase concentration $A_i$ of the solute $i$ as a function of the radius $r$ and the time $t$, the particle-phase bulk diffusion coefficient of the solute $D_{b,i}$, and the chemical reaction rate constant $k_{c,i}$ of the solute within the particle phase. Equation ( A1) is in spherical coordinates and there is a fundamental simplification concerning the diffusion coefficient. Here, the diffusion coefficient is assumed to be constant and, therefore, a bulk diffusion coefficient $D_{b,i}$ for the particle phase is introduced. This assumption simplifies the calculation of the integral (Eq. A1). For the solution of the transient partial differential equation (Eq. A1) the particle is assumed to be spherically symmetrical concerning the concentration profiles of the solute inside the considered particle. Therefore, the following initial and boundary conditions are defined by Zaveri et al. (2014):

Initial condition: $A_i(r,0) = 0$, \hfill (A2a)

Boundary condition 1: $A_i(r_p,t) = A_i^s$, \hfill (A2b)

Boundary condition 2: $\dfrac{\partial A_i(0,t)}{\partial r} = 0$. \hfill (A2c)

First, Eq. (A1) is analytically solved by means of (Eq. A2a to A2c) without consideration of the chemical reaction (Carslaw and Jaeger, 1959; Crank, 1975). The solution for taking account of a first-order chemical reaction in the particle phase is provided by Danckwarts (1951) and the integration of this solution in order to quantify the average particle-phase concentration $\overline{A}(t)$ yields:

$$\frac{\overline{A_i}(t)}{A_i^s} = \frac{\int_0^{r_p} 4\pi r^2 \frac{A_i(r,t)}{A_i^s} \mathrm{d}r}{\frac{4}{3}\pi r_p^3} = Q_i - U_i(t), \tag{A3}$$

where

$$Q_i = 3\left(\frac{q_i \coth q_i - 1}{q_i^2}\right), \tag{A4}$$

$$U_i(t) = \frac{6}{\pi^2} \sum_{n=1}^{\infty} \frac{exp\left\{-\left(k_{\mathrm{c},i} + \frac{n^2 \pi^2 D_{\mathrm{b},i}}{r_{\mathrm{p}}^2}\right) t\right\}}{(q_i/\pi)^2 + n^2}. \tag{A5}$$

Thereby, $r_{\mathrm{p}}$ is the particle radius and $q_i = r_{\mathrm{p}} \sqrt{k_{\mathrm{c},i}/D_{\mathrm{b},i}}$ is a dimensionless diffusion–reaction parameter. Zaveri et al. (2014)

describe $Q_i$ as steady state term and $U_i(t)$ as transient term, which equals $Q_i$ at $t = 0$ and decreases exponentially to zero when $t \to \infty$. Whereby, $t$ denotes here the time since the start, which can only be monitored in a Lagrangian box model for a "closed system". Zaveri et al. (2014) review that the gas-phase concentration profile of the solute around the particle is at a quasi-steady state. To describe the gas-particle partitioning, Zaveri et al. (2014) propose an ordinary differential equation:

$$\frac{\mathrm{d}\overline{A}_i}{\mathrm{d}t} = \frac{3}{r_{\mathrm{p}}} k_{\mathrm{g},i} \left(\overline{C}_{\mathrm{g},i} - C_{\mathrm{g},i}^{\mathrm{s}}\right) - k_{\mathrm{c},i} \overline{A}_i, \tag{A6}$$

whereby, $\overline{C}_{\mathrm{g},i}$ is the average bulk gas-phase concentration, $C_{\mathrm{g},i}^{\mathrm{s}}$ is the gas-phase concentration of the solute just outside the surface of the aerosol particle, and $k_{\mathrm{g},i}$ denotes the gas-side mass-transfer coefficient. The gas-side mass transfer coefficient depends on the gas diffusion coefficient, the particle radius, and the transition regime correction factor (Fuchs and Sutugin, 1971), which is a function of the Knudsen number and the mass accommodation coefficient ($0 \leq \alpha_i \leq 1$). Under atmospheric conditions interfacial phase equilibrium is achieved in a fractional amount of a second, which is meaningful for the description

of gas-to-particle mass transfer. The concentrations of an individual compound in the gas and the aerosol phase are linked by means of the effective saturation vapor concentration $C_{\mathrm{g},i}^*$ and the total organic aerosol mass at the surface $\sum_j A_j^{\mathrm{s}}$:

$$C_{\mathrm{g},i}^{\mathrm{s}} = \frac{A_i^{\mathrm{s}}}{\sum_j A_j^{\mathrm{s}}} C_{\mathrm{g},i}^*. \tag{A7}$$

Under consideration of Eq. (A3) and (A7) with the approximation of $\sum_j A_j^{\mathrm{s}}$ by $\sum_j \overline{A}_j$, the mass transfer in Eq. (A6) yields to:

$$\frac{\mathrm{d}\overline{A}_i}{\mathrm{d}t} = \frac{3}{r_{\mathrm{p}}} k_{\mathrm{g},i} \left\{\overline{C}_{\mathrm{g},i} - \frac{\overline{A}_i}{\sum_j \overline{A}_j} \frac{C_{\mathrm{g},i}^*}{(Q_i - U_i(t))}\right\} - k_{\mathrm{c},i} \overline{A}_i. \tag{A8}$$

Due to the time-dependent transient term $U_i(t)$ the approach is limited for usage in box models for "closed systems". To use the approach also in 3-D models without restrictions, some modifications have been proposed by Zaveri et al. (2014). According to their general sensitivity study concerning the time scales for a quasi-steady state, they can resolve that a distinction between two different reaction regimes is meaningful in this context. For chemical reactions with $k_{\mathrm{c},i} \geq 0.01\,\mathrm{s}$, a quasi-steady state is

reached in less than 1 minute and this is fast enough for usage in atmospheric Eulerian 3-D models. Therefore, this first case is valid for fast reactions and the term $U_i(t)$ can be neglected. The ordinary differential equation is rewritten for fast reactions to:

$$\frac{\mathrm{d}\overline{A}_i}{\mathrm{d}t} = \frac{3}{r_{\mathrm{p}}} k_{\mathrm{g},i} \left\{\overline{C}_{\mathrm{g},i} - \frac{\overline{A}_i}{\sum_j \overline{A}_j} \frac{C_{\mathrm{g},i}^*}{Q_i}\right\} - k_{\mathrm{c},i} \overline{A}_i. \tag{A9}$$

Thus, the second case comprises slow reactions, which means quasi-steady state times longer than 1 min and $k_{c,i} < 0.01$ s. For the description of the gas-particle interface in the slow reaction case the two-film theory (Lewis and Whitman, 1924) is utilized. Therefore, the gradients in the gas and the particle phase are limited on the hypothetical gas-side and particle-side film next to the interface between the aerosol and the gas phase. The usage of the two-film theory needs the formulation of an overall gas-side mass transfer coefficient $K_{g,i}$ (Zaveri et al., 2014):

$$\frac{1}{K_{g,i}} = \frac{1}{k_{g,i}} + \frac{1}{k_{p,i}} \left( \frac{C^*_{g,i}}{\sum_j \overline{A}_j} \right) . \tag{A10}$$

The particle-side mass transfer coefficient $k_{p,i}$ is not general known and, therefore, estimated by Zaveri et al. (2014) for the limiting case ($k_{c,i} \to 0$, $q \to 0$, $Q \to 1$) of a nonreactive solute:

$$k_{p,i} = 5\frac{D_{b,i}}{r_p} . \tag{A11}$$

Thus, for slow reactions the ordinary differential equation is rewritten to:

$$\frac{d\overline{A}_i}{dt} = \frac{3}{r_p} K_{g,i} \left\{ \overline{C}_{g,i} - \frac{\overline{A}_i}{\sum_j \overline{A}_j} C^*_{g,i} \right\} - k_{c,i}\overline{A}_i . \tag{A12}$$

Finally, Zaveri et al. (2014) also provides equations for polydisperse aerosol, whereby the total average concentration $\overline{C}_{a,i,m}$ of a solute $i$ in size-section $m$ is represented by:

$$\overline{C}_{a,i,m} = \frac{4}{3}\pi r^3_{p,m} N_m A_{i,m} . \tag{A13}$$

Here, $N_m$ denotes number concentration. For fast reactions the following equation is proposed:

$$\frac{d\overline{C}_{a,i,m}}{dt} = \xi_m k_{g,i,m} \left( \overline{C}_{g,i} - \overline{C}_{a,i,m}\frac{S_{i,m}}{Q_i} \right) - k_{c,i}\overline{C}_{a,i,m} , \tag{A14}$$

and for slow reactions holds:

$$\frac{d\overline{C}_{a,i,m}}{dt} = \xi_m K_{g,i,m} \left( \overline{C}_{g,i} - \overline{C}_{a,i,m} S_{i,m} \right) - k_{c,i}\overline{C}_{a,i,m} , \tag{A15}$$

whereby, $\xi_m$ denotes the surface area of the respective size-section $m$:

$$\xi_m = 4\pi r^2_{p,m} N_m , \tag{A16}$$

and $S_{i,m}$ is the saturation ratio:

$$S_{i,m} = \frac{C^*_{g,i}}{\sum_j \overline{C}_{a,j,m}} . \tag{A17}$$

The simulations in this study have been conducted after implementation of the general model equations for polydisperse aerosol (Eq. A14 and A15). This has been done with the aim to test the kinetic approach in the box model SPACCIM for a following implementation in the 3-D regional model framework COSMO-MUSCAT (Wolke et al., 2012), which requires the implementation of the general system.

*Acknowledgements.* We thank Dr. Simon O'Meara and Dr. David O. Topping for their help with the evaluation of applicability of the weighted particle-phase bulk diffusion coefficient. We thank Prof. Dr. Ina Tegen and Prof. Dr. Hartmut Herrmann for helpful discussions and comments on the manuscript.

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
