# Peer review of "Kinetic modeling studies of SOA formation from $\alpha$ -pinene ozonolysis"

_Atmospheric Chemistry and Physics, 2017_

## Referee Comment (RC1) · Anonymous Referee #1 · 8 May 2017

The manuscript titled "Modeling studies of SOA formation from α-pinene ozonolysis" describes a kinetic gas-particle partitioning approach for simulations of SOA formation. Further the authors perform a range of sensitivity tests their they evaluate the sensitivity of this approach (model) to the particle phase bulk diffusion coefficient, the first order particle reaction rates and the inclusions of gas-phase HOM formation. The model is finally implemented for simulations of smog chamber SOA formation experiment.

**General/major comments:**

The manuscript is generally well written and addresses an important and interesting topic. However, the theory and assumptions behind the derived Eq. 2 and 3 is not easy to follow by only reading Sect. 2.2.2. E.g. for me it is not possible to understand what the $Q_i$ steady state term stand for and what the overall gas-side mass transfer coefficient is ($K_{g,i}$). If the theory is taken directly from Zaveri et al., 2014 then I suggest that you remove Eq 2-4 from Sect 2.2.2 and only put the equations in Appendix A where you describe $Q_i$ and $K_{g,i}$. In Sect. 2.2.2 you instead just mention that you have implemented and used the model/theory from Zaveri et al., 2014.

To me it is not clear what is new with the model approach presented in this study compared to Zaveri et al., 2014. The only new detail if I understand it correctly is the consideration of reversible particle phase reactions. The authors state that the kinetic gas-particle approach has been implemented in SPACCIM model and that it can be used for 3D-Eularian model simulations but this is never tested or evaluated. What is evaluated is the kinetic gas-particle approach from Zaveri et al., 2014 if I understand the manuscript correctly. If this is the case it should be clearly stated and the Sect 2.2.1 could be replaced with one sentence where it is stated that evaluated kinetic gas-particle approach from Zaveri et al., 2014 has been implemented in SPACCIM. But since I don't think you don't test how the approach is working or supposed to work in SPACCIM I don't see the point in describing this model in detail. E.g. you write in Sect. 2.2.1 that SPACCIM considers size-resolved particle and cloud droplet formation, evolution and evaporation using a one-dimentional sectional approach. But this is not used in the current study where you only consider one particle size at the time and if I understand it correctly you use a fixed particle radius in the model despite that you simulate SOA growth experiments where the particles growth over time. If you state that the tested approach will be used in the particle and cloud droplet size resolved model SPACCIM you as referee/reader want to see some results where you demonstrate how the model can simulate the particle number size distribution evolution during some SOA experiments and how the different model parameters (e.g. bulk diffusion coefficient and particle phase reaction rates) influence the particle number size distribution evolution. E.g. as in the study by Zaveri et al. (2014). There are a range of more advanced models for smog chamber SOA formation simulations such as KM-GAP and ADCHAM that the authors refer to and the model from Zaveri et al. (2014) which I think already have been implemented in regional and maybe even global chemistry transport models. Thus, the authors need to clearly demonstrate what is unique and

important with their approach. In the current form of the manuscript this is not clear.

I understand that the efficiency of the reactive uptake will be different for liquid and semi-solid particles but I don't understand how the phase-state can have such a tremendous influence on the SOA formation if the particle phase reactions are negligible slow (Fig 1b and 4c). To me the SOA phase state should not have a large influence on the SOA formation as long as the particle surface layer is composed of amorphous SOA material with the same composition as the SOA bulk. Then based on Raoul's law the saturation concentration of any gas-phase species above the particles would not differ between semi-solid and liquid particles. Even if the particles would be composed on solid crystalline salts (e.g. dry ammonium sulfate seed particles) organic molecules start to grow these particles if the gas-phase concentration of some organic species reaches above their pure-liquid saturation concentration. But the current model approach does not seem to capture this. I wonder if there is some fundamental assumption that is wrong/limiting the use of Eq. 2 and 3?

It is not clear how the SOA material that are formed after the heterogeneous reactions are treated in the model. Is it assumed to be non-volatile but still part of the amorphous SOA phase that allows more dissolution of SVOCs into the particle phase? This needs to be explained.

The model framework does not take into account the Kelvin effect if I understand this correct. If this is the case it cannot be used to study new particle formation. Please clarify and clearly state this if this is the case. I don't understand how you can assume that the particle radius is fixed. In any SOA new particle formation experiment (without seed) the particle size will grow from initially around 1 nm to larger sizes. I am skeptical to weather this model framework can handle this size dependent particle growth? Doesn't the model framework handle the gradual growth of the particles and can it take into account coagulation?

How realistic is it to represent a dimerization or oligomerization process as a first order reaction. Dimerization will involve two organic monomers. Please clearly explain what the first order particle formation reactions are supposed to represent in the model. I would also like to see some reference on what values of $k_c$ that has been used in previous studies, I am sure that is also exists some experimental evidence of appropriate values and what reactions it may be.

In Sect. 3.4 you describe an approach of how to estimate a weighted particle-phase bulk diffusion coefficient that considers particle composition (Eq. 10). I don't understand this approach. Is $D_m$ referring to the diffusion coefficient of the organic compounds in the particle-phase? Is it further assumed that the organic compounds are water-soluble then and that the particles are composed on one SOA+water+inorganics phase? I think that the particles often will be composed of several phases (e.g. one hygrophobic organic phase and one water + inorganics + some water soluble organics phase). I think that differences in $D_{org}$ over time can also be due to particle phase oligomerization processes that gradually increase the average organic molecular mass. But this Eq. 10 does not

take this into consideration. To me Eq. 10 contains to many assumptions and is not evaluated properly to be able to be justified.

**Minor comments:**

Page 2, Line 17-18: I think this sentence needs to be reformulated.

Page 2, Line 25-28: This sentence is a bit hard to understand/follow. Can you reformulate it?

Page 2, last word, remove "the"

Page 3, Line 33. The molar yields are not 6 %.

Page 4, Line 2-3. I don't understand this sentence. I understand that the diffusion coefficient will depend on the composition but not how it depends on increasing organic matter. Indirectly it can of course be influenced by the organic mass since this in turn can influence the composition.

Sect 2.1. Here you describe the experiments and the instruments that were used. What I am missing is the information about the size of the smog chamber (volume) and the wall material (e.g. Teflon) and I also miss information about how the measurement results are used in the present manuscript. E.g. how were the measurements used to derive the SOA mass in Fig 9a?  Also I miss a discussion concerning VOC and particle wall losses that is known to be important in chambers.

Sect. 3.5. Why did you decide to use $k_c = 10^{-2}$ s$^{-1}$ and $k_c = 10^{-3}$ s$^{-1}$ and $D_{org} = 10^{-12}$ m$^2$ s$^{-1}$ and $D_{org} = 10^{-14}$ m$^2$ s$^{-1}$, respectively?

Table S4. I am missing a unit on the pure liquid saturation vapor pressures of the HOM. I also think that you should mention that these molecules only represent the HOMs but that the HOMs is a family of many organic molecules presumably formed from autoxidation with a wide range of volatility.  Can you justify why you decided to use these specific HOM molecules and specify how they are assumed to be formed?  It is not either clear how these three different HOM are used in the model because in the gas-phase mechanism you only seem to have one HOM molecule.

Eq. 1. Is $A_i$ representing a concentration of a species or is $A_i$ a solute as stated on Line 17? In Eq. 2 $C$ is used to represent concentrations.

The title is a bit imprecise. Their exist many model studies of SOA formation from α-pinene ozonolysis. Can you make it more precise?

---

## Referee Comment (RC2) · Anonymous Referee #2 · 8 May 2017

**Summary**

In the manuscript at hand, Gatzsche et al. combine an established model for formation of secondary organic aerosols (SOA), SPACCIM, with a kinetic gas-particle partitioning approach following the works of Zaveri et al. (2014). Comprehensive sensitivity studies show the importance of a wide variety of model input parameters, including particle-phase diffusion coefficients, particle-phase reaction rates and particle size. A few of these parameters are tuned to fit experimental results obtained in the LEAK aerosol chamber. The manuscript is well structured and contains many useful figures. The author's however have to make a better job at explaining what (a) the model exactly does and (b) which conditions and parameters are chosen in the manifold of presented simulation results. These points are outlined in more detail in the specific comments below. The contribution of the particle phase to SOA formation via chemical reaction and impeded diffusion is a highly interesting and under-studied topic and this study fits well within the scope of ACP. The study of Gatzsche et al. makes worthwhile progress towards understanding these concepts on the level of fundamental process understanding and method development. The execution, portability and application to real world applications or chamber experiments is less convincing on the other hand. The authors assume certain mechanisms for gas-particle partitioning (absorptive partitioning, which evidently fails in the case of low diffusivity) and chemical reaction (unimolecular reaction of all reaction products, which seems like a vast simplification), but do not discuss their choices and the potential alternative mechanisms that might be at work. All parameters that are varied in this study are only loosely ($D$) or almost completely unconstrained ($k_c$ and $k_b$) and chosen more or less arbitrarily. While this is acceptable in a rather exploratory study, there is a strong mismatch between the strong conclusions drawn (e.g. Sect. 4: "*particle-phase bulk diffusion coefficient is the key parameter for the simulation of SOA formation*") and the uncertainty in molecular processes and parameters in this study. This conclusion is obtained from fitting multiple flexible parameters to only a single experiment of SOA formation. Is there only a single combination of parameters that leads to a good fit between model and experiment? What role does the general uncertainty in the gas-phase reaction mechanism and estimated volatilities play? I hence can only recommend this manuscript for publication after major revisions. I am happy to expand on my thoughts of how I think the manuscript could be improved in three sections below: general comments, specific comments and technical comments. Quotes from the manuscript are given in italics, comments begin with a bullet point.

**General Comments**

- My biggest concern is how the authors justify their assumption that organic molecules must diffuse into the particle phase in order to contribute to particle growth. This assumption is crucial for all conclusions in this paper and finds too little scrutiny.

- In Sect. 2.2.2., which parts of these equations are necessary? Entrainment and outflow are not considered in this study and there is an argument for omitting it from the equations here. The authors have to add what $M$ stands for here, is it the number of size-section? I see a term for partitioning between aqueous and organic phases, how is this utilized in this manuscript? The manuscript never mentions phase-separation, so how does this play

a role? From reading the manuscript, it is also not clear how the mass-transfer term works and when it is needed.

- Regarding the sensitivity studies (Sect. 3.1), the authors must do a better job in highlighting that some of these simulations are probably far from reality. For example, Fig. 4c interestingly shows that omitting HOMs in the mechanism leads to an increase in SOA mass. However, the SOA mass yield is so low in these simulations that I strongly doubt their usefulness for real applications. The same argument can be made for Figs. 2b and 3b. These simulations show very different reaction regimes that might not be encountered in a simulation chamber experiment. While I find it interesting to show how a system reacts under strong perturbation, the questionable practicality must be indicated more clearly in the text.

- In Sect. 3.1.2, I find it imperative to describe what the reason is that there is no particle growth at low diffusivities. I would suggest expanding on the description about what happens in the model, in words or figures.

Section 3.3

- I am not sure if I understand the purpose of Sect. 3.3. This section essentially looks at the effect of different equilibrium constants of the particle phase reaction, but it is presented as effect of different first-order reaction rates. I tend to think that an equilibrium constant is more straightforward here as the particle will most likely be in reactive equilibrium anyway. I find Fig. 5 very instructive and what happens in Fig. 6 is just that the share of r-products (orange bands) is reduced, is that correct? This is not a very exciting result given that all reaction rates are arbitrary guesses, or are the amounts of p-products and HOMs also affected? A normalized sensitivity coefficient of the mass of r-products and p-products to a reaction rate might be much more instructive. On a different note, would it be possible to connect these cases to real scenarios, e.g. by making more realistic assumptions of $k_c$ and $k_b$ for examples at high RH / low RH?

Sect. 3.4

In this section, it is very hard to follow what exact calculations were performed. This leads to many open questions: What are the initial conditions for these experiments, especially $x_{inorg}$ and $x_{water}$? What were the diffusivties $D_{inorg}$ and $D_{water}$? On which humidity is the water concentration based? What is the hygroscopicity of these particles, how is it determined? Why is it safe to assume that organic and inorganic phases are mixed? Are we seeing an effect of $D_{inorg}$ or an effect of $D_{water}$ here?
It would be interesting to see $D_m$ plotted against humidity, following this framework. You could compare the humidity-dependence of your $D_m$ for self-diffusion in SOA to the values of tracer diffusion determined in Berkemeier et al. (2014), Lienhard et al. (2015), Price et al. (2015) or Berkemeier et al. (2016).

It would be helpful if it were more explicitly explained what happens in the simulations. What does the time profile of $D_m$ look like? I suppose it increases due to the smaller contribution of inorganics.

On, p. 17, l. 33, the wording seems suboptimal. You write that using a constant bulk diffusion coefficient lowers the total SOA mass. Along the lines of the argumentation, this should be formulated the other way around: implementing a weighted particle-phase diffusion scheme increases total SOA mass. I have to wonder though, what of this effect is due to water and what is due to inorganics? It seems very clear to me that you would need to compare the diffusivity of the equilibrium SOA-water mixture and only add in the inorganics. Otherwise, you mainly make the argument that humidity leads to more SOA mass and not so much investigate the effects of a time-dependent diffusivity coefficient.

**Specific Comments**

p. 12, l.27-28

*"No initial organic mass OM0 is utilized for the simulations with HOMs."*

- This change seems arbitrary when first reading the article and gets only clear upon further reading. I wonder how the simulations would look like if no initial organic mass would be utilized for the simulations without HOMs? I find that would be much more instructive.

p. 13, l.2

*"A rapid condensation of HOMs occurs due to their low vapor pressures (Fig. 4a)."*

- How can this be seen in Fig. 4a? It seems not easy to see whether the solid lines take off sooner than the dashed lines. Is also does not appear as if the solid lines separate from the dashed lines within the early moments of the experiment/simulation, but rather over the first half of the experiment. Maybe referring to Fig. 5 would be helpful here if it is showing what you mean here?

p. 13, Fig. 3

- Since particle radius is reduced down to 11 nm, are Kelvin effects considered in this study? I don't see this mentioned in the manuscript.

p. 14, l.5-7

*"This may be the reason for the convergence of formed SOA mass for $k_c = 1$ $s^{-1}$ combined with $D_b = 10^{-14}$ $m^2$ $s^{-1}$ and the overall more effective SOA formation without consideration of HOMs for semisolid particles ($D_b = 10^{-18}$ $m^2$ $s^{-1}$, Fig. 4c)."*

- Would it be possible to give more explanation on this odd result of Fig. 4c? Should this be left out if not realistic?

p. 14, l.10

*"In general the HOMs provide about 27% of the simulated final total SOA mass and introduce SOA mass formation."*

- How does this compare to the molar yields in Berndt et al. (2016)? In addition, there is a comma missing after "in general".

p. 15, l. 8f

*"The main benefit of the implementation of a sufficiently fast backward reaction ($k_b \geq 10^{-3}$ $s^{-1}$) is the asymptotic curve shape of the SOA mass for proceeding simulation times. This behavior is also observed during chamber studies, which indicate an equilibrium state of the gas and the particle phase after a proceeding oxidation time."*

- This reads interesting, but is difficult to understand without practical experience with smog chambers. Could this statement be explained in more detail and justified with examples?

p. 18, l. 25

*"Consideration of the weighted particle-phase bulk diffusion coefficient and additional considered HOMs lead to a faster SOA mass increase at the beginning of the simulation. The decreasing particle-phase bulk diffusion coefficient due to the uptake of further organic material and the backward reactions in the particle phase induce a flattening of the mass increase."*

- Where can this be seen? Why should a weighted particle-phase diffusion coefficient generally speed up SOA formation?

p. 20, Figure 8

- In Fig. 8, it is very difficult to understand the simulation conditions of each plotted line. Am I correct that the dashed lines are showing the same results in both panels? This would be worthwhile pointing out! It would be easier to see if both panels would show the same range on the y-axis. It is also difficult to spot the line that fits the experimental data well in Fig. 8b, maybe draw these on top of the markers or highlight it in another way.

p. 21, l. 4ff

*"For the preferred model setup of Fig. 8b with $k_b = 10^{-2}$ $s^{-1}$, the simulation is in a very good agreement with the measured concentration decrease of α-pinene (Fig. 10a)."*

- Why does the α-pinene concentration depend on $k_b$?

p. 21, l. 7

*"The depletion of ozone is slightly overestimated by the model after 1.5 hours (Fig. 10b). The measured gas-phase concentration of pinonaldehyde is underestimated by the model (Fig. 10c)."*

- Do you have ideas what could be the underlying reasons here? This would add much more to the manuscript than just plotting results.

**Technical Comments**

Figure 6 caption

*"… both combined with different fast chemical backward reactions $k_b$."*

- This sentence is difficult to understand in general and might deserve revision, but I believe what you mean here is "differently".

*"The reference simulations for the regarding $k_c$ and no $k_b$ are shown with dashed lines."*

- Do you mean "respective" instead of "regarding"? Do you mean "without backward reaction" instead of "no $k_b$"?

p. 18, l. 33ff

*"After 1 h simulation time, it is obvious that the simulated concentration profile agree well with the experimentally observed SOA mass with a backward reaction rate constant of $k_b = 10^{-2}$ s$^{-1}$."*

- Do the authors mean:

  "After 1 h simulation time, it is obvious that the simulated concentration profile agree*s* well with the experimentally observed SOA mass *when using a* backward reaction rate constant of $k_b = 10^{-2}$ s$^{-1}$."

Figure S5b

- This figure is not discussed in the manuscript.

**References**

Berkemeier, T., Shiraiwa, M., Pöschl, U., and Koop, T.: Competition between water uptake and ice nucleation by glassy organic aerosol particles, Atmos. Chem. Phys., 14, 12513-12531, 10.5194/acp-14-12513-2014, 2014.

Berkemeier, T., Steimer, S. S., Krieger, U. K., Peter, T., Poschl, U., Ammann, M., and Shiraiwa, M.: Ozone uptake on glassy, semi-solid and liquid organic matter and the role of reactive oxygen intermediates in atmospheric aerosol chemistry, Phys. Chem. Chem. Phys., 18, 12662-12674, 10.1039/C6CP00634E, 2016.

Lienhard, D. M., Huisman, A. J., Krieger, U. K., Rudich, Y., Marcolli, C., Luo, B. P., Bones, D. L., Reid, J. P., Lambe, A. T., Canagaratna, M. R., Davidovits, P., Onasch, T. B., Worsnop, D. R., Steimer, S. S., Koop, T., and Peter, T.: Viscous organic aerosol particles in the upper troposphere: diffusivity-controlled water uptake and ice nucleation?, Atmos. Chem. Phys., 15, 13599-13613, 10.5194/acp-15-13599-2015, 2015.

Price, H. C., Mattsson, J., Zhang, Y., Bertram, A. K., Davies, J. F., Grayson, J. W., Martin, S. T., O'Sullivan, D., Reid, J. P., Rickards, A. M. J., and Murray, B. J.: Water diffusion in atmospherically relevant alpha-pinene secondary organic material, Chem. Sci., 6, 4876-4883, 2015.

---

## Author Comment (AC1) · 25 Aug 2017

**Authors' Response to referee comments for 'Modeling studies for SOA formation from $\alpha$-pinene ozonolysis'**

The authors would like to thank both referees for the careful consideration of the manuscript and for the constructive comments and suggestions made to improve the manuscript. According to the reviewers' comments, the authors have further improved the manuscript. All comments and changes in the manuscript are addressed below. In the case, we do not concur with the reviewers' comments, adequate reasons are given.

Comments from referees are presented in blue, these are followed by authors' response in black, and changes to the manuscript are in green.

**Referee #1**

**Major/General comments**

However, the theory and assumptions behind the derived Eq. 2 and 3 is not easy to follow by only reading Sect. 2.2.2. E.g. for me it is not possible to understand what the $Q_i$ steady state term stand for and what the overall gasside mass transfer coefficient is ($K_{g,i}$). If the theory is taken directly from Zaveri et al. (2014) then I suggest that you remove Eq. 2–4 from Sect. 2.2.2 and only put the equations in Appendix A where you describe $Q_i$ and $K_{g,i}$. In Sect. 2.2.2 you instead just mention that you have implemented and used the model/theory from Zaveri et al. (2014).

In this Section, Sect. 2.2.2, the authors aimed at clarifying which equations of the approach proposed by Zaveri et al. (2014) have been implemented in the box model SPACCIM. Therefore, we have presented Eqs. (2−4) to clarify that the equations for a general system and polydisperse particles are utilized. In the paper of Zaveri et al. (2014), at first the general solution for a closed system is derived, then single particle equations with the approximation for a general system including fast and slow reactions are deduced and at the end the polydisperse equations for the two approximations are given. SPACCIM is a spectral parcel model (Wolke et al., 2005) and for this reason, the polydisperse equations are appropriate for implementation. The equations suitable for a general system were utilized because this approach should be subsequently applied in our 3-D model COSMO-MUSCAT (Wolke et al., 2012) and, therefore, extensive sensitivity studies are needed. Further, the mass balance equations of SPACCIM (Eqs. (6) and (8) of the presented paper) are based on Eqs. (2−4) and, consequently, it is necessary to introduce the model equations accounting for gas-to-particle phase mass transfer. Whereas, Eqs. (2−4) are deduced shortly in the Appendix of the presented paper to introduce the theory behind the kinetic approach for the sake of completeness. According to the reviewer's comment the description of $Q_i$ and $K_{g,i}$ is expanded in Sect. 2.2.2 as follows:

$Q_i$ represents the ratio of the average particle-phase concentration $\overline{A}_i$ to the surface concentration $A_i^S$ at steady-state and is named quasi-steady-state term (see Appendix A). $N_m$ denotes the number concentration, $r_{p,m}$ the respective particle radius, $k_{g,i}$ is the gas-side mass transfer coefficient, and

$K_{\mathrm{g},i}$ is the overall gas-side mass transfer coefficient, which is needed for the application of the two film theory (see Appendix A for details).

To me it is not clear what is new with the model approach presented in this study compared to Zaveri et al. (2014) The only new detail if I understand it correctly is the consideration of reversible particle phase reactions. The authors state that the kinetic gas-particle approach has been implemented in SPACCIM model and that it can be used for 3D-Eularian model simulations but this is never tested or evaluated. What is evaluated is the kinetic gas-particle approach from Zaveri et al. (2014) if I understand the manuscript correctly. If this is the case it should be clearly stated and the Sect. 2.2.1 could be replaced with one sentence where it is stated that evaluated kinetic gas-particle approach from Zaveri et al. (2014) has been implemented in SPACCIM. But since I don't think you don't test how the approach is working or supposed to work in SPACCIM I don't see the point in describing this model in detail. E.g. you write in Sect. 2.2.1 that SPACCIM considers size-resolved particle and cloud droplet formation, evolution and evaporation using a one-dimensional sectional approach. But this is not used in the current study where you only consider one particle size at the time and if I understand it correctly you use a fixed particle radius in the model despite that you simulate SOA growth experiments where the particles growth over time. If you state that the tested approach will be used in the particle and cloud droplet size resolved model SPACCIM you as referee/reader want to see some results where you demonstrate how the model can simulate the particle number size distribution evolution during some SOA experiments and how the different model parameters (e.g. bulk diffusion coefficient and particle phase reaction rates) influence the particle number size distribution evolution. E.g. as in the study by Zaveri et al. (2014).

The presented paper comprises different novelties concerning application as well as development of the kinetic approach. The utilization of the kinetic approach to a multiphase chemistry mechanism describing $\alpha$-pinene degradation and SOA formation, extensive sensitivity studies concerning this reaction system, and simulating a chamber study are three novelties. On the model development level, the implementation of additional backward reactions in the particle phase and a composition dependent diffusion coefficient $D_{\mathrm{m}}$ are new for this model approach. Further, the influence of HOMs on SOA formation is outlined in detail and accounts for existing vapor pressure estimation uncertainties on the partitioning of this compound groups. Further, supporting results concerning the applicability of the weighted bulk diffusion coefficient will be presented later in this document and has been added to the presented paper, which is an additional novelty. Therefore, the presented paper comprises several new results, which are not part of the investigations of Zaveri et al. (2014).

Within the LEAK chamber studies, seed aerosol with a quite narrow particle distribution is injected, which can be captured in one bin in a model. Due to the SOA formation, the aerosol spectrum is shifted to a larger aerosol size, but stays almost monodisperse. Therefore, for the simulation of the chamber studies, the mean radius of the initialized aerosol spectrum is utilized for the model initialization. The authors know that the fixed particle radius can not model the reality, however, the aim of this approach was to avoid overlapping sensitivity effects within this investigation.

The functionality of the polydisperse model features, using the kinetic approach, were evaluated with test scenarios because these features are important for the subsequent application, e.g. in the 3-D model. However, for the conducted sensitivity studies the consideration of a polydisperse aerosol distribution will increase the degree of freedom as well as the complexity and for the simulation of the LEAK chamber studies, this feature was not required because of the nearly monodisperse aerosol spectrum within the experiment.

Based on the reviewer's comment, the novelty of the paper is highlighted straighter in the introduction and information on the narrow range of seed aerosol particles is given in the chamber experiment section as follows:

Sect. 1 Introduction:

[revised manuscript text omitted]

...There are a range of more advanced models for smog chamber SOA formation simulations such as KM-GAP and ADCHAM that the authors refer to and the model from Zaveri et al. (2014) which I think already have been implemented in regional and maybe even global chemistry transport models.

Thus, the authors need to clearly demonstrate what is unique and important with their approach. In the current form of the manuscript this is not clear.

As described in our original manuscript, more kinetic model approaches exist, which also consider the particle-phase diffusion coefficient as a model parameter. The applications of these very detailed models differ from the aim of the investigations shown in the presented paper. KM-GAP considers a detailed description of the particle phase and, therefore, the influence of the aerosol composition and morphology on the gas-to-particle mass transfer can precisely investigated with this model (Shiraiwa et al., 2013). Further, the model is utilized for investigations concerning water uptake as well as ice nucleation by organic aerosols (Berkemeier et al., 2014). With ADCHAM, smog chamber studies beyond SOA formation are investigated, e.g. simulations concerning organic salt formation, the influence of heterogeneous reactions, and chamber wall effects (Roldin et al., 2014). This advanced model features are beyond the fields of application of SPACCIM, which treats the particle phase as bulk and, therefore, not contains the required model infrastructure. Consequently, more advanced models require the specification of a greater number of model parameters, e.g. to describe the particle phase in more detail. Since not all of these parameters might be determined by experimental studies as a consequence thereof more assumptions for additional parameters have to be made (e.g., diffusion coefficient for every particle layer). A recent study by Berkemeier et al. (2016) presented an approach to extend the dimensions of the experimental input data to maintain the advanced model description for a complex system. Nevertheless, this method is only applicable for detailed experimental studies on distinct compounds.

However, the kinetic approach of Zaveri et al. (2014) utilizes the basic assumption of a constant particle-phase bulk diffusion coefficient and an analytical solution for the description of SOA formation. As investigated by O'Meara et al. (2016), numerical solutions for aerosol transformation processes are computationally expensive and, therefore, a numerical approach with a constant bulk diffusion coefficient requires approximately 20 times more computational effort than an analytical approach (O'Meara et al., 2017). The analytical kinetic approach of Zaveri et al. (2014) is preferred to save computational effort and, therefore, was analyzed/tested for utilization in future 3-D model simulations.

To the authors' knowledge, the results of a 3-D model utilizing the approach of Zaveri et al. (2014) for SOA formation is up to now not presented in a peer-reviewed journal. Nevertheless, the authors showed preliminary results of the 3-D model COSMO-MUSCAT with this kinetic partitioning approach at the EGU General Assembly 2017 (http://meetingorganizer.copernicus.org/EGU2017/EGU2017-4557.pdf).

However, the authors are not aware of the application of the kinetic approach from Zaveri et al. (2014) in a regional or global model published in a peer-reviewed journal.

The highlights of this manuscript are already addressed in response to the first comment raised by the reviewer #1.

I understand that the efficiency of the reactive uptake will be different for liquid and semi-solid particles but I don't understand how the phase-state can have such a tremendous influence on the SOA formation if the particle phase reactions are negligible slow (Fig 1b and 4c). To me the SOA phase state should not have a large influence on the SOA formation as long as the particle surface layer is composed of amorphous SOA material with the same composition as the SOA bulk. Then based on Raoul's law the saturation concentration of any gas-phase species above the particles would not differ between semi-solid and liquid particles. Even if the particles would be composed on solid crystalline salts (e.g. dry ammonium sulfate seed particles) organic molecules start to grow these particles if the gas-phase concentration of some organic species reaches above their pure-liquid saturation concentration. But the current model approach does not seem to capture this. I wonder if there is some fundamental assumption that is wrong/limiting the use of Eq. 2 and 3?

The total particle-phase concentration within this kinetic approach depends on the gas-phase concentration of the condensable organic compounds, the transport of this compounds to the particle surface as well as the transport into the particle. As described in Mai et al. (2015), three different limitations for SOA formation exist for a kinetic approach: gas-phase-diffusion-limited partitioning, interfacial-transport-limited partitioning, and particle-phase-diffusion-limited partitioning. Due to the relatively small particle sizes and the mass accommodation coefficient of $\alpha = 1$ within the majority of the presented studies, consequently particle-phase-diffusion-limited partitioning can be observed for decreased particle-phase bulk diffusion coefficients within several sensitivity studies. The particle-phase bulk diffusion coefficient determines the transport into the particle bulk. An important parameter to interpret the interaction of the individual model parameters characterizing the particle bulk is the dimensionless diffusion-reaction parameter $q_i$ (Zaveri et al., 2014), which is defined as the ratio of the particle radius $r_\mathrm{p}$ and the reacto-diffusive length $\sqrt{D_{\mathrm{b},i}/k_{\mathrm{c},i}}$ (Pöschl et al., 2007):

$$q_i = r_\mathrm{p}\sqrt{\frac{k_{\mathrm{c},i}}{D_{\mathrm{b},i}}} \, . \tag{1}$$

Therein, $D_{\mathrm{b},i}$ is the particle-phase bulk diffusion coefficient and $k_{\mathrm{c},i}$ represents the pseudo-first-order rate constant for particle reactions of the compound $i$. In Fig. 4 of Zaveri et al. (2014), the normalized concentration profiles for the steady-state case are displayed for different $q_i$ (see Fig. 1a). For the

[Figure]

Figure 1: a) Normalized $(A(r)/A^\mathrm{s})_\mathrm{ss}$ profiles as a function of $r/r_\mathrm{p}$ ($R_\mathrm{p}$ in the variable declaration of Zaveri et al. (2014)) for different values of the dimensionless diffusion-reaction parameter $q_i$. Therein, $A(r)$ stands for the particle-phase concentration $A$ in dependence of the particle radius $r$, which is normalized by the surface concentration of the particle $A^\mathrm{s}$, and is displayed for the steady state (indicated by the index "ss"); Figure taken from Zaveri et al. (2014). b) $\log(q)$ values for the sensitivity study of case 1 from Table 1 in Gatzsche et al. (2017).

simulations performed in the sensitivity study of case 1 in Table 1 of the presented paper, $q_i$ cover the entire spectrum displayed in Fig. 1a. From Eq. (A3) of the presented paper it is obvious, that the total organic mass in the particle bulk can be calculated by the integral of the solute concentration over the sphere volume. From the distribution of $(A(r)/A^\mathrm{s})_\mathrm{ss}$ in Fig. 1a, we can conclude that with an increasing value of $q$ the particulate organic mass have to decrease because the organic mass only concentrates like a film on the outside of the particle. With regard to the variation of the diffusion coefficient of the sensitivity study case 1 (see Table 1 of the presented paper), the following values of $q$ have to be considered. For these studies, a constant particle radius of $r_\mathrm{p} = 35\,\mathrm{nm}$ have been utilized. For

[Figure]

Figure 2: Contour plots of a) the quasi-steady-state timescale $\tau_{QSS}$ of the particle-phase and b) the quasi-steady-state parameter $Q = (\overline{A}/A^S)_{QSS}$, both as functions of the particle-phase bulk diffusion coefficient $D_b$ and the pseudo-first-order rate constant $k_c$ for a fixed particle size of $d_p=100\,$nm; Figure taken from Zaveri et al. (2014).

the highest rate constant of $k_c = 1\,s^{-1}$ and $D_b = 10^{-9}\,m^2\,s^{-1}$, the dimensionless diffusion-reaction parameter equals $q = 1.11 \times 10^{-3}$ (see Fig. 1b). When decreasing the reaction rate to $k_c = 10^{-6}\,s^{-1}$, $q$ calculates to $q = 1.11 \times 10^{-6}$ (Fig. 1b). Thus for liquid particles ($q \to 0$), the concentration ratio at steady state, displayed in Fig. 1a, is near unity in the whole particle. Therefore, the organic mass contained in the particle phase reaches their maximum. For semi-solid particles $D_b = 10^{-21}\,m^2\,s^{-1}$ and $k_c = 1\,s^{-1}$, $q$ calculates to $q = 1.11 \times 10^3$ (see Fig. 1b). For this value of $q$, a highly non-uniform concentration ratio profile can be observed in the particle, which means that the condensed organic compounds are mainly located at the surface of the particle (like a film, see Fig. 1a). The organic mass reaches a minimum. Therefore, no further condensation of organic compounds on the particle will occur because a particle-phase-diffusion-limited partitioning occurs (Mai et al., 2015). For slow particle-phase reactions $k_c = 10^{-6}\,s^{-1}$, the dimensionless diffusion-reaction parameter equals to $q = 1.11$ (Fig. 1b), which indicates a slightly non-uniform concentration ratio in the particle (see Fig. 1a). In this case, the concentration gradient in the particle phase is not high, however, the particle phase reactions are too slow to shift the equilibrium towards the particle phase and leading to further condensation from the gas phase.

It is not clear how the SOA material that are formed after the heterogeneous reactions are treated in the model. Is it assumed to be non-volatile but still part of the amorphous SOA phase that allows more dissolution of SVOCs into the particle phase? This needs to be explained.

To address the reviewer's comment, the description of the considered particle phase chemistry has been extended in Sect. 3.1.2. as follows:

As indicated in Eq. (1), the organic compounds, which are partitioned from the gas phase into the particle phase, can further react in the particle phase with a constant reaction rate $k_c$. The from the gas into the particle phase partitioned organic compounds are named p-products. The products, which have been caused due to the reactions in the particle phase, are termed r-products. The r-products do not stay in equilibrium with the gas-phase compounds and, therefore, can not evaporate from

the particle phase. The r-products comprise particle-phase compounds resulting from aging of the condensed organic compounds (e.g., dimers, trimers or oligomers).

The model framework does not take into account the Kelvin effect if I understand this correct. If this is the case it cannot be used to study new particle formation. Please clarify and clearly state this if this is the case. I don't understand how you can assume that the particle radius is fixed. In any SOA new particle formation experiment (without seed) the particle size will grow from initially around 1 nm to larger sizes. I am skeptical to weather this model framework can handle this size dependent particle growth? Doesn't the model framework handle the gradual growth of the particles and can it take into account coagulation?

The Kelvin effect describes the change of the vapor pressure due to a curved liquid-vapor interface and is especially important for small particles because of their higher curvature (Seinfeld and Pandis, 2006; Pruppacher and Klett, 2010). The partial vapor pressure of a compound $i$ over a curved interface $p_i^{\ominus}$ (atm) can be related to the vapor pressure over a flat surface $p_{\mathrm{sat},i}$ (atm) with the following equation (Riipinen et al., 2010):

$$p_i^{\ominus} = x_i \gamma_i p_{\mathrm{sat},i} \exp\left(\frac{4 M_i \sigma_{\mathrm{p}}}{R T_{\mathrm{p}} \rho_{\mathrm{p}} d_{\mathrm{p}}}\right) = x_i \gamma_i p_{\mathrm{sat},i} \exp \zeta \tag{2}$$

The exponential term of Eq. (2) describes the Kelvin effect, whereas the multiplication with the mole fraction $x_i$ and the activity coefficient $\gamma_i$ owes to Raoult's law. Further, $M_i$ is the molar weight (g mol$^{-1}$) of the species $i$ in the particle with the diameter $d_{\mathrm{p}}$ (m), $\rho_{\mathrm{p}}$ (kg m$^{-3}$) the density, and $\sigma_{\mathrm{p}}$ (N m$^{-1}$) the surface tension of the particle. The particle temperature $T_{\mathrm{p}}$ and the universal gas constant $R$ (8.314 J mol$^{-1}$ K$^{-1}$) are included. The vapor pressure over the curved interface always exceeds the vapor pressure over a flat surface considering the same species. The surface tension is defined as the amount of energy, which is required to increase the area of a surface by 1 unit (Seinfeld and Pandis, 2006). The pure water surface tension $\sigma_{\mathrm{w}0}$ can be estimated in dependence of the temperature $T$ (in K, Pruppacher and Klett, 2010):

$$\sigma_{\mathrm{w}0} = 0.0761 - 1.55 \times 10^{-4}(T - 273)\,\mathrm{N\,m^{-1}}, \tag{3}$$

within a temperature range of -40 to 40°C. Dissolution of other compounds in water alter its surface tension. Salts increase the surface tension of the droplet, e.g. for ammonium sulfate the following expression is valid (Seinfeld and Pandis, 2006):

$$\sigma_{\mathrm{w}}(m_{\mathrm{(NH_4)_2SO_4}}, T) = \sigma_{\mathrm{w}0} + 2.17 \times 10^{-3} m_{\mathrm{(NH_4)_2SO_4}}, \tag{4}$$

with $m_{\mathrm{(NH_4)_2SO_4}}$ (mol l$^{-1}$) the molality of ammonium sulfate. In contrast to that, organics decrease the surface tension of a droplet because their surface tension is lower than that of pure water. For pure saturated organic liquids, the surface tensions alter between 20 and 40 mN m$^{-1}$ in the temperature range $280 - 320$ K (Jasper, 1972; Seinfeld and Pandis, 2006; Butt et al., 2004). In aerosol particles, water, dissolved inorganic salts, and organics are mixed and together effect the resulting surface tension. Different investigations on the surface tension of mixed aerosols/cloud droplets exist (Facchini et al., 1999; Hitzenberger et al., 2002; Ervens et al., 2004, 2005). Facchini et al. (1999) proposed a specific relation between the surface tension $\sigma$ and the dissolved organic carbon concentration $[C]$ (in mol l$^{-1}$):

$$\sigma = 72.8 - 0.0187\, T \ln\left(1 + 628.14\,[C]\right)\mathrm{mN\,m^{-1}}, \tag{5}$$

which is derived by fitting the Szyszkowski-Langmuir equation to their measurement data from Po valley fog. Whereby, Ervens et al. (2004) state that the application of Eq. (5) is only appropriate for higher molecular weight organic compounds because of the huge overestimation of the surface tension effect of small dicarboxylic acids (Shulman et al., 1996). The value of the surface tension for

a complex mixed aerosol particle/droplet is not definitely known. However, Eq. (2) indicates that the Kelvin effect is a correction to the vapor pressures over a flat surface $p_{\mathrm{sat},i}$, which are utilized in the presented paper. If only organic droplets are considered, the correction term is for particles smaller than 10 nm and surface tensions higher than 30 mN m$^{-1}$ greater than 2 (see Fig. 3a). For water

[Figure]

Figure 3: Kelvin term ($\exp\zeta$ in Eq. 2) depending on the surface tension $\sigma$ and the particle radius $r_{\mathrm{p}}$ for a) a typical surface tension range for organics and b) a typical surface tension range for water with ammonium sulfate.

containing ammonium sulfate, the Kelvin term reaches greater values for particles with $r_{\mathrm{p}} \leq 10$ nm, which might imply a larger effect for the first organic compounds that condense on small particles (see Fig. 3b). Thus, only for the smallest particles in our simulations ($r_{\mathrm{p}} = 11$ nm) the Kelvin effect might affect the initial condensation of the organic compounds on the particles (see Fig. 3a, b). Nevertheless in the presented paper, a group contribution method (EVAPORATION, Compernolle et al., 2011) is applied to estimate the liquid vapor pressures of the condensing organic species because no accurate measurements for the compounds are available as well as it is impracticable to measure the vapor pressures for the variety of compounds. An investigation of O'Meara et al. (2014) reveals that the vapor pressure estimates from the different group contribution methods vary from each other and deviate from existing measurements up to six orders of magnitude. Additionally, Kurtén et al. (2016) show the differences between the vapor pressures estimated by three different group contribution methods and COSMO-RS (conductor-like screening model for real solvents, Eckert and Klamt, 2002). Therein, 8 orders of magnitude lower vapor pressures are estimated by using group contribution methods than COSMO-RS for some highly oxidized monomers. Therefore, the correction of the vapor pressure by the Kelvin effect might be in the order of the error range of the applied group contribution method. For this reason, we have not considered the Kelvin effect in our calculations, but it is planned to be included in future investigations.

In the study of Zaveri et al. (2014), the Kelvin effect was also not considered for simplicity. In KM-GAP, the Kelvin effect is only considered for water within the Köhler equation (Shiraiwa et al., 2012). However, Roldin et al. (2014) stated that ADCHAM comprises the Kelvin effect for the organic compounds, utilizing surface tensions of 50 mN m$^{-1}$ according to the study from Riipinen et al. (2010).

Further, the simulations in the presented paper do not investigate new particle formation because seed aerosol particles are taken into account.

The parcel model SPACCIM includes coagulation as proposed in Wolke et al. (2005), wherein all microphysical features of SPACCIM have been already provided and tested. However, in the presented studies this process was not a matter of interest. The focus of the presented paper is to characterize the influence of the different model parameters on the SOA formation and not on aerosol microphysics. As explained before, the LEAK chamber studies are initialized with a quite narrow particle spectrum

with the mean radius of 35 nm. Therefore, we did not additionally vary the particle radius within this sensitivity study and use $r_p = 35$ nm, which was the mean radius of the initialized seed aerosol particles for the LEAK experiment.

To address the reviewer's comment concerning the Kelvin effect, in Sect. 3.2 it is clarified that new particle formation is not investigated and an additional section concerning the limitations of the model studies have been added. Further, in Sect. 2.2.1 the authors point out more clearly that the microphysical features of the utilized model framework are already published and are not part of the presented paper.

Sect. 2.2.2:

The size-resolved cloud microphysics of deliquesced particles and droplets including cloud droplet formation, evolution, and evaporation is considered using a one-dimensional sectional approach. Further microphysical features of SPACCIM are already described in Wolke et al. (2005) and results owing to these processes are presented in Tilgner et al. (2013); Rusumdar et al. (2016); Hoffmann et al. (2016). The implemented multiphase chemical model applies a high-order implicit time integration scheme, which utilizes the specific sparse structure of the model equations (Wolke and Knoth, 2002). SPACCIM was originally developed for parcel model studies, whereby, the considered air parcel can follow real or artificial trajectories. However, the partitioning of organic gases towards the particle phase was not considered in the original SPACCIM and the model was not exclusively designed for application on aerosol chamber studies. The existing model framework has been extended by gas-to-particle mass transfer via a kinetic partitioning approach (Zaveri et al., 2014), see Sect. 2.2.2 for details. Due to the focus of these studies on modeling aerosol chamber studies of gasSOA formation for monodisperse aerosol without entrainment and coagulation, microphysical processes are not included in the results of this study.

Sect. 3.2:

Nevertheless, the conducted simulations account not for new particle formation. The simulations without initial organic mass is consistent to previous studies, initialized with inorganic seed aerosol particles.

Sect. 3.6 Limitations of the present studies:

The presented model studies do not account for the Kelvin effect. The Kelvin effect describes the change of the vapor pressure due to a curved liquid-vapor interface and is especially important for small particles because of their higher curvature (Seinfeld and Pandis, 2006; Pruppacher and Klett, 2010). The vapor pressure of a compound $i$ over a flat surface $p_{sat,i}$ (atm) can be corrected to the partial vapor pressure over a curved interface $p_i^{\ominus}$ (atm, Seinfeld and Pandis, 2006). The correction factor depends strongly on the particle size and the surface tension of the considered aerosol particle/droplet. The surface tension varies with the composition of the aerosol particle (Facchini et al., 1999; Hitzenberger et al., 2002; Ervens et al., 2004, 2005), e.g. it is increased by dissolved salts (Seinfeld and Pandis, 2006) and decreased by organic compounds (Facchini et al., 1999; Ervens et al., 2005). However, for the estimation of the vapor pressures for the partitioning compounds a group contribution method (EVAPORATION, Compernolle et al., 2011) is applied in this study. An investigation of O'Meara et al. (2014) reveals that the vapor pressure estimates from the different group contribution methods vary from each other and deviate from existing measurements up to six orders of magnitude. Further, Kurtén et al. (2016) showed the differences between the vapor pressures estimated by three different group contribution methods and COSMO-RS (conductor-like screening model for real solvents, Eckert and Klamt, 2002). Therein, 8 orders of magnitude lower vapor pressures are estimated by group contribution methods than COSMO-RS for some highly oxidized monomers. Therefore, the correction of the vapor pressure by the Kelvin effect might be in the order of the error range of the applied group contribution method. For this reason, we have not considered the Kelvin effect in our calculations.

How realistic is it to represent a dimerization or oligomerization process as a first order reaction. Dimerization will involve two organic monomers. Please clearly explain what the first order particle formation reactions are supposed to represent in the model. I would also like to see some reference on what values of $k_c$ that has been used in previous studies, I am sure that is also exists some experimental evidence of appropriate values and what reactions it may be.

As already stated in Zaveri et al. (2014), the particle-phase reactions and their associated reaction rates are not well defined by measurements and, therefore, a constant pseudo-first-order-rate constant is introduced to approximate the particle-phase reactivity. Camredon et al. (2010) also described oligomerization with a pseudo-first-order loss rate from the monomers to consider the particle-phase reactivity in a box model study. Trump and Donahue (2014) utilize thermodynamic calculations for the equilibrium constants of organic compounds to describe oligomerization in a VBS framework. A detailed description of dimerization is contained in ADCHAM (Roldin et al., 2014), which utilizes measurements of the reaction rate coefficients for the formation of peroxyhemiactetals proposed by Antonovskii and Terent'ev (1967). The reaction rate coefficients of Antonovskii and Terent'ev (1967) are the only available measured values, which are also proposed in the review of Ziemann and Atkinson (2012).

However, Seinfeld and Pandis (2006) proposed that SOA often comprises high-molecular-weight species that are a kind of oligomers (e.g., dimers, trimers, tetramers, etc.) resulting from the reaction of VOC oxidation products (Kalberer et al., 2004; Gao et al., 2004b,a; Tolocka et al., 2004; Hall IV and Johnston, 2011). Further, oligomerization is enhanced due to the presence of strong acids (e.g., sulfuric acid, Jang et al., 2002, 2003; Iinuma et al., 2004). Barsanti and Pankow (2004, 2005) and Barsanti and Pankow (2006) suppose due to thermodynamic studies that accretion reactions are most likely for acids. Nevertheless, none of these recent studies provide a reaction mechanism and related reaction rate constants for the particle phase reactions. Up to now, the existence and qualitative values of oligomerization in the particle phase are investigated. In the study of Hosny et al. (2016), additional to the micro-viscosity measurements, monomer:dimer:trimer:tetramer signal intensities are shown and the oligomerization process is compared with the simulation of the oligomerization of oleic acid as model compound. KM-SUB (Shiraiwa et al., 2010) is utilized for the simulations in this study. However, the reactions and reaction rate coefficients therein are valid for oleic acid, which is a more investigated reaction system.

It is clear to the authors that the dimerization process can not described by using a first-order rate constant limited to the production of oligomers. This process is more accurately captured by an equilibrium reaction, but the equilibrium constants are also not well defined. Therefore, the authors' consideration was to add an additional backward reaction to initially review/verify the sensitivity of SOA formation on this additional feature. In the detailed approach of Roldin et al. (2014), the formation and degradation of dimers is also treated with two separate reactions.

To address the reviewer comment on the pseudo-first-order rate constant, text is modified/added in Sect. 3.3 and a description of the limitations of the current particle-phase reactivity scheme is added to the new inserted Sect. 3.6.

Text added/modified to Sect. 3.3:
Organic aerosol-phase reactions can be irreversible reactions such as oxidation reactions or reversible reactions as for instance dimerization/oligomerization (Hallquist et al., 2009; Ziemann and Atkinson, 2012). For the observed particle-phase dimerization, different possible reaction mechanisms can be found in the literature: (i) hemiacetal formation due to reactions between alcohols and aldehydes or carbonyl compounds (Iinuma et al., 2004; Ziemann and Atkinson, 2012; Carey and Sundberg, 2007), (ii) peroxyhemiacetal formation between hydroperoxides and carbonyl compounds (Tobias and Ziemann, 2000; Ziemann and Atkinson, 2012), (iii) aldol reaction products from the acid-catalyzed

dimerization of a ketone or aldehyde (Carey and Sundberg, 2007; Casale et al., 2007), and (iv) esterfication due to reactions of carboxylic acids with alcohols (Surratt et al., 2006; Ziemann and Atkinson, 2012; Carey and Sundberg, 2007). Thermodynamic calculations indicate ester formation and peroxyhemiacetal formation as most likely (Barsanti and Pankow, 2006; DePalma et al., 2013) and suggest hemiacetal formation as thermodynamically unfavorable (Barsanti and Pankow, 2004; DePalma et al., 2013). Therefore, an irreversible representation of the aerosol chemistry might lead to an overprediction of the formed SOA mass, which means that can be only considered as an upper limit approach. The formed oligomers are complex compounds, which consist of a few monomer units. The oligomer equilibrium can be influenced by ambient conditions such as the temperature, relative humidity, and the chemical composition of the aerosol. A reversible representation of oligomerization reactions can be considered by means of an implemented backward reaction. E.g. Roldin et al. (2014) treats the kinetics of the reversible dimerization also with two separate reactions. However, an advanced kinetic treatment of particle-phase reactions is utilized considering monomer concentrations combined with second-order rate constants and dimer first-order degradation rates separated for bulk and surface layers. However, measurement data concerning dimerization reaction rates are scarce for condensed organic compounds and vary over several orders of magnitude (Antonovskii and Terent'ev, 1967). For the sensitivity study concerning the influence of the backward reaction on the predicted SOA mass, a simplified approach is tested. Therefore, we considered different backward reaction rate constants for particle-phase reactions (see Table 1, case 7) in addition to the pseudo-first-order rate constants.

Text added to Sect. 3.6 Limitations of the present studies:
This study utilizes a simplified scheme to consider particle-phase reactions in order to account for SOA aging. The kinetic approach of Zaveri et al. (2014) is divided into two reaction cases according to the rate of the particle-phase reactivity based on the achievement of the steady state. In Sect. 3.3, a modification of the particle-phase reactivity is presented in order to improve the representation of SOA aging under preservation of the basic classification/separation in fast and slow particle-phase reactions. This simplified approach is appropriate for application in 3-D models, treating organic compounds in lumped groups, and saves computational effort. However, for future chamber simulations with the focus on SOA processes combined with advanced measurement data, accounting for SOA aging or oxidation state, an improved representation of particle-phase reactivity will be implemented to further develop SPACCIM. Therefore, the pseudo-first-order rate constants will be replaced, e.g. by second-order equilibrium reactions under consideration of equilibrium rates provided by Barsanti and Pankow (2004) for hydrate and hemiacetal formation or thermodynamic calculations for equilibrium constants of DePalma et al. (2013) for individual dimers.

In Sect. 3.4 you describe an approach of how to estimate a weighted particle-phase bulk diffusion coefficient that considers particle composition (Eq. 10). I don't understand this approach. Is $D_m$ referring to the diffusion coefficient of the organic compounds in the particle-phase? Is it further assumed that the organic compounds are water-soluble then and that the particles are composed on one SOA+water+inorganics phase? I think that the particles often will be composed of several phases (e.g. one hygrophobic organic phase and one water+inorganics+some water soluble organics phase). I think that differences in $D_{org}$ over time can also be due to particle phase oligomerization processes that gradually increase the average organic molecular mass. But this Eq. 10 does not take this into consideration. To me Eq. 10 contains to many assumptions and is not evaluated properly to be able to be justified.

In SPACCIM no phase separation is considered. We assume that the particle contains only one mixed phase, which is characterized by the bulk diffusion coefficient. In our study, the particle phase comprises water, ammonium sulfate, and organics. In the kinetic approach of Zaveri et al. (2014), the particle-phase bulk diffusion coefficient is constant over the whole simulation time. A plenty of studies

indicate that SOA might have under certain conditions a higher viscosity (Renbaum-Wolff et al., 2013; Abramson et al., 2013; Pajunoja et al., 2014; Zhang et al., 2015; Grayson et al., 2016) and we try to consider this effect in our model studies. The mean particle-phase bulk diffusion coefficient $D_{\mathrm{m}}$, defines the diffusion coefficient of the whole particle as the mixture of the three compounds (water, ammonium sulfate, and organics). To calculate the weighted diffusion coefficient with the Vignes type rule the self-diffusion coefficient of the organic compounds $D_{\mathrm{org}}$, of the dissolved inorganic ions (here from ammonium sulfate) $D_{\mathrm{inorg}}$, and of water $D_{\mathrm{water}}$ have to be considered. The individual self-diffusion coefficients ($D_{\mathrm{org}}$, $D_{\mathrm{inorg}}$, $D_{\mathrm{water}}$) are constant over time (only the self diffusion coefficient of water depends on the temperature, but in our study the temperature is constant). However, the mole fraction of the organic compounds $x_{\mathrm{org}}$ increase with increasing organic mass in the particle phase. Due to the assumed lower self-diffusion coefficient of the organic compounds ($D_{\mathrm{org}} = 10^{-12}\,\mathrm{m^2\,s^{-1}}$ or $D_{\mathrm{org}} = 10^{-14}\,\mathrm{m^2\,s^{-1}}$) than that of water with dissolved ions ($\approx 10^{-9}\,\mathrm{m^2\,s^{-1}}$) the weighted diffusion coefficient decreases. Hosny et al. (2016) showed in their study that the viscosity increases over their experiment time and that the amount of dimers, trimers, and tetramers increases (simultaneous decrease of monomers). Therefore, the oligomerization might also decrease the particle-phase bulk diffusion coefficient. Nevertheless, there are no appropriate measurements to parameterize this effect and for the implementation of this effect in a model more assumptions than for a uniform self-diffusion coefficient for the organic compounds have to be made. Therefore, the presented sensitivity study concerning the weighted diffusion coefficient aims at investigating the effect of a decreased particle-phase bulk diffusion coefficient induced to an increased amount of organics in the particle phase on SOA formation. For this issue, two self-diffusion coefficients of the organic compounds are assumed, which are related to material more viscous than water. An assumption was made for liquid ($D_{\mathrm{org}} = 10^{-12}\,\mathrm{m^2\,s^{-1}}$) and a second for the transition to semi-solid ($D_{\mathrm{org}} = 10^{-14}\,\mathrm{m^2\,s^{-1}}$) particles. The Vignes type rule (Vignes, 1966) is used in previous studies concerning water diffusion in SOA particles (Lienhard et al., 2014, 2015; Price et al., 2015) and to study the plasticizing effect of water (O'Meara et al., 2017). To evaluate the applicability of Eq. (10) combined with the kinetic approach of Zaveri et al. (2014), additional results are added to the revised manuscript (S. O'Meara, personal communication). For this purpose, the model of Zobrist et al. (2011) have been utilized as basis for evaluation of the composition dependent particle-phase bulk diffusion coefficient. According to the reviewer's comment and the additional results, the manuscript is extended as follows:

The applicability of Eq. (10) within the kinetic approach of Zaveri et al. (2014) is checked and verified under the utilization of the model by Zobrist et al. (2011) as basis for evaluation (S. O'Meara, personal communication). Fig. S7a and S7b in the Supplement display the differences for the numerical solution from the model of Zobrist et al. (2011) and the analytical solution of the kinetic approach with the weighted particle-phase bulk diffusion coefficient for $D_{\mathrm{org}} = 10^{-11}\,\mathrm{m^2\,s^{-1}}$ and $D_{\mathrm{org}} = 10^{-13}\,\mathrm{m^2\,s^{-1}}$, respectively. For both assumed self-diffusion coefficients of the organic fraction, the numerical and analytical solution are equal within $1 \times 10^{-6}\,\mathrm{s}$ and $1 \times 10^{-4}\,\mathrm{s}$. Thus, Eq. (10) can applied to the kinetic approach instead of a constant bulk diffusion coefficient.

**Minor comments:**

Page 2, Line 17-18: I think this sentence needs to be reformulated.

We have reformulated the sentence as follows in the revised manuscript:
The modeling approach, which is mainly utilized for gas-to-particle phase partitioning of semi-volatile organic compounds, based on gas-particle equilibrium for these compounds, was proposed by Pankow (1994).

 This sentence is a bit hard to understand/follow. Can you reformulate it?

We have reformulated the sentence as follows in the revised manuscript:
Bulk viscosity measurements demonstrate that SOA particles only exist at a high relative humidity (RH > 75 %) in a liquid state (Renbaum-Wolff et al., 2013; Kidd et al., 2014). At a lower relative humidity, the organic particles exhibit a higher viscosity indicating a semi-solid or glassy phase state (Renbaum-Wolff et al., 2013; Abramson et al., 2013; Pajunoja et al., 2014; Zhang et al., 2015; Grayson et al., 2016).

 last word, remove "the"

We followed the reviewer comment and removed the "the":
However, the data by Hosny et al. (2016) fit well with particle-phase diffusion coefficient measurements by Price et al. (2015), which were converted by the Stokes-Einstein (SE) relation to viscosity.

 The molar yields are not 6 %.

Adding up of the measured gas-phase HOM yields results the following:
2.4 % + 3.4 % = 5.8 % ≈ 6 %.

 I don't understand this sentence. I understand that the diffusion coefficient will depend on the composition but not how it depends on increasing organic matter. Indirectly it can of course be influenced by the organic mass since this in turn can influence the composition.

With increasing organic matter, the mole fraction of the organic fraction increases in the particle phase. Therefore, the weighted particle-phase bulk diffusion coefficient will decrease because the self diffusion coefficient of the organic material ($D_{\mathrm{org}} = 10^{-12}\,\mathrm{m^2\,s^{-1}}$ or $10^{-14}\,\mathrm{m^2\,s^{-1}}$) is lower than that of water with dissolved ions ($\approx 10^{-9}\,\mathrm{m^2\,s^{-1}}$). According to the reviewer's comment the description has been improved in the introduction:

The particle-phase bulk diffusion coefficient might be composition dependent and due to a lower self-diffusion coefficient of the organic material, increasing organic matter in the particle phase decreases the weighted particle-phase bulk diffusion coefficient.

Sect. 2.1. Here you describe the experiments and the instruments that were used. What I am missing is the information about the size of the smog chamber (volume) and the wall material (e.g. Teflon) and I also miss information about how the measurement results are used in the present manuscript. E.g. how were the measurements used to derive the SOA mass in Fig. 9a? Also I miss a discussion concerning VOC and particle wall losses that is known to be important in chambers.

Detailed information concerning the chamber are provided in Iinuma et al. (2009) and Mutzel et al. (2016). Briefly, LEAK is a cylindrical, $19\,\mathrm{m^3}$ Teflon bag with a surface-to-volume ratio of $2\,\mathrm{m^{-1}}$. This information is now also given in the revised manuscript.
The measurement results are used in Sect. 3.2 to evaluate the simulations. Fig. 8 and Fig. 9 comprises the SOA mass measured in LEAK. As described in Gatzsche et al. (2017), the volume size distribution was measured with a scanning mobility particle sizer (SMPS), which is converted with an average density of $1\,\mathrm{g\,cm^{-3}}$ into the increase of organic mass (Mutzel et al., 2016).

Wall loss effects are not considered in the presented paper, due to the difficulties associating with modeling the size resolved wall loss for mixed organic and inorganic particles.

According to the reviewer's comment, the description of the LEAK chamber has been improved (Sect. 2.1) and the neglected wall losses have been mentioned (Sect. 3.5):

Section 2.1:
A detailed description of the LEAK chamber together with the available equipment can be found in Iinuma et al. (2009) and Mutzel et al. (2016). Briefly, LEAK is a cylindrical, $19\,m^3$ Teflon bag with a surface-to-volume ratio of $2\,m^{-1}$.

Section 3.5:
Wall loss effects are not considered in this study, due to the short experiment time (2 h). However, for longer experiment duration particle and gas wall loss might be an important process for chamber studies and have to be considered in modeling (Zhang et al., 2014).

Sect. 3.5. Why did you decide to use $k_c = 10^{-2}\,s^{-1}$ and $k_c = 10^{-3}\,s^{-1}$ and $D_{org} = 10^{-12}\,m^2\,s^{-1}$ and $D_{org} = 10^{-14}\,m^2\,s^{-1}$, respectively?

The authors decide to use $D_{org} = 10^{-12}\,m^2\,s^{-1}$ and $D_{org} = 10^{-14}\,m^2\,s^{-1}$ because previous studies (Koop et al., 2011; Renbaum-Wolff et al., 2013; Zhang et al., 2015; Grayson et al., 2016; Hosny et al., 2016) indicate that SOA particles have a higher viscosity than water droplets. The particle-phase reactivity is subject of the sensitivity studies and results from $k_c = 10^{-2}\,s^{-1}$ and $k_c = 10^{-3}\,s^{-1}$ have been chosen as reference simulations within this section.

Table S4. I am missing a unit on the pure liquid saturation vapor pressures of the HOM. I also think that you should mention that these molecules only represent the HOMs but that the HOMs is a family of many organic molecules presumably formed from autoxidation with a wide range of volatility. Can you justify why you decided to use these specific HOM molecules and specify how they are assumed to be formed? It is not either clear how these three different HOM are used in the model because in the gas-phase mechanism you only seem to have one HOM molecule.

The missing unit in Table S4 is atm. The table has been revised. These HOM molecules have been utilized because their structures have been presented by Berndt et al. (2016) and these compounds have been identified from laboratory measurements. The reaction pathway for the formation of HOMs is already described in Berndt et al. (2016). The average vapor pressure of these three compounds is utilized as an approximation for the vapor pressure in the gas-phase chemistry mechanism. Therefore, no individual products are represented in the model. The manuscript description has been extended in Sect. 3.2 as follows:

The HOM compounds are lumped to one compound group and added to the gas-phase chemistry mechanism (see Table S1, reactions 1b and 3b). As an estimate for the vapor pressure of this compound group, the average vapor pressure of the compounds listed in Table S4 from COSMO-RS have been taken.

Eq. 1. Is $A_i$ representing a concentration of a species or is $A_i$ a solute as stated on Line 17? In Eq. 2 $C$ is used to represent concentrations.

Considering the reviewer's comment, the description in Sect. 2.2.2 has been changed as follows:

Thereby, the utilized parameters are the particle-phase concentration $A_i$ of the solute $i$ as a function of the radius $r$ and the time $t$, the particle-phase bulk diffusion coefficient of the solute $D_{b,i}$, and the chemical reaction rate constant $k_{c,i}$ of the solute within the particle phase.

The title is a bit imprecise. Their exist many model studies of SOA formation from $\alpha$-pinene ozonolysis. Can you make it more precise?

The title is specified:

Kinetic modeling studies of SOA formation from $\alpha$-pinene ozonolysis

**Referee #2**
**General comments**
My biggest concern is how the authors justify their assumption that organic molecules must diffuse into the particle phase in order to contribute to particle growth. This assumption is crucial for all conclusions in this paper and finds too little scrutiny.

In general, the utilized kinetic approach supplies only two options for the organic compounds in the particle phase to change the equilibrium between the gas and the particle phase:

1. The organic molecules diffuse into the particle, thus at the particle surface the equilibrium between the gas and the particle phase is changed.

2. The organic molecules react in the particle phase to the corresponding r-product, which is treated as non-volatile and, therefore, the equilibrium between the gas and the particle phase is altered.

In Fig. 2a, the influence of the particle-phase bulk diffusion coefficient $D_b$ and pseudo-first-order rate constant $k_c$ on the steady state timescale is shown. Therewith, it can be shown that two different regimes exist for the sensitivity of the kinetic approach. For particle-phase bulk diffusion coefficients $D_b \geq 10^{-15}\,\mathrm{m^2\,s^{-1}}$, the quasi-steady-state timescale is mainly sensitive to the value of the particle-phase bulk diffusion coefficient. In contrast, for particle-phase bulk diffusion coefficients $D_b < 10^{-15}\,\mathrm{m^2\,s^{-1}}$ the quasi-steady-state timescale predominantly depends on the pseudo-first-order rate constant. Therefore, the particle-phase bulk diffusion coefficient and the pseudo-first-order rate constant strongly effect the SOA formation within this approach. To improve the explanation of this context in the paper, we revised the text as follows:

Modified/Added explanation to Sect. 3.1.1:
For values $D_b < 10^{-16}\,\mathrm{m^2\,s^{-1}}$ the SOA formation is inhibited because the condensed organic material does not diffuse sufficiently into the particle bulk. According to the classification of Mai et al. (2015), this case is named particle-phase-diffusion-limited partitioning. After the formation of a thin organic shell/film around the particle, no effective SOA formation takes place because of the long mixing time inside the particle. Thus, the gas-phase concentrations of the condensing organic compounds as well as the interfacial transport of these compounds are not the limiting factors of SOA formation under these conditions.

In Sect. 2.2.2., which parts of these equations are necessary? Entrainment and outflow are not considered in this study and there is an argument for omitting it from the equations here. The authors have to add what $M$ stands for here, is it the number of size-section? I see a term for partitioning

between aqueous and organic phases, how is this utilized in this manuscript? The manuscript never mentions phase-separation, so how does this play a role? From reading the manuscript, it is also not clear how the mass-transfer term works and when it is needed.

First of all, $M$ is the number of size-sections. We included this in Sect. 2.2.2. of the revised version of the manuscript as follows:

Therein, $\overline{C}_{a,i,m}$ denotes the total average concentration of a solute $i$ in size-section $m$, with $M$ the number of size-sections.

Equation (6) and Eq. (8) summarizes the general model equations in the spectral model SPACCIM and offers the possibility to compare with previous publications using SPACCIM (Wolke et al., 2005; Rusumdar et al., 2016). The entrainment and outflow terms are included in both equations for the sake of completeness. As already indicated in the response to referee #1, the parcel model is not exclusively designed for simulation of aerosol chamber studies and additional text is provided in the revised manuscript.

At the current state, we treat the particle phase as one mixed phase without consideration of phase separation. Therefore, no partitioning term between the aqueous and the particle phase is considered. The aqueous phase transfer term (term II in Eq. (6) and Eq. (8)) describes the phase transfer between the gas and the aqueous phase with the Schwartz approach (Schwartz, 1986). According to the reviewer's comment, we changed it for the sake of clarity "aqueous phase transfer" to "gas-to-aqueous-phase mass transfer" in Eq. (6) and Eq. (8) in the revised manuscript.

The mass transfer term is already shortly described in the manuscript and a more detailed description is given in Wolke et al. (2005), which is referenced in the presented paper.

Regarding the sensitivity studies (Sect. 3.1), the authors must do a better job in highlighting that some of these simulations are probably far from reality. For example, Fig. 4c interestingly shows that omitting HOMs in the mechanism leads to an increase in SOA mass. However, the SOA mass yield is so low in these simulations that I strongly doubt their usefulness for real applications. The same argument can be made for Figs. 2b and 3b. These simulations show very different reaction regimes that might not be encountered in a simulation chamber experiment. While I find it interesting to show how a system reacts under strong perturbation, the questionable practicality must be indicated more clearly in the text.

The authors thank the reviewer for the interesting assessment concerning the simulation results with very low particle-phase bulk diffusion coefficients. Figs. 2b, 3b, and 4c display results for $D_b = 10^{-18}\,\mathrm{m^2\,s^{-1}}$, which indicates higher viscous particles in the upper range of the semi-solid phase state. We included the figures in our paper to show that the low particle-phase bulk diffusion coefficient will impede SOA formation regardless of the settings of the remaining model parameters within this approach. This is contrary to the diffusion coefficients calculated from viscosity measurements indicating a semi-solid phase state of the SOA particles for a broad humidity range (Renbaum-Wolff et al., 2013; Zhang et al., 2015; Grayson et al., 2016). As the reviewer mentioned, the formed SOA mass for this low diffusion coefficients is not useful for real application in simulating chamber studies or for regional model studies with effective SOA formation. Therefore, we determined the particle-phase bulk diffusion coefficient as key parameter in our approach. According to the reviewer's comment, the text in the manuscript has been revised at different places highlighting that several simulations reflect extreme environmental conditions not prevalent in the simulated aerosol chamber study.

Modifications and additions to Sect. 3.1.2/Fig. 2b are given in the next paragraph.

Sect. 3.1.3/Fig. 3b
Therefore, the influence of the particle radius is higher for semi-solid particles. However, the formed SOA mass is rather small due to the limited diffusion into the particle and it should be kept in mind that this diffusion coefficient is limited to dry conditions.

Sect. 3.2/Fig. 4c
As indicated before, semi-solid aerosol particles might occur for a limited range of atmospheric conditions when the relative humidity decreases below 40 %. Therefore, the HOMs might affect SOA formation as seen in Fig. 4a and 4b, regardless of their reactivity in the particle phase.

In Sect. 3.1.2, I find it imperative to describe what the reason is that there is no particle growth at low diffusivities. I would suggest expanding on the description about what happens in the model, in words or figures.

According to the reviewer's comment, the explanation for the semi-solid particles is extended in Sect. 3.1.2 as follows:

For semi-solid particles, the kinetic approach is predominantly sensitive for particle-phase reactions as indicated in Zaveri et al. (2014). The particle-phase-diffusion-limited partitioning for the viscous aerosol particles has been step-wise raised due to the increased particle-phase reaction rate constant (see Fig. 2b). The mass enhancement factor is about 2.5 for a seven orders of magnitude higher reactivity. It is noted that organic aerosol particles might exhibit such low particle-phase diffusivities $D_b \leq 10^{-18}\,\mathrm{m^2\,s^{-1}}$ mainly under dry conditions (RH < 40 %).

Section 3.3
I am not sure if I understand the purpose of Sect. 3.3. This section essentially looks at the effect of different equilibrium constants of the particle phase reaction, but it is presented as effect of different first-order reaction rates. I tend to think that an equilibrium constant is more straightforward here as the particle will most likely be in reactive equilibrium anyway. I find Fig. 5 very instructive and what happens in Fig. 6 is just that the share of r-products (orange bands) is reduced, is that correct? This is not a very exciting result given that all reaction rates are arbitrary guesses, or are the amounts of p-products and HOMs also affected? A normalized sensitivity coefficient of the mass of r-products and p-products to a reaction rate might be much more instructive. On a different note, would it be possible to connect these cases to real scenarios, e.g. by making more realistic assumptions of $k_c$ and $k_b$ for examples at high RH low RH?

Sect. 3.3. aims at the description of reversible particle-phase reactions. In the approach of Zaveri et al. (2014), only reactions in the particle phase are considered, which increase the particle mass due to the non-reversible reaction from "partitioning-products" (p-products) to "reacted-products" (r-products). In Sect. 3.1.2, it has been shown that the pseudo-first-order rate constant can be interpreted as an additional reactive uptake in the particle phase. An important detail of the kinetic approach of Zaveri et al. (2014), which might get not enough emphasis in the paper, is the distinction between two different model cases on the basis of the pseudo-first-order rate constant. For fast reactions ($k_c \geq 10^{-2}\,\mathrm{s^{-1}}$), Eq. (2) is valid and for slow reactions ($k_c < 10^{-2}\,\mathrm{s^{-1}}$), Eq. (3) is applied. The first difference between the two equations is the missing $Q_i$ in Eq. (3), which is the steady state term given in the Appendix A in the presented paper. As indicated in Fig. 2b, the quasi-steady-state term $Q$ attains the value of 1 ($Q \to 1$) for the limiting case ($k_{c,i} \to 0$, $q \to 0$). The second difference is that due to the usage of the two-film theory for slow reactions an overall mass transfer coefficient

$K_{g,i}$ is defined (see Eq. (A10) in the presented paper and Zaveri et al., 2014):

$$\frac{1}{K_{g,i}} = \frac{1}{k_{g,i}} + \frac{1}{k_{p,i}} \left( \frac{C^*_{g,i}}{\sum_j \overline{A}_j} \right). \tag{6}$$

Therefore, the choice of the pseudo-first-order rate constant influences, which regime is applied and, therefore, the partitioning equilibrium of the organic compounds. Due to the implementation of additional backward reactions, the approach of the two reaction regimes is preserved and the reaction system can be written as follows:

$$\text{p-product}_i \xrightarrow{k_c} \text{r-product}_i, \tag{7}$$

$$\text{r-product}_i \xrightarrow{k_b} \text{p-product}_i, \tag{8}$$

$$\text{p-product}_i \underset{k_b}{\overset{k_c}{\rightleftharpoons}} \text{r-product}_i. \tag{9}$$

The aging of the p-products is described by Eq. (7), the backward reaction is formulated in Eq. (8), and the net particle-phase reaction results in Eq. (9). The advantage of such a separate treatment of the production and reduction reaction of an equilibrium reaction is that the equilibrium state is adjusted over the time and is not instantaneously reached. A separate treatment of the dimerization reaction and the degradation of the dimers is also presented in Roldin et al. (2014) and an equilibrium of both reactions is reached after a distinct time. Nevertheless, the temporal evolution of dimers and monomers are solved with an own kinetic model in Roldin et al. (2014), whereby surface and bulk layers are separately handled. This approach demands a quite high computational effort inadequately for the subsequent application in a 3-D model. Further, measurements of particle-phase reactivity are scarce and the values for peroxyhemiactelas depend on acidity ranging from $2.3 \times 10^{-25}$ to $3.2 \times 10^{-23}$ molecules$^{-1}$ cm$^3$ s$^{-1}$ (Ziemann and Atkinson, 2012). As already mentioned above, the description of the particle phase reactivity modification in Sect. 3.3 is improved and the limitations of that approach are provided in an additional paragraph of Sect. 3.6.

Sect. 3.4
In this section, it is very hard to follow what exact calculations were performed. This leads to many open questions: What are the initial conditions for these experiments, especially $x_{inorg}$ and $x_{water}$? What were the diffusivities $D_{inorg}$ and $D_{water}$? On which humidity is the water concentration based? What is the hygroscopicity of these particles, how is it determined? Why is it safe to assume that organic and inorganic phases are mixed? Are we seeing an effect of $D_{inorg}$ or an effect of $D_{water}$ here? It would be interesting to see $D_m$ plotted against humidity, following this framework. You could compare the humidity-dependence of your $D_m$ for self-diffusion in SOA to the values of tracer diffusion determined in Berkemeier et al. (2014); Lienhard et al. (2015); Price et al. (2015) or Berkemeier et al. (2016).
It would be helpful if it were more explicitly explained what happens in the simulations. What does the time profile of $D_m$ look like? I suppose it increases due to the smaller contribution of inorganics. On, p. 17, l. 33, the wording seems suboptimal. You write that using a constant bulk diffusion coefficient lowers the total SOA mass. Along the lines of the argumentation, this should be formulated the other way around: implementing a weighted particle-phase diffusion scheme increases total SOA mass. I have to wonder though, what of this effect is due to water and what is due to inorganics? It seems very clear to me that you would need to compare the diffusivity of the equilibrium SOA-water mixture and only add in the inorganics. Otherwise, you mainly make the argument that humidity leads to more SOA mass and not so much investigate the effects of a time-dependent diffusivity coefficient.

The simulation is initialized with inorganic seed particles (ammonium sulfate) with a particle radius of

$r_p = 35$ nm. A relative humidity of 55 % leads in combination with the dissolved ammonium sulfate to the following mole fractions $x_{inorg} = 0.43$ and $x_{water} = 0.57$. The hygroscopicity is treated with the Köhler equation as described in Pruppacher and Klett (2010). For the self-diffusion coefficient of water, we utilize the relation proposed by Holz et al. (2000):

$$D_{water} = D_0[(T/T_S) - 1]^\gamma, \tag{10}$$

with:

$$D_0 = (1.635 \times 10^{-8} \pm 2.242 \times 10^{-11})\,\mathrm{m^2\,s^{-1}}, \tag{11}$$
$$T_S = (215.05 \pm 1.20)\,\mathrm{K}, \tag{12}$$
$$\gamma = 2.063 \pm 0.051. \tag{13}$$

This relation is valid between 0 and 100 °C and comprises an error limit of $\leq 1$ %. The self diffusion coefficients of dissolved ions are tabulated in Cussler (2009), $D_{NH_4^+} = 1.96 \times 10^{-9}\,\mathrm{m^2\,s^{-1}}$ and $D_{SO_4^{2-}} = 1.06 \times 10^{-9}\,\mathrm{m^2\,s^{-1}}$. For seed particles containing water with dissolved ammonium sulfate ions, an initial weighted particle-phase bulk diffusion coefficient of $D_m = 1.78 \times 10^{-9}\,\mathrm{m^2\,s^{-1}}$ is derived.

In the current SPACCIM version, no liquid-liquid phase separation of mixed organic-inorganic particles is considered to describe the coexistence of separate aqueous and organic phases. Therefore, this bulk approach is utilized to describe the particle as one mixed phase. To consider the additional effects of water and inorganics on partitioning, the simplified method of Zuend et al. (2010) might be implemented, which is also applicable for regional modeling.

Considering Eq. (10) of the presented paper, an effect of $D_{org}$ can be seen because of the increase of $x_{org}$ during the simulation.

The focus of the publications mentioned by the reviewer might not the same as that of the presented study. In the presented paper, a weighting rule for the three self-diffusion coefficients was applied to calculate a weighted particle-phase bulk diffusion coefficient $D_m$ that replicates the effect of the increasing organic fraction in particles due to SOA formation. For the chamber experiments, the relative humidity was constant (55 %), which was also considered in the model and thus, the LWC is constant. To illustrate the temporal evolution of the weighted particle-phase bulk diffusion coefficient $D_m$ depending on the SOA mass, an additional figure is provided in the revised manuscript (see Fig. 4). The authors decided to relate this additional figure to Sect. 3.5, where the weighted particle-phase bulk diffusion coefficient is applied to the chamber studies. The weighted particle-phase bulk diffusion coefficient $D_m$ decreases because more organic material (characterized by a smaller self diffusion coefficient) is consumed (see Fig. 4). Therefore, an increase in organic mass results in a decrease of the weighted particle-phase bulk diffusion coefficient $D_m$.

According to the reviewer's comment, the description of the calculation of the weighted diffusion coefficient is expanded in the Supplement as follows and Fig. 4 as well as the interpretation of the new figure are added in Sect. 3.5:

Text added to Supplement:
All simulations are initialized with inorganic seed particles containing water and dissolved ammonium sulfate, with a particle radius of $r_p = 35$ nm. A relative humidity of 55 % leads in combination with the dissolved ammonium sulfate to the following mole fractions $x_{inorg} = 0.43$ and $x_{water} = 0.57$. For the self-diffusion coefficient of water, we utilize the relation proposed by Holz et al. (2000):

$$D_{water} = D_0[(T/T_S) - 1]^\gamma, \tag{14}$$

with:

$$D_0 = (1.635 \times 10^{-8} \pm 2.242 \times 10^{-11})\,\mathrm{m^2\,s^{-1}}, \tag{15}$$
$$T_S = (215.05 \pm 1.20)\,\mathrm{K}, \tag{16}$$
$$\gamma = 2.063 \pm 0.051. \tag{17}$$

[Figure]

Figure 4: Simulated SOA mass as shown in Fig. 8b of Gatzsche et al. (2017) for $k_c = 10^{-2}\,\mathrm{s^{-1}}$, $k_b = 10^{-2}\,\mathrm{s^{-1}}$, under consideration of HOMs, and with the weighted diffusion coefficient utilizing $D_{\mathrm{org}} = 10^{-14}\,\mathrm{m^2\,s^{-1}}$ in comparison to the measured SOA mass from the LEAK experiment. The corresponding weighted particle-phase bulk diffusion coefficient $D_{\mathrm{m}}$ from the simulation is displayed on the secondary y-axis.

The self diffusion coefficients of dissolved ions are tabulated in Cussler (2009), $D_{\mathrm{NH_4^+}} = 1.96 \times 10^{-9}\,\mathrm{m^2\,s^{-1}}$ and $D_{\mathrm{SO_4^{2-}}} = 1.06 \times 10^{-9}\,\mathrm{m^2\,s^{-1}}$. For the aqueous ammonium sulfate seed particles an initial weighted particle-phase bulk diffusion coefficient of $D_{\mathrm{m}} = 1.78 \times 10^{-9}\,\mathrm{m^2\,s^{-1}}$ is derived. Due to the partitioning of organic compounds, the organic mole fraction $x_{\mathrm{org}}$ increases and $D_{\mathrm{org}}$ influences $D_{\mathrm{m}}$. Since $D_{\mathrm{org}}$ is estimated to $D_{\mathrm{org}} = 10^{-12}\,\mathrm{m^2\,s^{-1}}$ or $D_{\mathrm{org}} = 10^{-14}\,\mathrm{m^2\,s^{-1}}$, the increase of the organic mole fraction causes a decrease of the weighted particle-phase bulk diffusion coefficient.

Sect. 3.5:
The high initial particle-phase bulk diffusion coefficient, $D_{\mathrm{m}} \approx 2 \times 10^{-9}\,\mathrm{m^2\,s^{-1}}$ (see Fig. 9), for the aqueous ammonium sulfate seed particles enables a fast diffusion in the aerosol particles. Thus, immediately partitioning HOMs can be absorbed quickly into the particle phase. Within the first 30 min of the simulation time, the SOA mass sharply increases and the weighted particle-phase bulk diffusion coefficient drops about three orders of magnitude. Consequently, the mixing time in the particle phase increases and this leads to a slower SOA mass formation. This process is depicted in Fig. 9, where for the first hour of simulation time the major changes in the weighted particle-phase bulk diffusion coefficient and the SOA mass can be seen. After the weighted particle-phase bulk diffusion coefficient has reduced to a value $D_{\mathrm{m}} \approx 10^{-13}\,\mathrm{m^2\,s^{-1}}$, the longer mixing time will cause a slower SOA formation as already shown in Fig. 1b. This effect is further pronounced due to continuous SOA formation and a concomitant decrease in particle-phase diffusion.

We agree with the reviewer's comment that the wording on, p. 17, l. 33 of the presented paper is suboptimal to describe the comparison between a constant and a weighted particle phase bulk diffusion coefficient and have thus reformulated this sentence as follows:

Fig. 7c reveals that the total SOA mass is increased by about $40-50\,\%$ for a weighted particle-phase bulk diffusion coefficient ($D_{\mathrm{org}} = 10^{-14}\,\mathrm{m^2\,s^{-1}}$) compared to simulation results with a constant

particle-phase bulk diffusion coefficient of $D_\text{b} = 10^{-14}\,\text{m}^2\,\text{s}^{-1}$. Additionally, the SOA formation is faster for the weighted particle-phase bulk diffusion coefficient.

**Specific comments**

p. 12, l. $27-28$

*"No initial organic mass $OM_0$ is utilized for the simulations with HOMs."*
- This change seems arbitrary when first reading the article and gets only clear upon further reading. I wonder how the simulations would look like if no initial organic mass would be utilized for the simulations without HOMs? I find that would be much more instructive.

The partitioning approach requires initial organic mass in the aerosol phase ($OM_0$) according to Eq. (A7) in the presented manuscript. Therefore, a sufficient amount of organic mass is initialized to enable partitioning. For the simulations with HOMs, $OM_0 \approx 2 \times 10^{-13}\,\text{g}\,\text{m}^{-3} = 2 \times 10^{-4}\,\text{ng}\,\text{m}^{-3}$ have been utilized, which is negligible. For the simulations without HOMs, $OM_0 = 41.62\,\text{ng}\,\text{m}^{-3}$ have been initialized. When we initialize the simulations with considered HOMs with $OM_0 = 41.62\,\text{ng}\,\text{m}^{-3}$, the rapid partitioning of HOMs quite fast exceeds the effect of the initial particulate organic mass and, therefore, we decided to initialize these simulations with the negligible amount of organic mass. According to the reviewer's comment, the information of the negligible particulate organic mass concentration is provided in Table 1 for case 6a and 6b and the indicated sentence is slightly modified in Sect. 3.2:

Almost no initial organic mass $OM_0$ is utilized for the simulations with HOMs.

p. 13, l. 2
*"A rapid condensation of HOMs occurs due to their low vapor pressures (Fig. 4a)."*
- How can this be seen in Fig. 4a? It seems not easy to see whether the solid lines take off sooner than the dashed lines. Is also does not appear as if the solid lines separate from the dashed lines within the early moments of the experiment/simulation, but rather over the first half of the experiment. Maybe referring to Fig. 5 would be helpful here if it is showing what you mean here?

We agree with the reviewer's comment that the fast condensation of the HOMs can not be seen so easily in (Fig. 4a). Therefore, we add Fig. 5 to the Supplement, which displays only the first half hour of the simulation time. Fig. 5 shows that for the first 13 minutes, the dashed red lines lay above the solid lines. However for the simulation with HOMs and $k_\text{c} = 1\,\text{s}^{-1}$ (see Fig. 5, indicated by the solid red line), the condensation of organic material begins at about 4 minutes. The SOA mass increases so fast that until 13 minutes is higher than for the simulation without HOMs. This behavior is described as a very rapid condensation.

p. 13, Fig. 3 - Since particle radius is reduced down to $11\,\text{nm}$, are Kelvin effects considered in this study? I don't see this mentioned in the manuscript.

Until now we have not considered the Kelvin effect because the correction of the vapor pressure due to the Kelvin term (see Eq. 2) might be smaller than the indicated error range (O'Meara et al., 2014; Kurtén et al., 2016) of the group contribution methods applied for the estimation of the vapor pressures of the partitioning compounds. A longer explanation for this assumption is already given in the response for reviewer #1 and we have added this information to a new section as indicated earlier.

[Figure]

Figure 5: Simulated SOA mass including HOMs and additional variation of the pseudo-first-order rate constant of particle reactions $k_c$ (case 6a and 6b of Table 1) for liquid ($D_b = 10^{-12}\,\mathrm{m^2\,s^{-1}}$) aerosol particles as shown in Fig. 4a, but only for the first half hour of the simulation time.

p. 14, l. 5−7

*"This may be the reason for the convergence of formed SOA mass for $k_c = 1\,s^{-1}$ combined with $D_b = 10^{-14}\,m^2\,s^{-1}$ and the overall more effective SOA formation without consideration of HOMs for semisolid particles ($D_b = 10^{-18}\,m^2\,s^{-1}$, Fig. 4c)."*
- Would it be possible to give more explanation on this odd result of Fig. 4c? Should this be left out if not realistic?

Following the reviewer's comment, we modified the corresponding text and expanded the explanation to Fig. 4c as follows:

This may be the reason for the convergence of formed SOA mass for $k_c = 1\,\mathrm{s^{-1}}$ combined with $D_b = 10^{-14}\,\mathrm{m^2\,s^{-1}}$. For semi-solid particles ($D_b = 10^{-18}\,\mathrm{m^2\,s^{-1}}$, Fig. 4c), the SOA formation without consideration of HOMs is more effective. This circumstance is caused by the missing particle-phase reactions for HOMs because the utilized kinetic approach is for $D_b < 10^{-15}\,\mathrm{m^2\,s^{-1}}$ mainly sensitive to particle-phase reactions (Zaveri et al., 2014). Further, for semi-solid particles a particle-phase-diffusion-limited partitioning of HOMs occur with proceeding simulation time (Mai et al., 2015). Therefore, the equilibrium between the gas and the particle phase is quickly established for HOMs and their effect on the SOA mass differ from that for liquid particles.

p. 14, l. 10

*"In general the HOMs provide about 27 % of the simulated final total SOA mass and introduce SOA mass formation."*
- How does this compare to the molar yields in Berndt et al. (2016)? In addition, there is a comma missing after "in general".

The molar yields of Berndt et al. (2016) are determined in the gas-phase and in our study the 27 % are contained in the particle phase. The study of Berndt et al. (2016) aims not in measuring partitioning or particle-phase concentrations. Following the reviewer's comment, we added the comma after "in general" and changed "introduce" to "initiate":

"In general, the HOMs provide about 27 % of the simulated final total SOA mass and initiate SOA

mass formation."

p. 15, l. 8f
*"The main benefit of the implementation of a sufficiently fast backward reaction ($k_b \leq 10^{-2}\,s^{-1}$) is the asymptotic curve shape of the SOA mass for proceeding simulation times. This behavior is also observed during chamber studies, which indicate an equilibrium state of the gas and the particle phase after a proceeding oxidation time."*
- This reads interesting, but is difficult to understand without practical experience with smog chambers. Could this statement be explained in more detail and justified with examples?

This behavior is for example depicted in Fig. 1 of Ng et al. (2006) for $\alpha$-pinene ozonolysis. After $\alpha$-pinene is totally consumed, the maximum of organic mass is reached and remains constant thereafter. According to the reviewer's comment we have added this reference in the regarding sentence as follows:

The main benefit of the implementation of a sufficiently fast backward reaction ($k_b \leq 10^{-2}\,s^{-1}$) is the asymptotic curve shape of the SOA mass for proceeding simulation times. This behavior is also observed during chamber studies (Ng et al., 2006), which indicate an equilibrium state of the gas and the particle phase after a proceeding oxidation time and concomitant consumption of the hydrocarbon.

p. 18, l. 25
*"Consideration of the weighted particle-phase bulk diffusion coefficient and HOMs lead to a faster SOA mass increase at the beginning of the simulation. The decreasing particle-phase bulk diffusion coefficient due to the uptake of further organic material and the backward reactions in the particle phase induce a flattening of the mass increase."*
- Where can this be seen? Why should a weighted particle-phase diffusion coefficient generally speed up SOA formation?

This result can be seen in Fig. 8a in the period between $0.25 - 1.5\,h$ of the simulation time. We did not generalize this result and for clarity we insert after "the beginning of the simulation": *"when the organic amount is low in the particle phase"*.

Consideration of the weighted particle-phase bulk diffusion coefficient and HOMs lead to a faster SOA mass increase at the beginning of the simulation when the organic amount is low in the particle phase.

p. 20, Figure 8
- In Fig. 8, it is very difficult to understand the simulation conditions of each plotted line. Am I correct that the dashed lines are showing the same results in both panels? This would be worthwhile pointing out! It would be easier to see if both panels would show the same range on the y-axis. It is also difficult to spot the line that fits the experimental data well in Fig. 8b, maybe draw these on top of the markers or highlight it in another way.

In the caption of Fig. 8 the following sentence refers to the two reference cases indicated by the dashed lines: *"For comparison, in both plots the results for a constant particle-phase bulk diffusion coefficient $D_b = 10^{-12}\,m^2\,s^{-1}$ combined with two different pseudo-first-order rate constants of particle reactions $k_c = 10^{-2}\,s^{-1}$ (ref. case I) and $k_c = 10^{-3}\,s^{-1}$ (ref. case II) are included (dashed lines)."*. As also indicated by the key, the two reference cases are the same for both figures and this is additionally

mentioned in the text of the paper: p. 18, l. 18/19.

For an improved depiction, we adjusted similar y-ranges for Fig. 8a and Fig. 8b and we increased the line width of the LEAK measurements according to the reviewer's comment (see Fig. 6).

[Figure]

Figure 6: Simulated SOA mass considering an effective particle-phase bulk diffusion coefficient as well as HOMs, a constant pseudo-first-order rate constant of particle reactions $k_c = 10^{-2}\,s^{-1}$ and under additional variation of the chemical backward reaction rate constant of particle reactions $k_b$ in comparison with aerosol chamber measurements from LEAK. a) Organic material is considered with a diffusion coefficient of $D_{org} = 10^{-12}\,m^2\,s^{-1}$ in the weighted diffusivity $D_m$ (solid lines); b) Organic material is considered with a diffusion coefficient of $D_{org} = 10^{-14}\,m^2\,s^{-1}$ in the weighted diffusivity $D_m$ (solid lines). For comparison, in both plots the results for a constant particle-phase bulk diffusion coefficient $D_b = 10^{-12}\,m^2\,s^{-1}$ combined with two different pseudo-first-order rate constants of particle reactions $k_c = 10^{-2}\,s^{-1}$ (ref. case I) and $k_c = 10^{-3}\,s^{-1}$ (ref. case II) are included (dashed lines).

p. 20, l. 4ff
*"For the preferred model setup of Fig. 8b with $k_b = 10^{-2}\,s^{-1}$, the simulation is in a very good agreement with the measured concentration decrease of $\alpha$-pinene (Fig. 10a)."*
- Why does the $\alpha$-pinene concentration depend on $k_b$?

The $\alpha$-pinene concentration depends not on the value of $k_b$. We only would like to indicate with this wording that the corresponding gas-phase concentrations to this model case are displayed. According to the reviewer's comment, we improved the beginning of the paragraph describing the gas-phase concentration results as follows:

Corresponding gas-phase concentrations for the preferred model setup of Fig. 8b, with $k_b = 10^{-2}\,s^{-1}$, of the reactants $\alpha$-pinene and ozone (see Fig. 10a and 10b) as well as a first reaction product named pinonaldehyde (Fig. 10c) have also been compared with smog chamber measurements. The simulation for the $\alpha$-pinene depletion is in a very good agreement with the measured concentration decrease of $\alpha$-pinene (Fig. 10a).

p. 21, l. 7
*The depletion of ozone is slightly overestimated by the model after 1.5 hours (Fig. 10b). The measured gas-phase concentration of pinonaldehyde is underestimated by the model (Fig. 10c)."*
- Do you have ideas what could be the underlying reasons here? This would add much more to the manuscript than just plotting results.

The formation of pinonaldehyde was measured by a proton-transfer-reaction mass spectrometer (PTR-

MS) at $m/z$ 169 ($[M+H]^+$). The PTR-MS technique enables only the detection of the $m/z$ ratio. No further information were obtained. Thus, compounds or fragments with the same $m/z$ were detected as well resulting in an overestimation of the pinonaldehyde concentration measured by PTR-MS. This circumstance can cause the underestimation of the gas-phase concentration of pinonaldehyde (see Fig. 10c of the presented paper).

Ozone is measured with an ozone monitor (49c Ozone Analyzer, Thermo Scientific, USA) and this device is based on measuring absorption on characteristic wavelengths. For measuring ozone, the absorption at 254 nm is utilized. According to Docherty et al. (2005), the ozonolysis of $\alpha$-pinene yields up to 47 % organic peroxides. As organic peroxides absorb light at 254 nm, an overestimation of the signal detected by the ozone monitor caused by the high amount of organic peroxides cannot be excluded. Therefore, with the increase of the hydroperoxide concentration over the experiment time the overestimation of the ozone concentration by the monitoring system might increase and the underestimation by the model can be caused. According to the reviewer's suggestion, we included this information in the interpretation of the results of Sect. 3.5 as follows:

The depletion of ozone is slightly overestimated by the model (see Fig. 10b). After half an hour, the measured ozone concentration decreases not so fast as initially started. Experimentally, ozone is measured with an ozone monitor (49c Ozone Analyzer, Thermo Scientific, USA) and this device is based on measuring absorption on characteristic wavelengths. For measuring ozone, the absorption at 254 nm is utilized. According to Docherty et al. (2005), the ozonolysis of $\alpha$-pinene yields up to 47 % organic peroxides. As organic peroxides absorb light at 254 nm, an overestimation of the signal detected by the ozone monitor caused by the high amount of organic peroxides cannot be excluded. Therefore, with the increase of the hydroperoxide concentration over the experiment time the overestimation of the ozone concentration by the monitoring system might increase and the underestimation by the model can be caused. Further, the measured gas-phase concentration of pinonaldehyde is underestimated by the model (Fig. 10c). This cannot be caused by an excessive partitioning of pinonaldehyde into the particle phase because pinonaldehyde is characterized by a high saturation vapor pressure and there is no effective partitioning into the particle phase. However, the formation of pinonaldehyde is measured by a proton-transfer-reaction mass spectrometer (PTR-MS) at $m/z$ 169 ($[M + H]^+$). The PTR-MS technique enables only the detection of the $m/z$ ratio. No further information were obtained. Thus, compounds or fragments with the same $m/z$ were detected as well resulting in an overestimation of the pinonaldehyde concentration measured by PTR-MS. This circumstance can cause the underestimation of the gas-phase concentration of pinonaldehyde (see Fig. 10c). To investigate the underestimation of pinonaldehyde concentration additionally from the site of the model, we evaluated the branching ratios of the $\alpha$-pinene with ozone reaction. Based on the results of Berndt et al. (2003), the pinonaldehyde yield was artificially increased in the mechanism to investigate the sensitivity of the pinonaldehyde on this yield.

**Technical comments**

Figure 6 caption

"...both combined with different fast chemical backward reactions $k_b$."
- This sentence is difficult to understand in general and might deserve revision, but I believe what you mean here is "differently".

Caption changed to:

Simulated SOA mass including a chemical backward reaction in the particle phase for liquid ($D_b =$

$10^{-12}\,\text{m}^2\,\text{s}^{-1}$) aerosol particles (case 7 of Table 1) with a) a fast chemical reaction in the particle phase $k_c=1\,\text{s}^{-1}$ and b) a reduced rate constant $k_c=10^{-2}\,\text{s}^{-1}$, whereby backward reactions with different rate constants $k_b$ are additionally included.

*"The reference simulations for the regarding $k_c$ and no $k_b$ are shown with dashed lines."*
- Do you mean "respective" instead of "regarding"? Do you mean "without backward reaction" instead of "no $k_b$"?

Caption changed to:

The reference simulations for the respective $k_c$ without backward reactions (indicated by "no $k_b$" in the key) are shown with dashed lines.

p. 18, l. 33ff
*"After 1 h simulation time, it is obvious that the simulated concentration profile agree well with the experimentally observed SOA mass with a backward reaction rate constant of $k_b=10^{-2}\,s^{-1}$."*
- Do the authors mean:
"After 1 h simulation time, it is obvious that the simulated concentration profile agrees well with the experimentally observed SOA mass when using a backward reaction rate constant of $k_b=10^{-2}\,\text{s}^{-1}$."

Sentence changed to:

After 1 h simulation time, it is obvious that the simulated concentration profile agrees well with the experimentally observed SOA mass when using a backward reaction rate constant of $k_b=10^{-2}\,\text{s}^{-1}$.

Figure S5b
- This figure is not discussed in the manuscript.

Sentence added:
For semi-solid particles and moderate particle reactions, the same trend is observed but only very low SOA concentrations are formed (see Fig S5b in the Supplement). Due to the particle-phase-diffusion-limited partitioning for semi-solid particles combined with only moderate particle-phase rate constants, the formed SOA mass reaches quite small values, which are not observed under typical chamber study experiment conditions.

---

## Author Response (AR2)

**Authors' Response to referee comments for 'Kinetic modeling studies for SOA formation from α-pinene ozonolysis'**

The authors would like to thank both referees and the co-editor for the careful consideration of the revised manuscript. According to the additional comment of referee #1 and the suggestion of the co-editor, the authors have further improved the manuscript. All comments and changes in the manuscript are addressed below.

Comments from referees are presented in blue, these are followed by authors' response in black, and changes to the manuscript are in green.

**Referee #1**

**Accepted subject to technical corrections:**

I share reviewer 2 concern in the first comment: "how the authors justify their assumption that organic molecules must diffuse into the particle phase in order to contribute to particle growth. This assumption is crucial for all conclusions in this paper and finds too little scrutiny." I don't question that you have implemented the kinetic approach as described by Zaveri et al. (2014) and Mai et al. (2015) but I am skeptical to the fact that this approach require diffusion into the particle bulk for particle growth to occur. In the extreme case of non-volatile vapors the particles should growth without particle diffusion. I don't understand why the particle surface layer is not considered to be part of the condensed phase and thus influence the uptake of the compounds according to Raoult's law. To me it seem as if it should be the particle surface layer composition which is in contact with the gas-phase that will determine the uptake of the gas phase species and not the bulk phase composition that is not in contact with the gas-phase. I still think you more clearly need to demonstrate that this assumption is correct and not just refer to the previous studies that use this assumption, which may be incorrect or at least only valid for specific condition. Alternatively I think you need to change your model approach and allow the particles to growth even without diffusion into the bulk phase. Otherwise I am satisfied with the response to my comments.

According to the comment of referee #1 as well as the suggestion of the co-editor, we have added a further explanation why diffusion influences the uptake process in the model and that "burying processes" are not comprised in the model. The following text is added to the revised manuscript:

[revised manuscript text omitted]